# LOCAL STEPS SPEED UP LOCAL GD FOR HETEROGENEOUS DISTRIBUTED LOGISTIC REGRESSION

**Michael Crawshaw**
George Mason University
mcrawsha@gmu.edu*

**Blake Woodworth**
George Washington University
blakewoodworth@gmail.com

**Mingrui Liu**
George Mason University
mingruil@gmu.edu*

## ABSTRACT

We analyze two variants of Local Gradient Descent applied to distributed logistic regression with heterogeneous, separable data and show convergence at the rate $O(1/KR)$ for $K$ local steps and sufficiently large $R$ communication rounds. In contrast, all existing convergence guarantees for Local GD applied to any problem are at least $\Omega(1/R)$, meaning they fail to show the benefit of local updates. The key to our improved guarantee is showing progress on the logistic regression objective when using a large stepsize $\eta \gg 1/K$, whereas prior analysis depends on $\eta \leq 1/K$.

## 1 INTRODUCTION

In the practice of distributed optimization, local model updates are crucial for reducing communication cost. The standard distributed optimization algorithm is Local SGD (a.k.a., FedAvg) (McMahan et al., 2017; Stich, 2019; Lin et al., 2019; Koloskova et al., 2020; Patel et al., 2024), in which each communication round consists of $K$ SGD updates to each client model, followed by an aggregation step where local models are averaged. A practitioner using Local SGD can decrease the number of communication rounds $R$ while maintaining the same computational cost ($KR$ sequential gradient computations) by increasing the number of local steps $K$, accelerating optimization when communication is expensive, such as in federated learning (McMahan et al., 2017; Kairouz et al., 2021).

However, recent work characterizes the complexity of Local SGD for optimizing smooth, convex objectives under various heterogeneity assumptions (Woodworth et al., 2020a; Koloskova et al., 2020; Glasgow et al., 2022; Patel et al., 2024), showing that the worst-case communication complexity cannot be improved by increasing $K$: the dominating terms of the convergence rate do not depend on $K$. Even when the algorithm can access full gradients, increasing local steps does not decrease the number of rounds required to find an $\epsilon$-approximate solution, according to these guarantees.

Crucially, these complexity results should be interpreted in the context of the assumptions on which they rely: these works consider the worst-case over large classes of problems satisfying convexity, smoothness, and various heterogeneity assumptions. Therefore, the worst-case complexity may not be representative for particular problems that are relevant in practice. Many works (Haddadpour & Mahdavi, 2019; Woodworth et al., 2020b; Koloskova et al., 2020; Glasgow et al., 2022; Wang et al., 2022; Patel et al., 2023; 2024) approach this by modifying the problem class, in particular by trying to find the "right" heterogeneity assumptions that reflect objectives for practically relevant problems, where local steps are observed to help. We take an orthogonal approach, by directly considering a distributed version of a classical machine learning problem. The central question of our paper is:

*Can local steps provably accelerate Local GD for distributed logistic regression?*

According to empirical observation (e.g., Figure 1 in Woodworth et al. (2020b)), the answer may be positive. However, existing theory is insufficient to demonstrate such an acceleration, even for this simple case involving deterministic gradients and a linear model. The existing guarantees (see Section 3.2) require a small learning rate $\eta \leq 1/K$, so that increasing the number of local steps is counteracted by decreasing the learning rate, leading to no change in the dominating term of the

---

*Corresponding author.

Table 1: Communication rounds to find an $\epsilon$-approximate solution for distributed logistic regression. $K$ is the number of local steps, $M$ is the number of clients, and $\gamma$ is the maximum margin of the combined dataset. $(a)$ these rates are derived by extending the results of Woodworth et al. (2020b); Koloskova et al. (2020) to remove the assumption of existence of a global minimum (see Section 3.2). $(b)$ this result holds for the case of $M = 2$ and $n = 1$ data point per client. $(c)$ $C, \phi$, and $\beta$ depend on the dataset, and $\alpha = 1/\sqrt{\log(1/(C\epsilon))}$. Notice that $\alpha \to 0$ at a logarithmic rate as $\epsilon \to 0$.

| | Fixed $K$ | Best $K$ |
|---|---|---|
| Local GD (Woodworth et al., 2020b)$^{(a)}$ | $\tilde{\mathcal{O}}\left(\frac{1}{\gamma^2 \epsilon^{3/2}}\right)$ | $\tilde{\mathcal{O}}\left(\frac{1}{\gamma^2 \epsilon^{3/2}}\right)$ |
| Local GD (Koloskova et al., 2020)$^{(a)}$ | $\tilde{\mathcal{O}}\left(\frac{1}{\gamma^2 \epsilon} + \frac{1}{\gamma \epsilon^{3/4}}\right)$ | $\tilde{\mathcal{O}}\left(\frac{1}{\gamma^2 \epsilon} + \frac{1}{\gamma \epsilon^{3/4}}\right)$ |
| Two-Stage Local GD (Theorem 1) | $\tilde{\mathcal{O}}\left(\frac{KM}{\gamma^4} + \frac{1}{\gamma^2 K \epsilon}\right)$ | $\tilde{\mathcal{O}}\left(\frac{M^{1/2}}{\gamma^3 \epsilon^{1/2}}\right)$ |
| Local Gradient Flow (Theorem 2)$^{(b)}$ | $\tilde{\mathcal{O}}\left(C\phi(\eta K)^{\beta \log(\eta K)} + \frac{\phi}{\eta K \epsilon}\right)^{(c)}$ | $\tilde{\mathcal{O}}\left(\frac{\phi C^\alpha}{\epsilon^{1-\alpha}}\right)^{(c)}$ |

convergence rate. The best convergence rate from baseline analysis is $\tilde{\mathcal{O}}(1/(\gamma^2 R))$, where $\gamma$ is the maximum margin of the combined dataset (see Corollary 2).

**Contributions**. In this paper, we demonstrate that for distributed logistic regression, local steps can accelerate a two-stage variant of Local GD that uses learning rate warmup (Algorithm 2). Our analysis leverages properties of the logistic loss function, particularly that the "smoothness constant" of the loss decreases with the loss itself. This allows us to use a large learning rate after a warmup stage, and we can avoid the requirement $\eta \leq 1/K$. After warming up for $\mathcal{O}(KM/\gamma^4)$ rounds, the algorithm converges at a rate of $\mathcal{O}(1/(\gamma^2 KR))$. This result provides a guarantee for the particular problem of logistic regression, but it also suggests a possible insight for distributed optimization in general: the observed benefit of local steps could be due to properties of the loss landscape, rather than to similarity of client objectives. See Section 7 for further discussion of this point.

Additionally, we provide preliminary results for a variant of Local GD with a constant learning rate which uses gradient flow for local client updates, which we refer to as Local Gradient Flow (Algorithm 3). In the special case with $M = 2$ clients and $n = 1$ data points per client, we use a novel Lyapunov analysis to show that after sufficiently many rounds, Local GF converges at a rate of $\tilde{\mathcal{O}}(1/KR)$ (ignoring constants depending on the dataset).

**Organization**. The rest of the paper is structured as follows. Related work and preliminaries are discussed in Sections 2 and 3, respectively. Our main result, the convergence of Two-Stage Local GD, is presented in Section 4, while our analysis of Local GF is presented in Section 5. Section 6 contains experimental results, and we conclude with a discussion in Section 7.

**Notation**. $\|\boldsymbol{A}\|$ denotes the spectral norm for $\boldsymbol{A} \in \mathbb{R}^{d \times d}$, and $[n] := \{1, \ldots, n\}$. Beyond the abstract, $\mathcal{O}, \Omega$ and $\Theta$ only omit universal constants unless explicitly stated. Similarly, $\tilde{\mathcal{O}}, \tilde{\Omega}$, and $\tilde{\Theta}$ only omit universal constants and logarithmic terms.

## 2 RELATED WORK

**Distributed Convex Optimization.** Distributed convex optimization has been an active area of research for more than a decade, with early work that leverages parallelization for learning problems (Mcdonald et al., 2009; McDonald et al., 2010; Zinkevich et al., 2010; Dekel et al., 2012; Balcan et al., 2012; Shamir & Srebro, 2014). The concept of federated learning and the FedAvg algorithm were proposed by McMahan et al. (2017), which focuses on the machine learning setting where many clients collaboratively train a model without uploading their data to maintain privacy. The convergence analysis of FedAvg (a.k.a., local SGD) in the convex optimization setting was proved by Stich (2018); Woodworth et al. (2020a;b); Khaled et al. (2020); Koloskova et al. (2020); Glasgow et al. (2022). For a comprehensive survey for federated learning and distributed optimization

---

**Algorithm 1** Local GD

---

**Input:** Initialization $\bar{w}_0 \in \mathbb{R}^d$, rounds $R \in \mathbb{N}$, local steps $K \in \mathbb{N}$, learning rate $\eta > 0$, averaging weights $\{\alpha_{r,k}\}_{r,k}$

1: **for** $r = 0, 1, \ldots, R-1$ **do**
2:     **for** $m \in [M]$ **do**
3:         $\boldsymbol{w}_{r,0}^m \leftarrow \bar{w}_r$
4:         **for** $k = 0, \ldots, K-1$ **do**
5:             $\boldsymbol{w}_{r,k+1}^m \leftarrow \boldsymbol{w}_{r,k}^m - \eta \nabla F_m(\boldsymbol{w}_{r,k}^m)$
6:         **end for**
7:     **end for**
8:     $\bar{w}_{r+1} \leftarrow \frac{1}{M} \sum_{m=1}^M \boldsymbol{w}_{r,K}^m$
9: **end for**
10: **return** $\hat{w} = \sum_{r=0}^{R-1} \sum_{k=0}^{K-1} \alpha_{r,k} \left( \frac{1}{M} \sum_{m=1}^M \boldsymbol{w}_{r,k}^m \right)$

---

algorithms, we refer the readers to Kairouz et al. (2019); Wang et al. (2021) and references therein. The lower bounds of distributed convex optimization were studied in Zhang et al. (2013); Arjevani & Shamir (2015); Woodworth et al. (2018; 2021); Glasgow et al. (2022); Patel et al. (2024).

**Local SGD and Other Baselines.** There are several alternative algorithms for local SGD under various settings, including minibatch SGD (Dekel et al., 2012), accelerated minibatch SGD (Ghadimi & Lan, 2012), SCAFFOLD (Karimireddy et al., 2020), SlowcalSGD (Levy, 2023), federated accelerated SGD (Yuan & Ma, 2020). With convex, smooth, heterogeneous objectives, Patel et al. (2024) show that the existing analysis of local SGD (Koloskova et al., 2020; Glasgow et al., 2022) achieves the lower bound of local SGD, and that accelerated minibatch SGD is minimax optimal. This means that local SGD does not benefit from local updates in the worst case. Other algorithms can provably benefit from local steps. Levy (2023) show that SlowcalSGD provably benefits from local updates and is better than minibatch SGD and local SGD. Mishchenko et al. (2022) show that the ProxSkip algorithm can lead to communication acceleration by local gradient steps for strongly convex functions with deterministic gradient oracles and closed-form proximal mapping.

**Gradient Methods for Logistic Regression.** Early work studied the implicit bias of GD with small stepsize for logistic regression and exponentially-tailed loss functions (Soudry et al., 2018; Ji & Telgarsky, 2018). In the context of logistic regression, Gunasekar et al. (2018) studied the implicit bias of generic optimization methods. Ji et al. (2021) studied a momentum-based method for fast margin maximization. Nacson et al. (2019) studied the implicit bias of SGD. Recently, Wu et al. (2024b;a) studied the implicit bias and convergence rate of GD for logistic regression when the learning rate is large (i.e., the Edge-of-Stability regime (Cohen et al., 2021)). Logistic regression has also been studied under self-concordance (Nesterov, 2013; Bach, 2014), which resembles the properties of the logistic loss function that we use for our analysis, although we do not directly use self-concordance.

## 3 PRELIMINARIES

### 3.1 PROBLEM SETUP

We consider a distributed version of the linearly separable binary classification problem. Let $M \in \mathbb{N}$ be the number of clients, $d \in \mathbb{N}$ be the dimension of the input data, and $n \in \mathbb{N}$ be the number of samples per client. Then each client $m \in [M]$ will have a "local" dataset $D_m = \{(\boldsymbol{x}_{mi}, y_{mi})\}_{i=1}^n$, where $\boldsymbol{x}_{mi} \in \mathbb{R}^d$ and $y_i \in \{-1, 1\}$. We consider the logistic regression objective for this classification problem. Denoting $\ell(z) = \log(1 + \exp(-z))$, the objective for client $m$ is defined as

$$F_m(\boldsymbol{w}) = \frac{1}{n} \sum_{i=1}^n \ell(y_{mi}\langle \boldsymbol{w}, \boldsymbol{x}_{mi} \rangle), \tag{1}$$

and as usual the global objective is $F(\boldsymbol{w}) = \frac{1}{M} \sum_{m=1}^M F_m(\boldsymbol{w})$. Linear separability means that there exists some $\boldsymbol{w} \in \mathbb{R}^d$ such that $y_{mi}\langle \boldsymbol{w}, \boldsymbol{x}_{mi} \rangle > 0$ for all $m \in [M]$ and $i \in [n]$, that is, there exists a

solution $\boldsymbol{w}$ that correctly classifies the data from all clients simultaneously. We denote the maximum margin of the combined dataset and the corresponding classifier as

$$\gamma := \max_{\|\boldsymbol{w}\|=1} \min_{m\in[M],i\in[n]} y_{mi}\langle\boldsymbol{w},\boldsymbol{x}_{mi}\rangle, \quad \boldsymbol{w}_* = \arg\max_{\|\boldsymbol{w}\|=1} \min_{m\in[M],i\in[n]} y_{mi}\langle\boldsymbol{w},\boldsymbol{x}_{mi}\rangle. \tag{2}$$

We assume without loss of generality that $y_{mi} = 1$ for all $m, i$. Since the objective depends only on $y_{mi}\boldsymbol{x}_{mi}$, we can replace every input $\boldsymbol{x}_{mi}$ with $y_{mi}\boldsymbol{x}_{mi}$ and every label $y_{mi}$ with 1. We also assume that $\|\boldsymbol{x}_{mi}\| \leq 1$ for all $m, i$, which can be enforced by scaling each $\boldsymbol{x}_{mi}$ by the maximum data norm.

## 3.2 BASELINE GUARANTEES

Previous analysis of Local SGD for smooth, convex objectives (Woodworth et al., 2020b; Koloskova et al., 2020) additionally assumes the existence of a minimizer $\boldsymbol{w}_*$, which is not satisfied by logistic regression. To establish a baseline rate for logistic regression with Local GD, we modify these two analyses of Local SGD by removing the assumption of a global minimizer.

These two analyses use different assumptions on the objective heterogeneity, both of which can be applied to logistic regression; we therefore modify both analyses for the case where a global minimum may not exist. We use the approach outlined in Orabona (2024), which modifies the standard SGD analysis by replacing $\boldsymbol{w}_*$ with a comparator $\boldsymbol{u} \in \mathbb{R}^d$. The full results and proofs for this baseline analysis are given in Appendix C. Corollary 1 follows from Theorem 5 (extension of (Woodworth et al., 2020b)), and Corollary 2 follows from Theorem 6 (extension of (Koloskova et al., 2020)).

**Corollary 1.** *For distributed logistic regression, Local GD with $\eta = \tilde{\Theta}(1/(\gamma^{2/3}KR^{1/3}))$ satisfies*

$$F(\hat{\boldsymbol{w}}) \leq \tilde{\mathcal{O}}\left(\frac{1}{\gamma^2 KR} + \frac{1}{\gamma^{4/3}R^{2/3}}\right). \tag{3}$$

**Corollary 2.** *For distributed logistic regression, Local GD with $\eta = \tilde{\Theta}(1/K)$ satisfies*

$$F(\hat{\boldsymbol{w}}) \leq \tilde{\mathcal{O}}\left(\frac{1}{\gamma^2 R} + \frac{1}{\gamma^{4/3}R^{4/3}}\right). \tag{4}$$

Importantly, the dominating term in both upper bounds is not decreased by increasing the number of local steps $K$. This aligns with the worst-case rates of Local GD in the case that a minimizer does exist (Patel et al., 2024). Notice that in both cases, the choice of learning rate $\eta$ has a $1/K$ dependence on the number of local steps $K$, so increasing the number of local steps is countered by decreasing the learning rate. Therefore, the existing worst-case analysis cannot show that local steps can increase optimization performance of Local GD for distributed logistic regression.

Alternatively, we can establish a baseline by applying these analyses to a regularized version of our problem. This approach leads to similar complexities as in Corollaries 1 and 2 (see Appendix C.6).

## 4 CONVERGENCE OF TWO-STAGE LOCAL GD

In this section, we analyze a two-stage variant of Local GD, defined in Algorithm 2. This algorithm essentially runs Local GD twice, using the output of the first stage as the initialization for the second stage. In order to achieve an improved convergence rate, this algorithm uses a small learning rate $\eta_1$ in the first phase, and a large learning rate $\eta_2 \leq 4$ in the second phase. For sufficiently large $R$, the convergence rate of Two-Stage Local GD is $\mathcal{O}(1/(\eta_2\gamma^2 KR))$. The result is stated in Theorem 1, a sketch of the proof is given in Section 4.2, and the complete proof is contained in Appendix A.

## 4.1 STATEMENT OF RESULTS

**Theorem 1.** *Let $0 < \eta_2 \leq 4$, and*

$$r_0 \geq \tilde{\mathcal{O}}\left(\frac{\eta_2 KM}{\gamma^4} + \frac{(\eta_2 KM)^{3/4}}{\gamma^{5/2}}\right), \quad \eta_1 = \tilde{\Theta}\left(\min\left\{\frac{1}{K}, \frac{\eta_2^{1/3}M^{1/3}}{\gamma^2 K^{2/3}}\right\}\right), \tag{5}$$

*and $R \geq r_0$. Then Two-Stage Local GD (Algorithm 2) satisfies*

$$F(\hat{\boldsymbol{w}}_2) \leq \frac{2}{\eta_2\gamma^2 K(R - r_0)}. \tag{6}$$

---

**Algorithm 2** Two-Stage Local GD

---

**Input:** Initialization $\bar{\boldsymbol{w}}_0$, rounds $R \in \mathbb{N}$, local steps $K \in \mathbb{N}$, learning rates $\eta_1, \eta_2 > 0$, averaging weights $\{\alpha_{r,k}\}_{r,k}$, phase 1 rounds $r_0 \in \mathbb{N}$
1: $r_1 \leftarrow R - r_0$
2: $\beta_{r,k} \leftarrow \mathbb{1}\{r = R - 1 \text{ and } k = 0\}$
3: $\hat{\boldsymbol{w}}_1 \leftarrow \text{Local GD}(\bar{\boldsymbol{w}}_0, r_0, K, \eta_1, \{\alpha_{r,k}\}_{r,k})$
4: $\hat{\boldsymbol{w}}_2 \leftarrow \text{Local GD}(\hat{\boldsymbol{w}}_1, r_1, K, \eta_2, \{\beta_{r,k}\}_{r,k})$
5: **return** $\hat{\boldsymbol{w}}_2$

---

Notice that $\eta_2 K$ appears in the denominator of the second stage convergence rate, but importantly, the choice of $\eta_2$ is not constrained by $K$. The constraint $\eta_2 \leq 4$ comes from the fact that $F$ is $H$-smooth with $H = 1/4$, so that we require $\eta_2 \leq 1/H$. This means that we can set $\eta_2 = \Theta(1)$ and choose a large $K$ in order to speed up convergence. In other words, Two-Stage Local GD benefits from local steps. This is made formal in the following result.

**Corollary 3.** *Let $\epsilon > 0$. With $\eta_2 = 1, r_0 = \tilde{\Theta}(KM/\gamma^4 + (KM)^{3/4}/\gamma^{5/2})$, and $\eta_1$ chosen as in Theorem 1, the output of Two-Stage Local GD satisfies $F(\hat{\boldsymbol{w}}_2) \leq \epsilon$ as long as*

$$R \geq \tilde{\Omega}\left(\frac{KM}{\gamma^4} + \frac{(KM)^{3/4}}{\gamma^{5/2}} + \frac{1}{\gamma^2 K\epsilon}\right). \tag{7}$$

*Further, if we choose $K = \Theta(\gamma/\sqrt{M\epsilon})$, then $F(\hat{\boldsymbol{w}}_2) \leq \epsilon$ as long as*

$$R \geq \tilde{\Omega}\left(\frac{M^{1/2}}{\gamma^3 \epsilon^{1/2}} + \frac{1}{\gamma^{7/4}\epsilon^{3/8}}\right). \tag{8}$$

Corollary 3 also describes the number of rounds $r_0$ to transition from a small learning rate to a large one. With $\eta_2 = \Theta(1)$, the first stage requires $r_0 = \Omega(KM/\gamma^4)$ rounds. This aligns with the intuition that increasing $K$ necessitates a longer warmup before a large learning rate can be used without creating instability. Lastly, notice from Algorithm 2 that the output $\hat{\boldsymbol{w}}_2$ is the last iterate from the second stage. Therefore, Theorem 1 gives a last-iterate guarantee for Two-Stage Local GD, whereas the baseline analyses (Corollaries 1 and 2) provide average-iterate guarantees.

## 4.2 PROOF SKETCH

The main idea of the proof is to leverage the relationship between the loss value, first derivative, and second derivative, namely that $0 < \ell''(z) < |\ell'(z)| < \ell(z)$ for the logistic loss $\ell = \log(1 + \exp(-z))$ (see Lemma 24). This yields a similar relationship between the derivatives of the objective $F$:

$$\|\nabla F(\boldsymbol{w})\| \leq F(\boldsymbol{w}), \quad \|\nabla^2 F(\boldsymbol{w})\| \leq F(\boldsymbol{w}), \tag{9}$$

see Lemma 25. Intuitively, when the objective $F(\bar{\boldsymbol{w}}_r)$ is small, the Hessian $\|\nabla^2 F(\bar{\boldsymbol{w}}_r)\|$ is also small, so a large learning rate can be used while ensuring a decrease in the global objective.

Accordingly, we apply Corollary 2 to guarantee that the objective value after the first phase $F(\hat{\boldsymbol{w}}_1)$ is sufficiently small. For the second phase, we treat each round's update $\bar{\boldsymbol{w}}_{r+1} - \bar{\boldsymbol{w}}_r$ as a biased gradient step on $F$, with bias introduced by heterogeneous local objectives. With small local Hessians $\|\nabla^2 F_m(\bar{\boldsymbol{w}}_r)\|$, we can further bound the change of local gradient within a round, i.e. $\|\nabla F_m(\boldsymbol{w}_{r,k}^m) - \nabla F_m(\bar{\boldsymbol{w}}_r)\|$, thereby bounding the bias of the aforementioned biased gradient step. We elaborate on this idea below. All results in this section apply to Local GD, and the application to Two-Stage Local GD will occur at the end of the proof.

We want to bound the bias of $\bar{\boldsymbol{w}}_{r+1} - \bar{\boldsymbol{w}}_r$ as a gradient step on $F$, that is, we want to bound

$$B_r := \|(\bar{\boldsymbol{w}}_{r+1} - \bar{\boldsymbol{w}}_r) + \eta K \nabla F(\bar{\boldsymbol{w}}_r)\| = \left\|\frac{\eta}{M}\sum_{m=1}^{M}\sum_{k=0}^{K-1}(\nabla F_m(\boldsymbol{w}_{r,k}^m) - \nabla F_m(\bar{\boldsymbol{w}}_r))\right\| \tag{10}$$

$$\leq \eta K \frac{1}{MK}\sum_{m=1}^{M}\sum_{k=0}^{K-1}\left\|\nabla F_m(\boldsymbol{w}_{r,k}^m) - \nabla F_m(\bar{\boldsymbol{w}}_r)\right\| \tag{11}$$

$$\leq \eta K \frac{1}{MK} \sum_{m=1}^{M} \sum_{k=0}^{K-1} \left( \underbrace{\sup_{t \in [0,1]} \left\| \nabla^2 F_m(t \boldsymbol{w}_{r,k}^m + (1-t) \bar{\boldsymbol{w}}_r) \right\|}_{A_1} \right) \underbrace{\left\| \boldsymbol{w}_{r,k}^m - \bar{\boldsymbol{w}}_r \right\|}_{A_2}. \tag{12}$$

First, Equation 9 provides a bound on $\|\nabla^2 F(\bar{\boldsymbol{w}}_r)\|$, but not immediately on $\|\nabla^2 F(\boldsymbol{w})\|$ for $\boldsymbol{w}$ close to $\bar{\boldsymbol{w}}_r$; such a bound is needed to bound $A_1$. The following lemma bounds the Hessian of $F$ in a neighborhood of a point $\boldsymbol{w}_1$.

**Lemma 1.** *For all $\boldsymbol{w}_1, \boldsymbol{w}_2 \in \mathbb{R}^d$ and all $m \in [M]$,*

$$\|\nabla^2 F_m(\boldsymbol{w}_2)\| \leq F_m(\boldsymbol{w}_1) \left( 1 + \|\boldsymbol{w}_2 - \boldsymbol{w}_1\| \left( 1 + \exp(\|\boldsymbol{w}_2 - \boldsymbol{w}_1\|^2) \left( 1 + \frac{1}{2} \|\boldsymbol{w}_2 - \boldsymbol{w}_1\|^2 \right) \right) \right). \tag{13}$$

To prove Lemma 1, we bound $\|\nabla^2 F_m(\boldsymbol{w}_2)\| \leq F_m(\boldsymbol{w}_2)$ with Lemma 25, then bound $F_m(\boldsymbol{w}_2)$ by a second-order Taylor series of $F_m$ centered at $\boldsymbol{w}_1$. The quadratic term of this Taylor series depends on $\|\nabla^2 F_m(\boldsymbol{w})\|$ for all $\boldsymbol{w}$ between $\boldsymbol{w}_1$ and $\boldsymbol{w}_2$, so we have an integral inequality in $\|\nabla^2 F_m(\boldsymbol{w})\|$. Applying a variation of Gronwall's inequality yields Lemma 1. With Lemma 1, the task of bounding $A_1$ and $A_2$ is reduced to bounding $\|\boldsymbol{w}_{r,k}^m - \bar{\boldsymbol{w}}_r\|$. This is achieved by the following lemma.

**Lemma 2.** *If $\eta \leq 8$, then $\|\boldsymbol{w}_{r,k}^m - \bar{\boldsymbol{w}}_r\| \leq \eta K F_m(\bar{\boldsymbol{w}}_r)$ for every $r \geq 0$ and $k \leq K$.*

The idea of the proof of Lemma 2 is to show that each local step does not increase the local objective, so $F_m(\boldsymbol{w}_{r,k}^m) \leq F_m(\bar{\boldsymbol{w}}_r)$. Combining this with $\|\nabla F_m(\boldsymbol{w})\| \leq F_m(\boldsymbol{w})$ (Equation 9),

$$\|\nabla F_m(\boldsymbol{w}_{r,k}^m)\| \leq F_m(\boldsymbol{w}_{r,k}^m) \leq F_m(\bar{\boldsymbol{w}}_r), \tag{14}$$

which we can use to upper bound $\|\boldsymbol{w}_{r,k}^m - \bar{\boldsymbol{w}}_r\| \leq \eta \sum_{j=0}^{k-1} \|\nabla F_m(\boldsymbol{w}_{r,k}^m)\| \leq \eta K F_m(\bar{\boldsymbol{w}}_r)$. Combining this with Lemma 1 yields a bound for $A_1$ and $A_2$.

**Lemma 3.** *If $\eta \leq 8$, and $F(\bar{\boldsymbol{w}}_r) \leq 1/(\eta K M)$, then for every $m \in [M]$ and $k \in [K]$,*

$$\|\nabla F_m(\boldsymbol{w}_{r,k}^m) - \nabla F_m(\bar{\boldsymbol{w}}_r)\| \leq 7\eta K F_m(\bar{\boldsymbol{w}}_r)^2. \tag{15}$$

Note that the condition $F(\bar{\boldsymbol{w}}_r) \leq 1/(\eta K M)$ in Lemma 3 ensures that the RHS of Equation 13 is $\mathcal{O}(F_m(\bar{\boldsymbol{w}}_1))$. Plugging Equation 15 back to Equation 12 yields a bound for the bias $B_r$ as

$$B_r \leq 7\eta^2 K^2 F_m(\bar{\boldsymbol{w}}_r)^2. \tag{16}$$

With this bound of the bias, we can bound $F(\bar{\boldsymbol{w}}_{r+1}) - F(\bar{\boldsymbol{w}}_r)$ using classical techniques.

**Lemma 4.** *Suppose that $\eta \leq 4$ and $F(\bar{\boldsymbol{w}}_0) \leq \gamma^2/(42\eta K M)$. Then for every $r \geq 0$,*

$$F(\bar{\boldsymbol{w}}_r) \leq \frac{2}{\eta \gamma^2 K r}. \tag{17}$$

Finally, with Lemma 4, we can analyze Two-Stage Local GD. First, we use Corollary 2 to guarantee that the first stage output $\hat{\boldsymbol{w}}_1$ satisfies the condition of Lemma 4, i.e. $F(\hat{\boldsymbol{w}}_1) \leq \gamma^2/(42\eta_2 K M)$. Then, we can apply Lemma 4 to the second stage, which gives exactly the conclusion of Theorem 1.

## 5 CONVERGENCE OF LOCAL GRADIENT FLOW

Section 4 shows that Local GD with a two-stage learning rate can achieve a convergence rate with dominating term $\mathcal{O}(1/(\gamma^2 K R))$, by initially using a small learning rate, then transitioning to a larger one. However, experiments show that Local GD can converge with a fixed, large learning rate, albeit with non-monotonicity of the global objective early in training (see Figure 1a). It is then natural to ask whether Local GD can provably converge with sufficient rounds for any fixed $\eta$.

In this section, we make progress towards answering this question by considering the special case of $M = 2$ clients, $n = 1$ data point per client, and a variant of Local GD which we refer to as Local Gradient Flow (defined in Algorithm 3). In each round of Local GF, client models are updated by running gradient flow on each local objective for $K$ units of time. The global model is updated in the same way as Local GD, i.e. by setting the new global model as the average of updated client models. Our main result for this section is Theorem 2, which shows that for sufficiently large $R$, Local GF converges at a rate of $\tilde{\mathcal{O}}(1/\eta K R))$ (ignoring constants that depend on the dataset).

---

**Algorithm 3** Local Gradient Flow

---

**Input:** Initialization $\bar{w}_0 \in \mathbb{R}^d$, rounds $R \in \mathbb{N}$, local steps $K \in \mathbb{N}$, learning rate $\eta > 0$
 1: **for** $r = 0, 1, \ldots, R - 1$ **do**
 2:    **for** $m \in [M]$ **do**
 3:       Set $w_r^m(t)$ as the solution to:

$$w_r^m(0) = \bar{w}_r \qquad\qquad \dot{w}_r^m(t) = -\eta \nabla F_m(w_r^m(t))$$

 4:    **end for**
 5:    $\bar{w}_{r+1} \leftarrow \frac{1}{M} \sum_{m=1}^M w_r^m(K)$
 6: **end for**
 7: **return** $\bar{w}_R$

---

### 5.1 STATEMENT OF RESULTS

For the case $n = 1$, we re-index the local data as $x_1, \ldots x_M$, and denote $\gamma_m = \|x_m\|$ and $w_*^m = x_m / \|x_m\|$. Recall our assumption that all data points have label 1. Then each local objective $F_m$ is

$$F_m(w) = \log(1 + \exp(-\langle w, x_m \rangle)) = \log(1 + \exp(-\gamma_m \langle w, w_*^m \rangle)). \tag{18}$$

Define $W \in \mathbb{R}^{M \times d}$ so that the $m$-th row equals $w_m^*$, and define $G = WW^\mathsf{T} \in \mathbb{R}^{M \times M}$. For $M = 2$ clients, the Gram matrix $G$ is parameterized by a scalar, that is, $G = \begin{pmatrix} 1 & c \\ c & 1 \end{pmatrix}$, where $c = \langle w_1^*, w_2^* \rangle$. Notice that, up to rotation of the data, a dataset is characterized by $\gamma_1, \gamma_2$, and $c$: the magnitudes of $\gamma_1, \gamma_2$ affect the relative sizes of local updates, and $c$ determines the angle between the local client updates. In what follows, we denote $\gamma_{\min} = \min\{\gamma_1, \gamma_2\}$ and $\gamma_{\max} = \max\{\gamma_1, \gamma_2\}$.

**Theorem 2.** *Define*

$$L_0 = \max_{m \in [M]} \frac{1}{\gamma_m} \log\left(1 + \eta K \gamma_m^2\right), \qquad\qquad H_0 = \min_{m \in [M]} \frac{1}{\gamma_m} \log\left(1 + \eta K \gamma_m^2\right), \tag{19}$$

*and*

$$\tau = \frac{16(L_0 + 1)^2}{(1 + c)\gamma_{\min}} \left( \left(\frac{1}{H_0} - \frac{1}{L_0}\right) \left(\frac{L_0}{H_0}\right)^{\frac{3(L_0+1)^2(1-c)\gamma_{\max}}{(1+c)\gamma_{\min}}} + 4\gamma_{\max} + \frac{2}{H_0} \right). \tag{20}$$

*Then for every $r \geq \tau$, Local GD initialized with $\bar{w}_0 = 0$ satisfies*

$$F(\bar{w}_r) \leq \frac{32(1 + \log(1 + \eta K))^2}{(1 + c)\gamma_{\min}^4 \eta K (r - \tau)}. \tag{21}$$

Theorem 2 shows that Local GF will converge at the desired rate after $\tau$ rounds. so $\tau + 1/((1 + c)\gamma_{\min}^4 \eta K \epsilon)$ rounds are sufficient to find an $\epsilon$-approximate solution. The transition time $\tau$ can be bounded as $\tau \leq B(1 + \eta K)^{\beta \log(1 + \eta K)}$, where $B = \mathrm{poly}(\exp(1/\gamma_{\min}), \exp(1/(1 + c)))$ and $\beta = \mathrm{poly}(1/\gamma_{\min}, 1/(1 + c))$. Denoting $C = B(1 + c)\gamma_{\min}^4$, we can then choose $\eta K = \tilde{\Theta}\left(\exp(\sqrt{\log(1/(C\epsilon))/\beta})\right)$, which yields a communication cost of

$$R \leq \tilde{\mathcal{O}}\left( \frac{\exp\left(-\sqrt{\log(1/(C\epsilon))}\right)}{(1 + c)\gamma_{\min}^4 \epsilon} \right) = \tilde{\mathcal{O}}\left( \frac{C^\alpha}{(1 + c)\gamma_{\min}^4 \epsilon^{1-\alpha}} \right), \tag{22}$$

where $\alpha = 1/\sqrt{\log(1/(C\epsilon))}$. Therefore, the communication cost $R$ has dependence $\epsilon^{-(1-\alpha)}$ on $\epsilon$, which is smaller than the $\epsilon^{-1}$ cost guaranteed by existing baselines (see Table 1). The exponent $1 - \alpha$ goes to 1 from below at a logarithmic rate as $\epsilon \to 0$.

### 5.2 PROOF SKETCH

To prove Theorem 2, we construct a novel Lyapunov function $L_r$ with respect to the update of each communication round, show that it converges to 0 at a rate of $\mathcal{O}(1/r)$, and bound the client losses in terms of the Lyapunov function, i.e. $F_m(\bar{w}_r) \leq \mathcal{O}(L_r/(\eta K \gamma_m))$ when $L_r$ is sufficiently small.

Our Lyapunov function is defined in terms of a surrogate loss for each client. As observed experimentally (see Figure 1a), the client losses $F_m(\bar{w}_r)$ may not decrease monotonically. In particular, a round update $\bar{w}_{r+1} - \bar{w}_r$ can be dominated by the local update $w_r^m(K) - \bar{w}_r$ of a single client $m$ even when the local loss $F_m(\bar{w}_r)$ is small compared to the other client. Essentially, one client might be implicitly prioritized based on the relative magnitudes of (and angle between) the client data.

Our surrogate losses are designed to capture this implicit prioritization of clients. Denote $a_r^m = \langle \bar{w}_r, w_m^* \rangle$, so that $F_m(\bar{w}_r) = \log(1 + \exp(-\gamma_m a_r^m))$. Then, letting $W$ denote the Lambert W function, we define the surrogate client losses as

$$\rho_r^m = \frac{1}{\gamma_m} \log \left( \frac{W(\exp(\eta K \gamma_m^2 + \exp(\gamma_m a_r^m) + \gamma_m a_r^m))}{\exp(\gamma_m a_r^m)} \right). \tag{23}$$

Just as with the original losses $F_m(\bar{w}_r)$, the surrogate losses $\rho_r^m$ are monotonically decreasing functions of $a_r^m$ (see Lemma 30). Also, the client loss can be bounded in terms of the surrogate loss, provided the surrogate loss is sufficiently small (see Lemma 12). However, the surrogate losses are distinguished from the original losses by the following useful property: if at round $r$, client $m$ has the largest surrogate loss among all clients, then the surrogate loss for client $m$ will decrease from round $r$ to round $r + 1$. This suggests the following Lyapunov function: $L_r = \max_{m \in [M]} \rho_r^m$. The following lemma demonstrates such properties of the surrogate losses and Lyapunov function.

**Lemma 5.** $L_{r+1} \leq L_r$ for every $r \geq 0$. Further, if $\rho_r^m = \max_{m' \in [M]} \rho_r^{m'}$, then

$$\rho_{r+1}^m \leq L_r - \frac{(1+c)\gamma_m}{4(L_0+1)^2(1+\exp(-\gamma_m a_r^m))} L_r^2, \tag{24}$$

and if $\rho_{r+1}^m \geq \rho_r^m$, then $\rho_{r+1}^m \leq ((1-c)/2)L_r$.

Since the above lemma doesn't provide an upper bound on the surrogate losses $\rho_r^m$ for every pair of $r, m$, it doesn't immediately yield a convergence rate for the Lyapunov function $L_r$. However, we can use the previous lemma to upper bound the change in $L_r$ after two consecutive rounds, and applying this recursively yields the following lemma.

**Lemma 6.** Denote $m_r = \arg\max_{m \in [M]} \rho_r^m$ and $\alpha_r = a_r^{m_r}$. Let $0 \leq q \leq r$. If $\alpha_s \geq -A$ for every $q \leq s \leq r$, then

$$L_r \leq \frac{1}{1/L_q + \nu(r-q)/2}, \tag{25}$$

where $\nu := (1+c)\gamma_{\min}/(4(L_0+1)^2(1+\exp(\gamma_{\max}A)))$.

Notice that the above lemma gives an upper bound of $L_r$ which depends on some constant $A$ which lower bounds $a_r^m$. However, we do not have an a priori lower bound for $a_r^m$; while $a_0^m = 0$ for every $m$, it is possible that $a_r^m$ becomes negative in early rounds. To address this, we can combine Lemmas 5 and 6 to show that the decrease of the Lyapunov function gives a lower bound $a_r^m \geq -A_0$ that holds for all $r, m$. This argument is formalized in Lemma 15.

Knowing that $a_r^m \geq -A_0$, we can use Lemma 6 to get an upper bound of $L_r$ that approaches 0, but with a dependence on $\exp(A_0)$. However, we can use this upper bound to show that there exists a transition time $\tau$ for which $a_r^m \geq 0$ for every $m$ and every $r \geq \tau$. This is proven in Lemma 16.

Finally, we can apply Lemma 6 in two phases. For $r \leq \tau$, Lemma 6 implies that $L_r$ decreases with a dependence on $\exp(A_0)$, and for $r \geq \tau$, Lemma 6 implies that $L_r$ decreases at the desired rate. Theorem 2 follows by bounding the client losses in terms of $L_r$ (see Lemma 12).

## 6 EXPERIMENTS

We experimentally study the behavior of Local GD under different choices of learning rate and local steps. We use two datasets: (1) a synthetic dataset with $M = 2$ clients and $n = 1$ data point per client, (2) a heterogeneous dataset of MNIST images with binary labels. We evaluate three stepsize choices for Local GD: (1) small stepsize $\eta = 1/(KH)$ as required by baseline guarantees, (2) two-stage step size $\eta_1 = 1/(KH)$ and $\eta_2 = 1/H$, as in our Theorem 1, and (3) large stepsize $\eta = 1/H$. Note that Minibatch SGD in the deterministic setting reduces to Local GD with a single local step, which is included in the evaluation.

Additionally, to verify that Local SGD outperforms Minibatch SGD in practice (Woodworth et al., 2020b; 2021; Glasgow et al., 2022; Patel et al., 2024), we compare these two algorithms for training ResNets (He et al., 2016) on CIFAR-10. Results are included in Appendix F.

## 6.1 Setup

**Datasets.** For the synthetic dataset, the data $x_1, x_2$ have significantly different magnitudes, the angle between them is close to 180 degrees, but they have the same label. Using the notation of Section 5, this means $\gamma_{\max}/\gamma_{\min}$ is large and $c$ is close to $-1$ (full details in Appendix E).

Following recent work on GD for logistic regression (Wu et al., 2024b;a), we also evaluate on a dataset of 1000 MNIST images. We sample these images uniformly at random from the MNIST training set, then partition them into $M = 5$ client datasets with $n = 200$ images each. To create heterogeneity, we partition the data using a common protocol in which a large proportion of each client's data comes from a small number of classes (Karimireddy et al., 2020). After partitioning data based on digit labels, we binarize the problem by reducing each image's label mod 2. See Appendix E for a complete description. For both datasets, we scale every sample so that the maximum data norm is 1. This means $\|\nabla^2 F(w)\| \leq 1/4$ (see Appendix C.5.1), so we use $H = 1/4$ to set stepsizes.

**Stepsizes.** We set $\eta$ according to the requirements of theoretical guarantees, i.e. $\eta = 1/(KH)$ from Corollary 2, and $\eta_1 = 1/(KH), \eta_2 = 1/H$ from Theorem 1. For simplicity, we ignore constants and logarithmic terms. We also evaluate Local GD with a large stepsize, i.e. $\eta = 1/H$. For the two-stage stepsize, we choose $r_0$ (the number of rounds in the first stage) as a linear function of $K$, as required by Theorem 1. Accordingly, we set $r_0 = \lambda K$ and tune $\lambda$ to ensure that the loss remains stable when transitioning to the second stage. See Appendix E for the search space and tuned values.

## 6.2 Results

**Benefit of Local Steps.** Figure 1 shows that the small stepsize $\eta \leq 1/(KH)$ required by baseline guarantees is overly conservative. All choices of $K$ in this regime lead to overlapping loss curves, since the choice of more local steps $K$ is essentially cancelled out by a smaller step size $\eta$. On the other hand, the two-stage stepsize yields faster convergence with larger $K$, and mostly maintains stability throughout optimization. This underscores our discussion in Section 7: operating under worst-case assumptions can lead to suboptimal performance on particular problems.

**Instability with Large K.** Local GD with a large stepsize $\eta = 1/H$ exhibits a significant increase in loss during early rounds when training on the synthetic dataset with large $K$. Still, the large stepsize exhibits the fastest convergence in the long term for both datasets. This behavior is reminiscent of GD for logistic regression, where a large stepsize was shown to create instability early in training while eventually leading to faster convergence (Wu et al., 2024a). Therefore, explaining the superior practical performance of Local GD may require a theoretical framework that allows for unstable convergence. One possible explanation for the superiority of the large stepsize is that the two-stage stepsize prioritizes stability over speed: the second stage does not start until the loss is low enough that a large stepsize will not create instability. On the other hand, Local GD with a large stepsize remains stable with MNIST data even with very large $K$. This highlights another factor affecting performance of Local GD: the structure in the training data, rather than the prediction problem alone.

## 7 Discussion

**Heterogeneity Assumptions for Worst-Case Analysis** The pessimism of existing worst-case guarantees has motivated the search for "better" heterogeneity assumptions (Woodworth et al., 2020b; Wang et al., 2022; Patel et al., 2023; 2024), that is, assumptions that accurately capture practically relevant problems. However, the heterogeneity assumptions yet explored do not lead to guarantees for Local SGD that align with empirical observations on practical problems (Wang et al., 2022; Patel et al., 2023). Indeed, the de facto standard heterogeneity assumption — that there exists some $\kappa > 0$ such that $\|\nabla F_m(w) - \nabla F(w)\| \leq \kappa$ for all $w$ — yields guarantees which imply that Local SGD is significantly outperformed by Minibatch SGD (Woodworth et al., 2020b) for problems with moderate heterogeneity. Yet, Local SGD remains the standard distributed optimization algorithm and usually outperforms Minibatch SGD in practice. An alternative to searching for heterogeneity

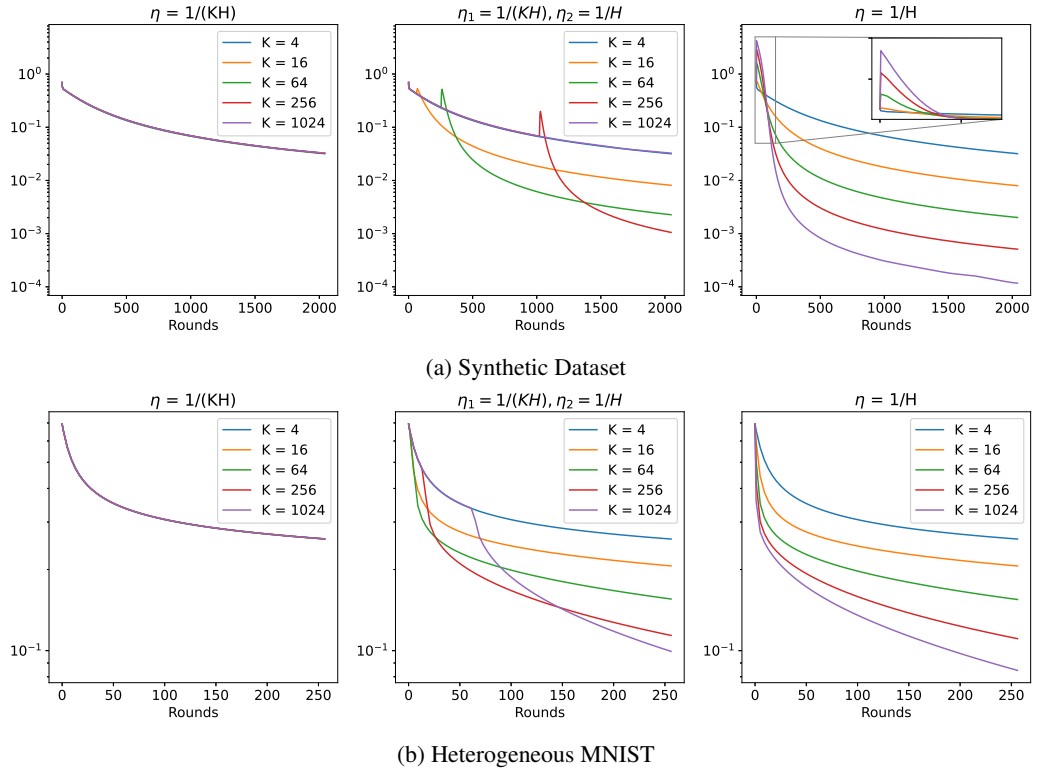

(a) Synthetic Dataset

(b) Heterogeneous MNIST

Figure 1: Train loss of Local GD for a synthetic dataset and MNIST. Left: Small stepsize $\eta = 1/(KH)$, as required by baselines (Corollary 2). Middle: Two stage stepsize with $\eta_1 = 1/(KH)$ and $\eta_2 = 1/H$, as in our Theorem 1. Right: Large stepsize $\eta = 1/H$. For the synthetic dataset, a large stepsize causes the loss to increase significantly during early rounds.

assumptions is to analyze practical problems directly; this perspective was also discussed by Patel et al. (2024). Indeed, in Section 4, we showed that local steps are provably beneficial due to the structure of the loss landscape — that the Hessian vanishes with the objective — instead of the similarity of client objectives. We are optimistic that studying particular problems may provide insights into general structure that could explain algorithmic behavior for other practical problems.

**Non-Monotonic Loss** Our experimental results suggest that Local GD with constant $\eta$ and large $K$ may converge for the distributed logistic regression problem, potentially with non-monotonic decrease of the loss function. The unstable convergence of GD for logistic regression was recently studied by Wu et al. (2024b;a), who showed that GD with any learning rate can converge, but with a non-monotonic loss decrease. In our experiments, non-monotonicity of the loss does not come from $\eta > 1/H$, but rather from large $\eta K$, which creates large updates to client models between averaging steps. With highly heterogeneous objectives, this can cause the global loss to increase from one round to the next. Based on these experiments, it is possible that proving the benefits of local steps for vanilla Local GD will require a theoretical framework that allows for unstable convergence.

**Limitations and Future Work** The most important limitation of the current work is that, while we analyze two variants of Local GD, our results do not apply for the vanilla Local GD algorithm. Although Two-Stage Local GD enjoys a strong guarantee due to the learning rate warmup, the question remains whether this warmup is necessary to achieve convergence. Indeed, experiments indicate that vanilla Local GD can converge faster with large $K$, even if this creates instability during the initial part of training. It remains open whether vanilla Local GD can converge at a rate of $\mathcal{O}(1/KR)$ for distributed logistic regression. Our analysis of Local GF is a step towards vanilla Local GD (in that the learning rate is fixed throughout optimization), but these results are preliminary in that they require strong assumptions on the number of clients and size of the datasets. We leave the problem of analyzing vanilla Local GD for future work.

ACKNOWLEDGMENTS

We would like to thank the anonymous reviewers for their helpful comments. This work is supported by the Institute for Digital Innovation fellowship, a ORIEI seed funding, an IDIA P3 fellowship from George Mason University, a Cisco Faculty Research Award, and NSF award #2436217, #2425687.

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

CONTENTS

## A  PROOFS FOR SECTION 4

**Lemma 7** (Restatement of Lemma 1). *For all $\boldsymbol{w}_1, \boldsymbol{w}_2 \in \mathbb{R}^d$ and all $m \in [M]$,*

$$\|\nabla^2 F_m(\boldsymbol{w}_2)\| \le F_m(\boldsymbol{w}_1)\left(1 + \|\boldsymbol{w}_2 - \boldsymbol{w}_1\|\left(1 + \exp(\|\boldsymbol{w}_2 - \boldsymbol{w}_1\|^2)\left(1 + \frac{1}{2}\|\boldsymbol{w}_2 - \boldsymbol{w}_1\|^2\right)\right)\right). \tag{26}$$

*Proof.* Let $\boldsymbol{v} = (\boldsymbol{w}_2 - \boldsymbol{w}_1)/\|\boldsymbol{w}_2 - \boldsymbol{w}_1\|, t > 0$, and $\boldsymbol{w} = \boldsymbol{w}_1 + t\boldsymbol{v}$. Then starting with Equation 568 from Lemma 25,

$$\|\nabla^2 F_m(\boldsymbol{w})\| \le F_m(\boldsymbol{w}) \tag{27}$$

$$= F_m(\boldsymbol{w}_1) + \langle \nabla F_m(\boldsymbol{w}_1), \boldsymbol{w} - \boldsymbol{w}_1 \rangle + \int_0^t (t - s)\boldsymbol{v}^\mathsf{T}\nabla^2 F_m(\boldsymbol{w}_1 + s\boldsymbol{v})\boldsymbol{v}\,ds \tag{28}$$

$$= F_m(\boldsymbol{w}_1) + t\langle \nabla F_m(\boldsymbol{w}_1), \boldsymbol{v} \rangle + \int_0^t (t - s)\boldsymbol{v}^\mathsf{T}\nabla^2 F_m(\boldsymbol{w}_1 + s\boldsymbol{v})\boldsymbol{v}\,ds \tag{29}$$

$$\le F_m(\boldsymbol{w}_1) + t\langle \nabla F_m(\boldsymbol{w}_1), \boldsymbol{v} \rangle + \int_0^t (t - s)\left\|\nabla^2 F_m(\boldsymbol{w}_1 + s\boldsymbol{v})\right\|ds \tag{30}$$

$$\le F_m(\boldsymbol{w}_1) + t\|\nabla F_m(\boldsymbol{w}_1)\| + t\int_0^t \left\|\nabla^2 F_m(\boldsymbol{w}_1 + s\boldsymbol{v})\right\|ds. \tag{31}$$

Denoting $a = F_m(\boldsymbol{w}_1)$, $b = \|\nabla F_m(\boldsymbol{w}_1)\|$, $\phi_1(t) = a + bt$, $\phi_2(t) = t$, and

$$f(t) = \int_0^t \|\nabla^2 F_m(\boldsymbol{w}_1 + s\boldsymbol{v})\|ds, \tag{32}$$

Equation 31 becomes

$$f'(t) \le \phi_1(t) + \phi_2(t)f(t). \tag{33}$$

We can then apply Lemma 27 to obtain

$$f'(t) \le \phi_1(t) + \phi_2(t)\exp\left(\int_0^t \phi_2(s)ds\right)\left(f(0) \right. \tag{34}$$

$$\left. + \int_0^t \exp\left(-\int_0^s \phi_2(r)dr\right)\phi_1(s)ds\right) \tag{35}$$

$$\|\nabla^2 F_m(\boldsymbol{w}_1 + t\boldsymbol{v})\| \le a + bt + t\exp\left(\frac{1}{2}t^2\right)\int_0^t \exp\left(\frac{1}{2}s^2\right)(a + bs)ds \tag{36}$$

$$\|\nabla^2 F_m(\boldsymbol{w}_1 + t\boldsymbol{v})\| \le a + bt + t\exp(t^2)\int_0^t (a + bs)ds \tag{37}$$

$$\|\nabla^2 F_m(\boldsymbol{w}_1 + t\boldsymbol{v})\| \le a + bt + t\exp(t^2)\left(at + \frac{1}{2}bt^2\right). \tag{38}$$

Therefore

$$\|\nabla^2 F_m(\boldsymbol{w}_1 + t\boldsymbol{v})\| \le F_m(\boldsymbol{w}_1) + t\|\nabla F_m(\boldsymbol{w}_1)\| + t\exp(t^2)\left(F_m(\boldsymbol{w}_1) + \frac{1}{2}t^2\|\nabla F_m(\boldsymbol{w}_1)\|\right), \tag{39}$$

and finally, choosing $t = \|\boldsymbol{w}_2 - \boldsymbol{w}_1\|$ implies

$$\|\nabla^2 F_m(\boldsymbol{w}_2)\| \le F_m(\boldsymbol{w}_1) + \|\nabla F_m(\boldsymbol{w}_1)\| \|\boldsymbol{w}_2 - \boldsymbol{w}_1\| \tag{40}$$

$$+ \|\boldsymbol{w}_2 - \boldsymbol{w}_1\| \exp(\|\boldsymbol{w}_2 - \boldsymbol{w}_1\|^2) \left( F_m(\boldsymbol{w}_1) + \frac{1}{2} \|\nabla F_m(\boldsymbol{w}_1)\| \|\boldsymbol{w}_2 - \boldsymbol{w}_1\|^2 \right) \tag{41}$$

$$\overset{(i)}{\le} F_m(\boldsymbol{w}_1) \left( 1 + \|\boldsymbol{w}_2 - \boldsymbol{w}_1\| \left( 1 + \exp(\|\boldsymbol{w}_2 - \boldsymbol{w}_1\|^2) \left( 1 + \frac{1}{2} \|\boldsymbol{w}_2 - \boldsymbol{w}_1\|^2 \right) \right) \right), \tag{42}$$

where $(i)$ uses Equation 567. $\square$

**Lemma 8** (Restatement of Lemma 2). *If $\eta \le 8$, then for every $r \ge 0$ and $k \le K$,*

$$\|\boldsymbol{w}_{r,k}^m - \bar{\boldsymbol{w}}_r\| \le \eta K F_m(\bar{\boldsymbol{w}}_r). \tag{43}$$

*Proof.* Let $r \ge 0$ and $m \in [M]$. Recall that $0 \le \ell''(z) \le 1/4$, so $\|\nabla^2 F_m(\boldsymbol{w})\| \le 1/4$. Therefore, for every $k \le K - 1$,

$$F_m(\boldsymbol{w}_{r,k+1}^m) \le F_m(\boldsymbol{w}_{r,k}^m) + \langle \nabla F_m(\boldsymbol{w}_{r,k}^m), \boldsymbol{w}_{r,k+1}^m - \boldsymbol{w}_{r,k}^m \rangle + \frac{1}{8} \left\| \boldsymbol{w}_{r,k+1}^m - \boldsymbol{w}_{r,k}^m \right\|^2 \tag{44}$$

$$\le F_m(\boldsymbol{w}_{r,k}^m) - \eta \left\| \nabla F_m(\boldsymbol{w}_{r,k}^m) \right\|^2 + \frac{\eta^2}{8} \left\| \nabla F_m(\boldsymbol{w}_{r,k}^m) \right\|^2 \tag{45}$$

$$\le F_m(\boldsymbol{w}_{r,k}^m) - \eta \left( 1 - \frac{\eta}{8} \right) \left\| \nabla F_m(\boldsymbol{w}_{r,k}^m) \right\|^2 \tag{46}$$

$$\overset{(i)}{\le} F_m(\boldsymbol{w}_{r,k}^m), \tag{47}$$

where $(i)$ uses the condition $\eta \le 8$. Induction over $k$ implies $F_m(\boldsymbol{w}_{r,k}^m) \le F_m(\bar{\boldsymbol{w}}_r)$. Therefore

$$\|\boldsymbol{w}_{r,k}^m - \bar{\boldsymbol{w}}_r\| = \left\| \frac{\eta}{M} \sum_{m=1}^{M} \sum_{j=0}^{k-1} \nabla F_m(\boldsymbol{w}_{r,j}^m) \right\| \tag{48}$$

$$\le \frac{\eta}{M} \sum_{m=1}^{M} \sum_{j=0}^{k-1} \left\| \nabla F_m(\boldsymbol{w}_{r,j}^m) \right\| \tag{49}$$

$$\overset{(i)}{\le} \frac{\eta}{M} \sum_{m=1}^{M} \sum_{j=0}^{k-1} F_m(\boldsymbol{w}_{r,j}^m) \tag{50}$$

$$\overset{(ii)}{\le} \frac{\eta}{M} \sum_{m=1}^{M} \sum_{j=0}^{k-1} F_m(\bar{\boldsymbol{w}}_r) \tag{51}$$

$$\le \eta k F_m(\bar{\boldsymbol{w}}_r) \tag{52}$$

$$\le \eta K F_m(\bar{\boldsymbol{w}}_r), \tag{53}$$

where $(i)$ uses Equation 567 and $(ii)$ uses Equation 47. $\square$

**Lemma 9** (Restatement of Lemma 3). *If $\eta \le 8$, and $F(\bar{\boldsymbol{w}}_r) \le 1/(\eta K M)$, then for every $m \in [M]$ and $k \in [K]$,*

$$\|\nabla F_m(\boldsymbol{w}_{r,k}^m) - \nabla F_m(\bar{\boldsymbol{w}}_r)\| \le 7 \eta K F_m(\bar{\boldsymbol{w}}_r)^2. \tag{54}$$

*Proof.* Let $t \in [0, 1]$. Then

$$\|(t \boldsymbol{w}_{r,k}^m + (1-t) \bar{\boldsymbol{w}}_r) - \bar{\boldsymbol{w}}_r\| \le \|\boldsymbol{w}_{r,k}^m - \bar{\boldsymbol{w}}_r\| \overset{(i)}{\le} \eta K F_m(\bar{\boldsymbol{w}}_r) \overset{(ii)}{\le} 1, \tag{55}$$

where $(i)$ uses Lemma 2 and $(ii)$ uses the condition $F(\bar{\boldsymbol{w}}_r) \le 1/(\eta K M)$. So we can use Lemma 1 to bound

$$\|\nabla^2 F_m(t \boldsymbol{w}_{r,k}^m + (1-t) \bar{\boldsymbol{w}}_r)\| \le F_m(\boldsymbol{w}_r) \left( 1 + \left( 1 + \exp(1) \left( 1 + \frac{1}{2} \right) \right) \right) \le 7 F_m(\boldsymbol{w}_r). \tag{56}$$

Finally, let $\boldsymbol{v} = \frac{\boldsymbol{w}_{r,k}^m - \bar{\boldsymbol{w}}_r}{\|\boldsymbol{w}_{r,k}^m - \bar{\boldsymbol{w}}_r\|}$ and $\lambda = \|\boldsymbol{w}_{r,k}^m - \bar{\boldsymbol{w}}_r\|$. Then

$$\|\nabla F_m(\boldsymbol{w}_{r,k}^m) - \nabla F_m(\bar{\boldsymbol{w}}_r)\| = \left\| \int_0^\lambda \nabla^2 F(s\boldsymbol{w}_{r,k}^m + (1-s)\bar{\boldsymbol{w}}_r)\boldsymbol{v}\,ds \right\| \tag{57}$$

$$= \int_0^\lambda \left\| \nabla^2 F(s\boldsymbol{w}_{r,k}^m + (1-s)\bar{\boldsymbol{w}}_r) \right\| ds \tag{58}$$

$$\overset{(i)}{\leq} 7\lambda F_m(\bar{\boldsymbol{w}}_r) \tag{59}$$

$$= 7\|\boldsymbol{w}_{r,k}^m - \bar{\boldsymbol{w}}_r\| F_m(\bar{\boldsymbol{w}}_r) \tag{60}$$

$$\overset{(ii)}{\leq} 7\eta K F_m(\bar{\boldsymbol{w}}_r)^2, \tag{61}$$

where $(i)$ uses Equation 56 and $(ii)$ uses Lemma 2. $\qquad\square$

**Lemma 10** (Restatement of Theorem 4). *Suppose that $\eta \leq 4$ and let $r_0 \geq 0$ such that $F(\bar{\boldsymbol{w}}_{r_0}) \leq \gamma^2/(42\eta KM)$. Then for every $r \geq 2r_0$,*

$$F(\bar{\boldsymbol{w}}_r) \leq \frac{4}{\eta\gamma^2 Kr}. \tag{62}$$

*Proof.* We will show by induction that

$$F(\bar{\boldsymbol{w}}_r) \leq \frac{1}{1/F(\bar{\boldsymbol{w}}_{r_0}) + \eta\gamma^2 K(r - r_0)/2} \tag{63}$$

for all $r \geq r_0$. Clearly it holds for $r_0$, so suppose that it holds for some $r \geq r_0$. Then $F(\bar{\boldsymbol{w}}_r) \leq F(\bar{\boldsymbol{w}}_{r_0}) \leq 1/(\eta KM)$. From the definition of Local GD,

$$\bar{\boldsymbol{w}}_{r+1} - \bar{\boldsymbol{w}}_r = -\frac{\eta}{M} \sum_{m=1}^M \sum_{k=0}^{K-1} \nabla F_m(\boldsymbol{w}_{r,k}^m) \tag{64}$$

$$= -\frac{\eta}{M} \sum_{m=1}^M \sum_{k=0}^{K-1} \nabla F_m(\bar{\boldsymbol{w}}_r) - \frac{\eta}{M} \sum_{m=1}^M \sum_{k=0}^{K-1} (\nabla F_m(\boldsymbol{w}_{r,k}^m) - \nabla F_m(\bar{\boldsymbol{w}}_r)) \tag{65}$$

$$= -\eta K \nabla F(\bar{\boldsymbol{w}}_r) - \frac{\eta}{M} \sum_{m=1}^M \sum_{k=0}^{K-1} (\nabla F_m(\boldsymbol{w}_{r,k}^m) - \nabla F_m(\bar{\boldsymbol{w}}_r)). \tag{66}$$

Denoting $b_r = \frac{1}{KM} \sum_{m=1}^M \sum_{k=0}^{K-1} (\nabla F_m(\boldsymbol{w}_{r,k}^m) - \nabla F_m(\bar{\boldsymbol{w}}_r))$, this means

$$\bar{\boldsymbol{w}}_{r+1} - \bar{\boldsymbol{w}}_r = \eta K (\nabla F(\bar{\boldsymbol{w}}_r) + b_r). \tag{67}$$

Notice that

$$\|b_r\| \leq \frac{1}{KM} \sum_{m=1}^M \sum_{k=0}^{K-1} \left\| \nabla F_m(\boldsymbol{w}_{r,k}^m) - \nabla F_m(\bar{\boldsymbol{w}}_r) \right\| \tag{68}$$

$$\overset{(i)}{\leq} \frac{7\eta K}{M} \sum_{m=1}^M F_m(\bar{\boldsymbol{w}}_r)^2 \tag{69}$$

$$\leq \frac{7\eta K}{M} \left( \sum_{m=1}^M F_m(\bar{\boldsymbol{w}}_r) \right)^2 \tag{70}$$

$$= 7\eta K M F(\bar{\boldsymbol{w}}_r)^2, \tag{71}$$

where $(i)$ uses Lemma 3. Also, by Lemma 2,

$$\|\bar{\boldsymbol{w}}_{r+1} - \bar{\boldsymbol{w}}_r\| \leq \eta K F(\bar{\boldsymbol{w}}_r) \overset{(i)}{\leq} 1, \tag{72}$$

where $(i)$ uses the condition $F(\bar{\boldsymbol{w}}_r) \le 1/(\eta KM)$. By Lemma 1, this means for all $t \in [0,1]$:

$$\left\| \nabla^2 F((1-t)\bar{\boldsymbol{w}}_r + t\bar{\boldsymbol{w}}_{r+1}) \right\| \le F(\bar{\boldsymbol{w}}_r) \left( 1 + \left( 1 + \exp(1) \left( 1 + \frac{1}{2} \right) \right) \right) \le 7F(\bar{\boldsymbol{w}}_r). \tag{73}$$

We can then use Equation 67, Equation 71, and Equation 73 to upper bound $F(\bar{\boldsymbol{w}}_{r+1})$. Letting $\lambda = \|\bar{\boldsymbol{w}}_{r+1} - \bar{\boldsymbol{w}}_r\|$ and $v = \frac{\bar{\boldsymbol{w}}_{r+1} - \bar{\boldsymbol{w}}_r}{\|\bar{\boldsymbol{w}}_{r+1} - \bar{\boldsymbol{w}}_r\|}$,

$$F(\bar{\boldsymbol{w}}_{r+1}) \tag{74}$$

$$= F(\bar{\boldsymbol{w}}_r) + \langle \nabla F(\bar{\boldsymbol{w}}_r), \bar{\boldsymbol{w}}_{r+1} - \bar{\boldsymbol{w}}_r \rangle + \int_0^\lambda (\lambda - t) \boldsymbol{v}^\intercal \nabla^2 F(\bar{\boldsymbol{w}}_r + t\boldsymbol{v}) \boldsymbol{v} \, dt \tag{75}$$

$$\le F(\bar{\boldsymbol{w}}_r) + \langle \nabla F(\bar{\boldsymbol{w}}_r), \bar{\boldsymbol{w}}_{r+1} - \bar{\boldsymbol{w}}_r \rangle + \int_0^\lambda (\lambda - t) \left\| \nabla^2 F(\bar{\boldsymbol{w}}_r + t\boldsymbol{v}) \right\| \, dt \tag{76}$$

$$\overset{(i)}{\le} F(\bar{\boldsymbol{w}}_r) + \langle \nabla F(\bar{\boldsymbol{w}}_r), \bar{\boldsymbol{w}}_{r+1} - \bar{\boldsymbol{w}}_r \rangle + \frac{7}{2} \lambda^2 F(\bar{\boldsymbol{w}}_r) \tag{77}$$

$$\overset{(ii)}{=} F(\bar{\boldsymbol{w}}_r) - \eta K \|\nabla F(\bar{\boldsymbol{w}}_r)\|^2 + \eta K \langle \nabla F(\bar{\boldsymbol{w}}_r), b_r \rangle + \frac{7}{2} \eta^2 K^2 \|\nabla F(\bar{\boldsymbol{w}}_r) + b_r\|^2 F(\bar{\boldsymbol{w}}_r) \tag{78}$$

$$\le F(\bar{\boldsymbol{w}}_r) - \eta K \|\nabla F(\bar{\boldsymbol{w}}_r)\|^2 + \eta K \|\nabla F(\bar{\boldsymbol{w}}_r)\| \|b_r\| + 7\eta^2 K^2 \left( \|\nabla F(\bar{\boldsymbol{w}}_r)\|^2 + \|b_r\|^2 \right) F(\bar{\boldsymbol{w}}_r) \tag{79}$$

$$\overset{(iii)}{\le} F(\bar{\boldsymbol{w}}_r) - \eta K \|\nabla F(\bar{\boldsymbol{w}}_r)\|^2 + 7\eta^2 K^2 M \|\nabla F(\bar{\boldsymbol{w}}_r)\| F(\bar{\boldsymbol{w}}_r)^2 \tag{80}$$

$$+ 7\eta^2 K^2 \left( \|\nabla F(\bar{\boldsymbol{w}}_r)\|^2 + 49\eta^2 K^2 M^2 F(\bar{\boldsymbol{w}}_r)^4 \right) F(\bar{\boldsymbol{w}}_r) \tag{81}$$

$$\overset{(iv)}{\le} F(\bar{\boldsymbol{w}}_r) - \eta K \|\nabla F(\bar{\boldsymbol{w}}_r)\|^2 + \left( 7\eta^2 K^2 M F(\bar{\boldsymbol{w}}_r) + 7\eta^2 K^2 F(\bar{\boldsymbol{w}}_r) + 343\eta^4 K^4 M^2 F(\bar{\boldsymbol{w}}_r)^3 \right) F(\bar{\boldsymbol{w}}_r)^2 \tag{82}$$

$$\overset{(v)}{\le} F(\bar{\boldsymbol{w}}_r) - \left( \eta\gamma^2 K - 7\eta^2 K^2 M F(\bar{\boldsymbol{w}}_r) - 7\eta^2 K^2 F(\bar{\boldsymbol{w}}_r) - 343\eta^4 K^4 M^2 F(\bar{\boldsymbol{w}}_r)^3 \right) F(\bar{\boldsymbol{w}}_r)^2 \tag{83}$$

$$= F(\bar{\boldsymbol{w}}_r) - \eta\gamma^2 K \left( 1 - \frac{7\eta K M}{\gamma^2} F(\bar{\boldsymbol{w}}_r) - \frac{7\eta K}{\gamma^2} F(\bar{\boldsymbol{w}}_r) - \frac{343\eta^3 K^3 M^2}{\gamma^2} F(\bar{\boldsymbol{w}}_r)^3 \right) F(\bar{\boldsymbol{w}}_r)^2 \tag{84}$$

$$\overset{(vi)}{\le} F(\bar{\boldsymbol{w}}_r) - \frac{1}{2} \eta\gamma^2 K F(\bar{\boldsymbol{w}}_r)^2, \tag{85}$$

where $(i)$ uses Equation 73, $(ii)$ uses Equation 67, $(iii)$ uses Equation 71, $(iv)$ uses Equation 569, $(v)$ uses Lemma 26, and $(vi)$ uses $F(\bar{\boldsymbol{w}}_r) \le F(\bar{\boldsymbol{w}}_{r_0})$ from the inductive hypothesis together with $F(\bar{\boldsymbol{w}}_{r_0}) \le \frac{\gamma^2}{42\eta KM}$.

Therefore

$$\frac{1}{F(\bar{\boldsymbol{w}}_r)} \le \frac{1}{F(\bar{\boldsymbol{w}}_{r+1})} - \frac{1}{2} \eta\gamma^2 K \frac{F(\bar{\boldsymbol{w}}_r)}{F(\bar{\boldsymbol{w}}_{r+1})} \tag{86}$$

$$\frac{1}{F(\bar{\boldsymbol{w}}_{r+1})} \ge \frac{1}{F(\bar{\boldsymbol{w}}_r)} + \frac{1}{2} \eta\gamma^2 K \frac{F(\bar{\boldsymbol{w}}_r)}{F(\bar{\boldsymbol{w}}_{r+1})} \tag{87}$$

$$\frac{1}{F(\bar{\boldsymbol{w}}_{r+1})} \ge \frac{1}{F(\bar{\boldsymbol{w}}_r)} + \frac{1}{2} \eta\gamma^2 K \tag{88}$$

$$\frac{1}{F(\bar{\boldsymbol{w}}_{r+1})} \overset{(i)}{\ge} \frac{1}{F(\bar{\boldsymbol{w}}_{r_0})} + \frac{1}{2} \eta\gamma^2 K(r - r_0) + \frac{1}{2} \eta\gamma^2 K \tag{89}$$

$$\frac{1}{F(\bar{\boldsymbol{w}}_{r+1})} \ge \frac{1}{F(\bar{\boldsymbol{w}}_{r_0})} + \frac{1}{2} \eta\gamma^2 K(r + 1 - r_0) \tag{90}$$

$$F(\bar{\boldsymbol{w}}_{r+1}) \le \frac{1}{\frac{1}{F(\bar{\boldsymbol{w}}_{r_0})} + \frac{1}{2} \eta\gamma^2 K(r + 1 - r_0)}, \tag{91}$$

where $(i)$ uses the inductive hypothesis. This completes the induction and proves Equation 63. Therefore, for every $r \geq r_0$,

$$F(\bar{\boldsymbol{w}}_r) \leq \frac{1}{\frac{1}{F(\bar{\boldsymbol{w}}_{r_0})} + \frac{1}{2}\eta\gamma^2 K(r-r_0)} \tag{92}$$

$$\leq \frac{2}{\eta\gamma^2 K(r-r_0)} \tag{93}$$

$$\leq \frac{4}{\eta\gamma^2 K r}. \tag{94}$$

$\square$

**Theorem 3** (Restatement of Theorem 1). *Let $\eta_2 > 0$ and denote $\tilde{\eta} = \eta_2 K M$. Suppose*

$$r_0 \geq \max\left\{2, \frac{126\tilde{\eta}}{\gamma^4}, \frac{252\tilde{\eta}}{\gamma^4}\log^2\left(\frac{504\tilde{\eta}}{\gamma^4}\right), \frac{76\tilde{\eta}^{3/4}}{\gamma^{5/2}}\log\left(\frac{38\tilde{\eta}^{3/4}}{\gamma^{5/2}}\right)\right\}, \tag{95}$$

*and $R \geq r_0$. Then with*

$$\eta_1 = \tilde{\mathcal{O}}\left(\min\left\{\frac{1}{K}, \frac{\eta_2^{1/3}M^{1/3}}{\gamma^2 K^{2/3}}\right\}\right), \tag{96}$$

*Two-Stage Local GD (Algorithm 2) satisfies for all $r \geq r_0$:*

$$F(\bar{\boldsymbol{w}}_r) \leq \frac{2}{\eta_2\gamma^2 K(r-r_0)}. \tag{97}$$

*Proof.* We would like to apply Lemma 4 to the second phase of Two-Stage Local GD, but in order to do so we must show that

$$F(\hat{\boldsymbol{w}}_1) \leq \frac{\gamma^2}{42\eta_2 K M}. \tag{98}$$

We already know from Corollary 2 that

$$F(\hat{\boldsymbol{w}}_1) \leq \frac{1}{\gamma^2 r_0} + \frac{\log^2(r_0)}{\gamma^2 r_0} + \frac{\log^{4/3}(r_0)}{\gamma^{4/3} r_0^{4/3}}. \tag{99}$$

From our choice of $r_0$,

$$\frac{1}{\gamma^2 r_0} \leq \frac{1}{\gamma^2}\frac{\gamma^4}{126\eta_2 K M} = \frac{\gamma^2}{126\eta K M}. \tag{100}$$

Also, applying Lemma 28,

$$r_0 \geq \max\left\{2, \frac{252\tilde{\eta}}{\gamma^4}\log^2\left(\frac{504\tilde{\eta}}{\gamma^4}\right)\right\} \implies \frac{r_0}{\log^2(r_0)} \geq \frac{126\eta K M}{\gamma^4}, \tag{101}$$

so

$$\frac{\log^2(r_0)}{\gamma^2 r_0} \leq \frac{\gamma^2}{126\eta K M}. \tag{102}$$

Similarly, applying Lemma 28 again,

$$r_0 \geq \max\left\{2, \frac{76\tilde{\eta}^{3/4}}{\gamma^{5/2}}\log\left(\frac{38\tilde{\eta}^{3/4}}{\gamma^{5/2}}\right)\right\} \implies \frac{r_0}{\log(r_0)} \geq \frac{38\tilde{\eta}^{3/4}}{\gamma^{5/2}} \tag{103}$$

so

$$\frac{\log^{4/3}(r_0)}{\gamma^{4/3} r_0^{4/3}} \leq \frac{1}{\gamma^{4/3}}\left(\frac{\log(r_0)}{r_0}\right)^{4/3} \leq \frac{\gamma^2}{126\tilde{\eta}}. \tag{104}$$

Plugging Equation 100, Equation 102, and Equation 104 into Equation 99 yields

$$F(\hat{\boldsymbol{w}}_1) \leq \frac{\gamma^2}{42\eta K M}, \tag{105}$$

which is exactly Equation 98. Therefore, the condition of Lemma 4 is satisfied by $\hat{\boldsymbol{w}}_1$. Theorem 4 implies that, for all $r \geq r_0$,

$$F(\bar{\boldsymbol{w}}_r) \leq \frac{4}{\eta_2\gamma^2 K(r-r_0)}. \tag{106}$$

$\square$

# B PROOFS FOR SECTION 5

**Lemma 11.** *Denote $\boldsymbol{a}_r = \boldsymbol{W}\bar{\boldsymbol{w}}_r$, and*

$$\Phi(b, x) = \frac{W(\exp(b + \exp(x) + x))}{\exp(x)}, \tag{107}$$

*where $W$ denotes the Lambert W function. Then*

$$\boldsymbol{a}_{r+1} = \boldsymbol{a}_r + \frac{1}{M}\boldsymbol{G}\left(\frac{1}{\boldsymbol{\gamma}} \odot \log\left(\Phi(\eta K \boldsymbol{\gamma}^2, \boldsymbol{\gamma} \odot \boldsymbol{a}_r)\right)\right). \tag{108}$$

*Proof.* We can rewrite the gradient flow dynamics as

$$\dot{\boldsymbol{w}}_r^m(t) = \frac{\eta\gamma_m}{\exp(\gamma_m a_r^m(t)) + 1}\boldsymbol{w}_m^*, \tag{109}$$

so

$$\dot{a}_r^m(t) = \langle \dot{\boldsymbol{w}}_r^m(t), \boldsymbol{w}_m^* \rangle \tag{110}$$

$$= \left\langle \frac{\eta\gamma_m}{\exp(\gamma_m a_r^m(t)) + 1}\boldsymbol{w}_m^*, \boldsymbol{w}_m^* \right\rangle \tag{111}$$

$$= \frac{\eta\gamma_m}{\exp(\gamma_m a_r^m(t)) + 1}, \tag{112}$$

and

$$\dot{\boldsymbol{w}}_r^m(t) = \dot{a}_r^m(t)\boldsymbol{w}_m^*. \tag{113}$$

In other words, $\boldsymbol{w}_r^m(t)$ only changes in the direction of $\boldsymbol{w}_m^*$. Therefore the total update to a local model during a single round is:

$$\boldsymbol{w}_r^m(K) = \bar{\boldsymbol{w}}_r + (\boldsymbol{w}_r^m(K) - \boldsymbol{w}_r^m(0)) \tag{114}$$

$$= \bar{\boldsymbol{w}}_r + \int_0^K \dot{\boldsymbol{w}}_r^m(s)ds \tag{115}$$

$$= \bar{\boldsymbol{w}}_r + \left(\int_0^K \dot{a}_r^m(s)ds\right)\boldsymbol{w}_m^* \tag{116}$$

$$= \bar{\boldsymbol{w}}_r + (a_r^m(K) - a_r^m(0))\boldsymbol{w}_m^*. \tag{117}$$

Notice that Equation 112 is a separable ODE in the unknown $a_r^m(t)$, so we can solve by separation to obtain

$$\exp(\gamma_m a_r^m(t)) + \gamma_m a_r^m(t) = \eta\gamma_m^2 t + C. \tag{118}$$

Using the initial condition $a_r^m(0) = a_r^m$, we get $C = \exp(\gamma_m a_r^m) + \gamma_m a_r^m$, so

$$\exp(\gamma_m a_r^m(t)) + \gamma_m a_r^m(t) = \eta\gamma_m^2 t + \exp(\gamma_m a_r^m) + \gamma_m a_r^m. \tag{119}$$

This is a transcendental equation in $a_r^m(t)$ without a closed form solution: however the solution can be expressed in terms of the Lambert $W$ function. Let $z = \exp(\gamma_m a_r^m(t))$ and $x = \eta\gamma_m^2 t + \exp(\gamma_m a_r^m) + \gamma_m a_r^m$. Then

$$z + \log z = x \tag{120}$$
$$z\exp(z) = \exp(x) \tag{121}$$
$$z = W(\exp(x)), \tag{122}$$

so

$$\exp(\gamma_m a_r^m(t)) = W(\exp(\eta\gamma_m^2 + \exp(\gamma_m a_r^m) + \gamma_m a_r^m)) \tag{123}$$

$$a_r^m(t) = \frac{1}{\gamma_m}\log\left(W(\exp(\eta\gamma_m^2 + \exp(\gamma_m a_r^m) + \gamma_m a_r^m))\right). \tag{124}$$

To plug this into Equation 117, we first simplify

$$a_r^m(K) - a_r^m(0) = \frac{1}{\gamma_m} \log \left( W(\exp(\eta K \gamma_m^2 + \exp(\gamma_m a_r^m) + \gamma_m a_r^m))) \right) - a_r^m \tag{125}$$

$$= \frac{1}{\gamma_m} \log \left( W(\exp(\eta K \gamma_m^2 + \exp(\gamma_m a_r^m) + \gamma_m a_r^m))) \right) - \frac{1}{\gamma_m} \log \left( \exp(\gamma_m a_r^m) \right) \tag{126}$$

$$= \frac{1}{\gamma_m} \log \left( \frac{W(\exp(\eta K \gamma_m^2 + \exp(\gamma_m a_r^m) + \gamma_m a_r^m))}{\exp(\gamma_m a_r^m)} \right) \tag{127}$$

$$= \frac{1}{\gamma_m} \log \left( \Phi(\eta K \gamma_m^2, \gamma_m a_r^m) \right), \tag{128}$$

and plugging this into Equation 117 yields

$$\boldsymbol{w}_r^m(K) = \bar{\boldsymbol{w}}_r + \frac{1}{\gamma_m} \log \left( \Phi(\eta K \gamma_m^2, \gamma_m a_r^m) \, \boldsymbol{w}_m^* \right). \tag{129}$$

Then we can rewrite the global update as

$$\bar{\boldsymbol{w}}_{r+1} = \frac{1}{M} \sum_{m=1}^{M} \boldsymbol{w}_r^m(K) = \bar{\boldsymbol{w}}_r + \frac{1}{M} \sum_{m=1}^{M} \frac{1}{\gamma_m} \log \left( \Phi(\eta K \gamma_m^2, \gamma_m a_r^m) \, \boldsymbol{w}_m^* \right). \tag{130}$$

Applying $\langle \cdot, \boldsymbol{w}_m^* \rangle$ to each side:

$$a_{r+1}^m = a_r^m + \frac{1}{M} \sum_{n=1}^{M} \frac{1}{\gamma_n} \log \left( \Phi(\eta K \gamma_n^2, \gamma_n a_r^n) \right) \langle \boldsymbol{w}_n^*, \boldsymbol{w}_m^* \rangle. \tag{131}$$

This relation can be written in vector notation as

$$\boldsymbol{a}_{r+1} = \boldsymbol{a}_r + \frac{1}{M} \boldsymbol{G} \left( \frac{1}{\boldsymbol{\gamma}} \odot \log \left( \Phi(\eta K \boldsymbol{\gamma}^2, \boldsymbol{\gamma} \odot \boldsymbol{a}_r) \right) \right). \tag{132}$$

$\square$

Based on the above lemma, we can write a recurrence relation for the coordinates of $\boldsymbol{a}_r$ for the case of $M = 2$ as:

$$a_{r+1}^1 = a_r^1 + \frac{1}{M \gamma_1} \log \left( \Phi(\eta K \gamma_1^2, \gamma_1 a_r^1) \right) + \frac{c}{M \gamma_2} \log \left( \Phi(\eta K \gamma_2^2, \gamma_2 a_r^2) \right) \tag{133}$$

$$a_{r+1}^2 = a_r^2 + \frac{c}{M \gamma_1} \log \left( \Phi(\eta K \gamma_1^2, \gamma_1 a_r^1) \right) + \frac{1}{M \gamma_2} \log \left( \Phi(\eta K \gamma_2^2, \gamma_2 a_r^2) \right). \tag{134}$$

**Lemma 12.** *For each $m \in [M]$, if*

$$L_r \leq \min \left\{ \frac{1}{2\gamma_m}, \frac{1}{\gamma_m} \log \left( 1 + \frac{\eta K \gamma_m^2}{2 + \eta K \gamma_m^2} \right) \right\}, \tag{135}$$

*then*

$$F_m(\bar{\boldsymbol{w}}_r) \leq \frac{4 L_r}{\eta K \gamma_m}. \tag{136}$$

*Proof.* Recall that

$$F_m(\bar{\boldsymbol{w}}_r) = \log(1 + \exp(-\langle \bar{\boldsymbol{w}}_r, \boldsymbol{x}_m \rangle)) = \log(1 + \exp(-\gamma_m a_r^m)). \tag{137}$$

We can also bound $\exp(-\gamma_m a_r^m)$ in terms of $L_r$ as follows:

$$L_r = \max_{m \in [M]} \frac{1}{\gamma_m} \log \left( \Phi(\eta K \gamma_m^2, \gamma_m a_r^m) \right) \tag{138}$$

$$\geq \frac{1}{\gamma_m} \log \left( \Phi(\eta K \gamma_m^2, \gamma_m a_r^m) \right) \tag{139}$$

$$\overset{(i)}{\geq} \frac{1}{2\gamma_m} \log \left( 1 + \frac{\eta K \gamma_m^2}{\exp(\gamma_m a_r^m)} \right), \tag{140}$$

where $(i)$ uses Lemma 32. Note that the condition of Lemma 32 is satisfied in this circumstance, since the condition $L_r \leq \frac{1}{\gamma_m} \log\left(1 + \frac{\eta K \gamma_m^2}{2 + \eta K \gamma_m^2}\right)$ implies that $\Phi(\eta K \gamma_m^2, \gamma_m a_r^m) \geq 1 + \frac{\eta K \gamma_m^2}{2 + \eta K \gamma_m^2}$; the condition of Lemma 32 then follows from Lemma 31.

Rearranging Equation 140,

$$\exp(-\gamma_m a_r^m) \leq \frac{\exp(2\gamma_m L_r) - 1}{\eta K \gamma_m^2} \overset{(i)}{\leq} \frac{4\gamma_m L_r}{\eta K \gamma_m^2} = \frac{4 L_r}{\eta K \gamma_m}, \tag{141}$$

where $(i)$ uses convexity of $\exp(x) - 1$ together with the condition $L_r \leq 1/(2\gamma_m)$ to obtain $\exp(x) - 1 \leq (1 - x)f(0) + x f(1) = (e - 1)x \leq 2x$.

Finally, we combine Equation 137 and Equation 141:

$$F_m(\bar{\boldsymbol{w}}_r) = \log(1 + \exp(-\gamma_m a_r^m)) \leq \exp(-\gamma_m a_r^m) \leq \frac{4 L_r}{\eta K \gamma_m}. \tag{142}$$

$\square$

**Lemma 13** (Restatement of Lemma 5). $L_{r+1} \leq L_r$ *for every* $r \geq 0$. *Further, if* $\rho_r^m = \max_{m' \in [M]} \rho_r^{m'}$, *then*

$$\rho_{r+1}^m \leq L_r - \frac{(1+c)\gamma_m}{4(L_0 + 1)^2 (1 + \exp(-\gamma_m a_r^m))} L_r^2, \tag{143}$$

*and if* $\rho_{r+1}^m \geq \rho_r^m$, *then*

$$\rho_{r+1}^m \leq \frac{1-c}{2} L_r. \tag{144}$$

*Proof.* Assume without loss of generality that $\rho_r^1 \geq \rho_r^2$ (an identical proof works in the remaining case by switching indices), so that $L_r = \rho_r^1$. To show that $L_{r+1} \leq L_r$, we must show that

$$\rho_{r+1}^m \leq \rho_r^1 \tag{145}$$

for $m \in \{1, 2\}$.

Starting with $m = 1$, the recurrence relation of $a_r^1$ from Equation 133 implies

$$a_{r+1}^1 = a_r^1 + \frac{1}{2}\rho_r^1 + \frac{c}{2}\rho_r^2 \tag{146}$$

$$= a_r^1 + \frac{1+c}{4}\left(\rho_r^1 + \rho_r^2\right) + \frac{1-c}{4}\left(\rho_r^1 - \rho_r^2\right) \tag{147}$$

$$\overset{(i)}{\geq} a_r^1 + \frac{1+c}{4}\left(\rho_r^1 + \rho_r^2\right) \tag{148}$$

$$\geq a_r^1 + \frac{1+c}{4}\rho_r^1 \tag{149}$$

where $(i)$ uses the assumption $\rho_r^1 \geq \rho_r^2$. Therefore

$$\rho_{r+1}^1 = \frac{1}{\gamma_1} \log\left(\Phi(\eta K \gamma_1^2, \gamma_1 a_{r+1})\right) \overset{(i)}{\leq} \frac{1}{\gamma_1} \log\left(\Phi(\eta K \gamma_1^2, \gamma_1 a_r)\right) = \rho_r^1, \tag{150}$$

where $(i)$ uses $a_{r+1}^1 \geq a_r^1$ together with the fact that $\Phi(b, x)$ is decreasing in $x$ (from Lemma 30). This proves Equation 145 for $m = 1$.

For client $m = 2$, we consider two cases. If $\rho_{r+1}^2 \leq \rho_{r+1}^1$, then we are done, since

$$\rho_{r+1}^2 \leq \rho_r^2 \leq \rho_r^1. \tag{151}$$

In the other case, $\rho_{r+1}^2 \geq \rho_r^2$. Then

$$\frac{1}{\gamma_2}\Phi(\eta K \gamma_2^2, \gamma_2 a_{r+1}^2) \geq \frac{1}{\gamma_2}\Phi(\eta K \gamma_2^2, \gamma_2 a_r^2), \tag{152}$$

so $a_{r+1}^2 \leq a_r^2$, since $\Phi(b, \cdot)$ is decreasing (Lemma 30). Therefore

$$\rho_{r+1}^2 = \frac{1}{\gamma_2} \log\left(\Phi(\eta K \gamma_2^2, \gamma_2 a_{r+1}^2)\right) \tag{153}$$

$$= \frac{1}{\gamma_2} \log\left(\Phi(\eta K \gamma_2^2, \gamma_2 a_r^2 + \gamma_2(a_{r+1}^2 - a_r^2))\right) \tag{154}$$

$$\overset{(i)}{\leq} \frac{1}{\gamma_2} \log\left(\Phi(\eta K \gamma_2^2, \gamma_2 a_r^2) \exp(\gamma_2(a_r^2 - a_{r+1}^2))\right) \tag{155}$$

$$= \frac{1}{\gamma_2} \log\left(\Phi(\eta K \gamma_2^2, \gamma_2 a_r^2)\right) + (a_r^2 - a_{r+1}^2) \tag{156}$$

$$\overset{(ii)}{=} \rho_r^2 - \frac{c}{2}\rho_r^1 - \frac{1}{2}\rho_r^2 \tag{157}$$

$$= \frac{1}{2}\rho_r^2 - \frac{c}{2}\rho_r^1 \tag{158}$$

$$\overset{(iii)}{\leq} \frac{1-c}{2}\rho_r^1, \tag{159}$$

where $(i)$ uses Lemma 34, $(ii)$ uses the recurrence relation of $a_r^2$ from Equation 134, and $(iii)$ uses the assumption $\rho_r^1 \geq \rho_r^2$. This proves Equation 145 for $m = 2$, so that $L_{r+1} \leq L_r$.

To prove Equation 143, we continue from Equation 149. Denoting $\lambda = (1+c)/4$ and plugging into the definition of $\rho_{r+1}^1$,

$$\rho_{r+1}^1 = \frac{1}{\gamma_1} \log\left(\Phi\left(\eta K \gamma_1^2, \gamma_1 a_{r+1}^1\right)\right) \tag{160}$$

$$\overset{(i)}{\leq} \frac{1}{\gamma_1} \log\left(\Phi\left(\eta K \gamma_1^2, \gamma_1 a_r^1 + \lambda\gamma_1\rho_r^1\right)\right) \tag{161}$$

$$\overset{(ii)}{\leq} \frac{1}{\gamma_1} \log\left(\Phi\left(\eta K \gamma_1^2, \gamma_1 a_r^1\right)\left(1 + (\exp(-\lambda\gamma_1\rho_r^1) - 1)\frac{\Phi\left(\eta K \gamma_1^2, \gamma_1 a_r^1\right) - 1}{\Phi\left(\eta K \gamma_1^2, \gamma_1 a_r^1\right) + \exp(-\gamma_1 a_r^1)}\right)\right) \tag{162}$$

$$= \frac{1}{\gamma_1} \log\left(\Phi\left(\eta K \gamma_1^2, \gamma_1 a_r^1\right)\right) + \frac{1}{\gamma_1} \log\left(1 + \frac{(\exp(-\lambda\gamma_1\rho_r^1) - 1)(\Phi\left(\eta K \gamma_1^2, \gamma_1 a_r^1\right) - 1)}{\Phi\left(\eta K \gamma_1^2, \gamma_1 a_r^1\right) + \exp(-\gamma_1 a_r^1)}\right) \tag{163}$$

$$\leq \rho_r^1 + \frac{1}{\gamma_1}\frac{(\exp(-\lambda\gamma_1\rho_r^1) - 1)(\Phi\left(\eta K \gamma_1^2, \gamma_1 a_r^1\right) - 1)}{\Phi\left(\eta K \gamma_1^2, \gamma_1 a_r^1\right) + \exp(-\gamma_1 a_r^1)} \tag{164}$$

$$\overset{(iii)}{=} \rho_r^1 + \frac{1}{\gamma_1}\frac{(\exp(-\lambda\gamma_1\rho_r^1) - 1)(\exp(\gamma_1\rho_r^1) - 1)}{\exp(\gamma_1\rho_r^1) + \exp(-\gamma_1 a_r^1)} \tag{165}$$

$$= \rho_r^1 - \frac{1}{\gamma_1}\frac{(1 - \exp(-\lambda\gamma_1\rho_r^1))(1 - \exp(-\gamma_1\rho_r^1))}{1 + \exp(-\gamma_1 a_r^1 - \gamma_1\rho_r^1)} \tag{166}$$

$$\leq \rho_r^1 - \frac{1}{\gamma_1}\frac{(1 - \exp(-\lambda\gamma_1\rho_r^1))(1 - \exp(-\gamma_1\rho_r^1))}{1 + \exp(-\gamma_1 a_r^1)}, \tag{167}$$

where $(i)$ uses that $\Phi(b, \cdot)$ is decreasing together with Equation 149, $(ii)$ uses Lemma 34, and $(iii)$ uses the substitution $\Phi(\eta K \gamma_m^2, \gamma_m a_r^m) = \exp(\gamma_m \rho_r^m)$. We can further bound the terms in the numerator of Equation 167 as follows.

$$\lambda\gamma_1\rho_r^1 \leq \gamma_1\rho_r^1 \leq \rho_r^1 \leq L_r \leq L_0, \tag{168}$$

so $-\gamma_1\rho_r^1 \in [-L_0, 0]$. By convexity of $\exp(x) - 1$,

$$\exp(-\gamma_1\rho_r^1) - 1 \leq \frac{\gamma_1\rho_r^1}{L_0}(\exp(-L_0) - 1) + \left(1 - \frac{\gamma_1\rho_r^1}{L_0}\right)(\exp(0) - 1) \tag{169}$$

$$= \gamma_1\rho_r^1\frac{1 - \exp(L_0)}{L_0\exp(L_0)}, \tag{170}$$

and similarly

$$\exp(-\lambda\gamma_1\rho_r^1) - 1 \leq \lambda\gamma_1\rho_r^1 \frac{1 - \exp(L_0)}{L_0 \exp(L_0)}. \tag{171}$$

Notice that

$$\frac{L_0 \exp(L_0)}{\exp(L_0) - 1} \overset{(i)}{\leq} \frac{L_0 \exp(L_0) + \exp(L_0) - (L_0 + 1)}{\exp(L_0) - 1} = \frac{(L_0 + 1)(\exp(L_0) - 1)}{\exp(L_0) - 1} = L_0 + 1, \tag{172}$$

where $(i)$ uses $\exp(x) - (x + 1) \geq 0$, so

$$\frac{1 - \exp(L_0)}{L_0 \exp(L_0)} \leq -\frac{1}{L_0 + 1}. \tag{173}$$

Combining this with Equation 170 and Equation 171 yields

$$(1 - \exp(-\lambda\gamma_1\rho_r^1))(1 - \exp(-\gamma_1\rho_r^1)) \geq \frac{\lambda\gamma_1^2}{(1 + L_0)^2}(\rho_r^1)^2. \tag{174}$$

Plugging this back into Equation 167,

$$\rho_{r+1}^1 = \rho_r^1 - \frac{\lambda\gamma_1}{(1 + L_0)^2(1 + \exp(-\gamma_1 a_r^1))}(\rho_r^1)^2, \tag{175}$$

and plugging in the definition of $\lambda$ gives Equation 143.

Finally, we have already proven Equation 144 in Equation 159. $\qquad\square$

**Lemma 14** (Restatement of Lemma 6). *Denote* $m_r = \arg\max_{m \in [M]} \rho_r^m$ *and* $\alpha_r = a_r^{m_r}$. *Let* $0 \leq q \leq r$. *If* $\alpha_s \geq A$ *for every* $q \leq s \leq r$, *then*

$$L_r \leq \frac{1}{1/L_q + \nu(r - q)/2}, \tag{176}$$

*where*

$$\nu := \frac{(1 + c)\gamma_{\min}}{4(L_0 + 1)^2(1 + \exp(-\gamma_{\max}A))}. \tag{177}$$

*Proof.* Lemma 5 gives an upper bound on the change of the surrogate losses $\rho_r^m$ after a single round, under some conditions. In this proof, we use these one-step decreases to derive an upper bound on $L_r$. The idea is to show that, after two steps, $L_s$ decreases proportionally to its square:

$$L_{s+2} \leq L_s - \nu L_s^2. \tag{178}$$

For each $s$ with $q \leq s \leq r$, we prove Equation 178 by considering three cases. Again, we assume without loss of generality that $\rho_s^1 \geq \rho_s^2$.

**Case 1:** $\rho_{s+1}^2 \leq \rho_{s+1}^1$. This case is easy, since

$$L_{s+2} \overset{(i)}{\leq} L_{s+1} \tag{179}$$

$$\overset{(ii)}{=} \rho_{s+1}^1 \tag{180}$$

$$\overset{(iii)}{\leq} L_s - \frac{(1 + c)\gamma_1}{4(1 + L_0)^2(1 + \exp(-\gamma_1 a_s^1))} L_s^2 \tag{181}$$

$$\overset{(iv)}{\leq} L_s - \frac{(1 + c)\gamma_{\min}}{4(1 + L_0)^2(1 + \exp(-\gamma_{\max}A))} L_s^2 \tag{182}$$

$$= L_s - \nu L_s^2, \tag{183}$$

where $(i)$ uses the fact that $L_r$ decreases monotonically (Lemma 5), $(ii)$ uses the assumption $\rho_{r+1}^2 \leq \rho_{r+1}^1$, $(iii)$ uses Equation 24 from Lemma 5, and $(iv)$ uses the condition $A \leq \alpha_s$ together with $\alpha_s = a_s^1$.

**Case 2: $\rho_{s+1}^2 \geq \rho_{s+1}^1$ and $\rho_{s+2}^2 \geq \rho_{s+2}^1$.** Since $\rho_{s+2}^2 \geq \rho_{s+2}^1$, we have $L_{s+2} = \rho_{s+2}^2$. Therefore

$$L_{s+2} = \rho_{s+2}^2 \tag{184}$$

$$\overset{(i)}{\leq} \rho_{s+1}^2 - \frac{(1+c)\gamma_1}{4(1+L_0)^2(1+\exp(-\gamma_1 a_s^1))}(\rho_{s+1}^2)^2 \tag{185}$$

$$\overset{(ii)}{=} L_{s+1} - \frac{(1+c)\gamma_1}{4(1+L_0)^2(1+\exp(-\gamma_1 a_s^1))}L_{s+1}^2 \tag{186}$$

$$\overset{(iii)}{=} L_{s+1} - \frac{(1+c)\gamma_{\min}}{4(1+L_0)^2(1+\exp(-\gamma_{\max}A))}L_{s+1}^2 \tag{187}$$

$$= L_{s+1} - \nu L_{s+1}^2 \tag{188}$$

$$\overset{(iv)}{\leq} L_s - \nu L_s^2 \tag{189}$$

where $(i)$ uses the case assumption $\rho_{r+1}^2 \geq \rho_{r+1}^1$ together with Equation 24 of Lemma 5, $(ii)$ uses the same case assumption, $(iii)$ uses ther condition $A \leq \alpha_s$ together with $\alpha_s = a_s^1$, and $(iv)$ uses the fact that the mapping $x \mapsto x - a(x-1)^2$ is increasing on $[0, 1/2a]$ together with $L_{s+1} \leq L_s \leq \ldots \leq L_0$ from Lemma 5.

**Case 3: $\rho_{s+1}^2 \geq \rho_{s+1}^1$ and $\rho_{s+2}^2 \leq \rho_{s+2}^1$.** From the case assumptions, $L_{s+2} = \rho_{s+2}^1$ and $L_{s+1} = \rho_{s+1}^2$. The bound on $L_{s+2}$ in this case will depend on whether $\rho_{s+2}^1 \leq \rho_{s+1}^1$. If this happens, then

$$L_{s+2} = \rho_{s+2}^1 \tag{190}$$

$$\leq \rho_{s+1}^1 \tag{191}$$

$$\overset{(i)}{\leq} \rho_s^1 - \frac{(1+c)\gamma_{\min}}{4(1+L_0)^2(1+\exp(-\gamma_1 a_s^1))}(\rho_s^1)^2 \tag{192}$$

$$= L_s - \frac{(1+c)\gamma_1}{4(1+L_0)^2(1+\exp(-\gamma_1 a_s^1))}L_s^2 \tag{193}$$

$$\overset{(ii)}{\leq} L_s - \frac{(1+c)\gamma_{\min}}{4(1+L_0)^2(1+\exp(-\gamma_{\max}A))}L_s^2 \tag{194}$$

$$= L_s - \nu L_s^2, \tag{195}$$

where $(i)$ uses Equation 24 from Lemma 5 and $(ii)$ uses the condition $A \geq \alpha_s$ together with $\alpha_s = a_s^1$.

On the other hand, if $\rho_{s+2}^1 \geq \rho_{s+1}^1$, then

$$L_{s+2} = \rho_{s+2}^1 \overset{(i)}{\leq} \frac{1-c}{2}L_{s+1} \overset{(ii)}{\leq} \frac{1-c}{2}L_s \tag{196}$$

where $(i)$ uses Lemma 5 and $(ii)$ uses $L_{s+1} \leq L_s$ from Lemma 5. Notice that

$$\frac{1+c}{2}\frac{1}{\nu} = \frac{1+c}{2}\frac{4(1+L_0)^2(1+\exp(-\gamma_{\max}A))}{(1+c)\gamma_{\min}} \tag{197}$$

$$= \frac{2}{\gamma_{\min}}(1+L_0)^2(1+\exp(-\gamma_{\max}A)) \tag{198}$$

$$\geq L_0, \tag{199}$$

so

$$\nu L_s \leq \nu L_0 \leq \frac{1+c}{2} \tag{200}$$

$$1 - \frac{1+c}{2} \leq 1 - \nu L_s \tag{201}$$

$$\frac{1-c}{2} \leq 1 - \nu L_s \tag{202}$$

$$\frac{1-c}{2}L_s \leq L_s - \nu L_s^2. \tag{203}$$

Therefore the RHS of Equation 196 can be bounded as

$$L_{s+2} \leq \frac{1-c}{2} L_2 \leq L_2 - \nu L_s^2. \tag{204}$$

This covers all cases and completes the proof of Equation 178. All that remains is to unroll the recursive upper bound of $L_s$ given by Equation 178. For every $k$ with $0 \leq k \leq (r-q)/2$,

$$L_{q+2k} \leq L_{q+2(k-1)} - \nu L_{q+2(k-1)}^2 \tag{205}$$

$$\frac{1}{L_{q+2(k-1)}} \leq \frac{1}{L_{q+2k}} - \nu \frac{L_{q+2(k-1)}}{L_{q+2k}} \tag{206}$$

$$\frac{1}{L_{q+2k}} \geq \frac{1}{L_{q+2(k-1)}} + \nu \frac{L_{q+2(k-1)}}{L_{q+2k}} \tag{207}$$

$$\frac{1}{L_{q+2k}} \overset{(i)}{\geq} \frac{1}{L_{q+2(k-1)}} + \nu \tag{208}$$

$$\frac{1}{L_{q+2k}} \geq \frac{1}{L_q} + \nu k \tag{209}$$

$$L_{q+2k} \leq \frac{1}{1/L_q + \nu k}, \tag{210}$$

where $(i)$ uses the fact that $L_s$ is monotonically decreasing (Lemma 5). Choosing $k = (r-q)/2$ gives Equation 176. $\qquad \square$

**Lemma 15.** *Denote $H_0 = \min_{m \in [M]} \rho_0^m$. For every $m \in [M]$ and $r \geq 0$,*

$$a_r^m \geq -\frac{3(1-c)(L_0+1)^2}{(1+c)\gamma_{\min}} \log\left(\frac{L_0}{H_0}\right). \tag{211}$$

*Proof.* Assume without loss of generality that $\alpha_0^1 \geq \alpha_0^2$. Define $X = \{r \geq 0 : (\rho_s^1 \geq \rho_s^2) \text{ for all } s \leq r \text{ and } \rho_r^1 \geq \rho_0^2\}$ and $q = \sup X$. Then for every $r \geq 0$,

$$\frac{1}{\gamma_1} \log\left(\Phi(\eta K \gamma_1^2, \gamma_1 a_r^1)\right) = \rho_r^1 \leq L_r \overset{(i)}{\leq} L_0 = \rho_0^1 = \Phi(\eta K \gamma_1^2, \gamma_1 a_0^1) = \frac{1}{\gamma_1} \log\left(\Phi(\eta K \gamma_1^2, 0)\right), \tag{212}$$

where $(i)$ uses the fact that $L_r$ is monotonically decreasing (Lemma 5). Also, since $\Phi(b, \cdot)$ is strictly decreasing, the above implies that $a_r^1 \geq 0$ for every $r \geq 0$.

Now, for every $r \leq q$, the definition of $q$ implies that $\rho_r^1 \geq \rho_r^2$. Therefore Equation 24 from Lemma 5 implies that

$$\rho_{r+1}^1 \leq L_r - \frac{(1+c)\gamma_1}{4(L_0+1)^2(1+\exp(-\gamma_1 a_r^1))} L_r^2 \tag{213}$$

$$= \rho_r^1 - \frac{(1+c)\gamma_1}{4(L_0+1)^2(1+\exp(-\gamma_1 a_r^1))} (\rho_r^1)^2 \tag{214}$$

$$\overset{(i)}{\leq} \rho_r^1 - \frac{(1+c)\gamma_1}{8(L_0+1)^2} (\rho_r^1)^2, \tag{215}$$

where $(i)$ uses the fact that $a_r^1 \geq 0$. Denoting $\beta = \frac{(1+c)\gamma_1}{8(L_0+1)^2}$, we can then unroll this recursion:

$$\rho_{r+1}^1 \leq \rho_r^1 - \beta(\rho_r^1)^2 \tag{216}$$

$$\frac{1}{\rho_r^1} \leq \frac{1}{\rho_{r+1}^1} - \beta\frac{\rho_r^1}{\rho_{r+1}^1} \tag{217}$$

$$\frac{1}{\rho_{r+1}^1} \geq \frac{1}{\rho_r^1} + \beta\frac{\rho_r^1}{\rho_{r+1}^1} \tag{218}$$

$$\frac{1}{\rho_{r+1}^1} \overset{(i)}{\geq} \frac{1}{\rho_r^1} + \beta \tag{219}$$

$$\frac{1}{\rho_{r+1}^1} \geq \frac{1}{\rho_0^1} + \beta(r+1) \tag{220}$$

$$\rho_{r+1}^1 \leq \frac{1}{1/\rho_0^1 + \beta(r+1)}, \tag{221}$$

where $(i)$ uses $\rho_{r+1}^1 \leq \rho_r^1$ from Equation 215. Choosing $r = q - 1$ yields

$$\rho_q^1 \leq \frac{1}{1/\rho_0^1 + \beta q}. \tag{222}$$

From the definition of $q$, we also have $\rho_0^2 \leq \rho_q^1$, so

$$\rho_0^2 \leq \frac{1}{1/\rho_0^1 + \beta q} \tag{223}$$

$$\frac{1}{\rho_0^1} + \beta q \leq \frac{1}{\rho_0^2} \tag{224}$$

$$q \leq \frac{1}{\beta}\left(\frac{1}{\rho_0^2} - \frac{1}{\rho_0^1}\right). \tag{225}$$

Now, we claim that for all $r \geq 0$,

$$a_r^2 \geq \min_{s \leq q+1} a_s^2. \tag{226}$$

To see this, we consider two cases. Since $q + 1 \notin X$, we know that either (1) $\rho_{q+1}^2 > \rho_{q+1}^1$ or (2) $\rho_{q+1}^1 < \rho_0^2$.

**Case 1:** $\rho_{q+1}^2 > \rho_{q+1}^1$. In this case, $L_{q+1} = \rho_{q+1}^2$, so for all $r > q$,

$$\frac{1}{\gamma_2}\log\left(\Phi(\eta K\gamma_2^2, \gamma_2 a_r^2)\right) = \rho_r^2 \leq L_r \overset{(i)}{\leq} L_{q+1} = \rho_{q+1}^2 = \frac{1}{\gamma_2}\log\left(\Phi(\eta K\gamma_2^2, \gamma_2 a_{q+1}^2)\right), \tag{227}$$

where $(i)$ uses that $L_r$ is monotonically decreasing (Lemma 5). Since $\Phi(b, \cdot)$ is decreasing (Lemma 30), this means $a_r^2 \geq a_{q+1}^2$. Therefore, more generally, $a_r^2 \geq \min_{s \leq q+1} a_s^2$ for all $r \geq 0$.

**Case 2:** $\rho_{q+1}^2 \leq \rho_{q+1}^1$. In this case, we must have $\rho_{q+1}^1 < \rho_0^2$. Therefore, for all $r > q$,

$$\frac{1}{\gamma_2}\log\left(\Phi(\eta K\gamma_2^2, \gamma_2 a_r^2)\right) = \rho_r^2 \leq L_r \overset{(i)}{\leq} L_{q+1} = \rho_{q+1}^1 < \rho_0^2 = \frac{1}{\gamma_2}\log\left(\Phi(\eta K\gamma_2^2, 0)\right), \tag{228}$$

where $(i)$ uses that $L_r$ is monotonically decreasing (Lemma 5). Since $\Phi(b, \cdot)$ is decreasing (Lemma 30), this means $a_r^2 \geq 0$. Therefore, more generally, $a_r^2 \geq \min_{s \leq q} a_s^2$ for all $r \geq 0$, since $a_0^2 = 0$ implies that $\min_{s \leq q} a_s^2 \leq 0$.

This proves the claim in both cases. To finish the proof, we will use Equation 225 together with the recurrence relation of $a_r^m$ to lower bound $\min_{s \leq q+1} a_s^2$. Starting from Equation 134, for every

$s \leq q$,

$$a_{s+1}^2 = a_s^2 + \frac{c}{2}\rho_s^1 + \frac{1}{2}\rho_s^2 \tag{229}$$

$$= a_s^2 + \frac{1+c}{4}(\rho_s^1 + \rho_s^2) + \frac{1-c}{4}(\rho_s^2 - \rho_s^1) \tag{230}$$

$$\geq a_s^2 + \frac{1-c}{4}(\rho_s^2 - \rho_s^1) \tag{231}$$

$$\geq a_s^2 - \frac{1-c}{4}\rho_s^1, \tag{232}$$

and unrolling yields that

$$a_s^2 \geq a_0^2 - \frac{1-c}{4}\sum_{t=0}^{s-1}\rho_t^1 = -\frac{1-c}{4}\sum_{t=0}^{s-1}\rho_t^1 \geq -\frac{1-c}{4}\sum_{t=0}^{q}\rho_t^1. \tag{233}$$

We can plug in the bound of $\rho_t^1$ from Equation 221 to obtain

$$a_s^2 \geq -\frac{1-c}{4}\sum_{t=0}^{q}\frac{1}{1/\rho_0^1 + \beta t} \tag{234}$$

$$= -\frac{1-c}{4}\sum_{t=0}^{q}\frac{1}{1/L_0 + \beta t} \tag{235}$$

$$\geq -\frac{1-c}{4}\left(L_0 + \int_0^q \frac{1}{1/L_0 + \beta x}dx\right) \tag{236}$$

$$= -\frac{1-c}{4}\left(L_0 + \left[\frac{1}{\beta}\log(1/L_0 + \beta x)\right]_0^q\right) \tag{237}$$

$$= -\frac{1-c}{4}\left(L_0 + \frac{1}{\beta}\log(1 + \beta L_0 q)\right) \tag{238}$$

$$\overset{(i)}{\geq} -\frac{1-c}{4}\left(L_0 + \frac{1}{\beta}\log\left(1 + L_0\left(\frac{1}{\rho_0^2} - \frac{1}{\rho_0^1}\right)\right)\right) \tag{239}$$

$$= -\frac{1-c}{4}\left(L_0 + \frac{1}{\beta}\log\left(\frac{\rho_0^1}{\rho_0^2}\right)\right) \tag{240}$$

$$\overset{(ii)}{=} -\frac{1-c}{4}\left(L_0 + \frac{8(L_0+1)^2}{(1+c)\gamma_1}\log\left(\frac{\rho_0^1}{\rho_0^2}\right)\right) \tag{241}$$

$$\geq -\frac{3(1-c)(L_0+1)^2}{(1+c)\gamma_{\min}}\log\left(\frac{\rho_0^1}{\rho_0^2}\right), \tag{242}$$

where $(i)$ uses the bound of $q$ in Equation 225 and $(ii)$ uses the definition of $\beta$. Combining this with Equation 226 gives the desired result. □

**Lemma 16.** *Let*

$$A_0 = \frac{3(1-c)(L_0+1)^2}{(1+c)\gamma_{\min}}\log\left(\frac{L_0}{H_0}\right) \tag{243}$$

$$\nu_0 = \frac{(1+c)\gamma_{\min}}{4(L_0+1)^2(1+\exp(-\gamma_{\max}A_0))} \tag{244}$$

$$\tau_0 = \frac{2}{\nu_0}\left(\frac{1}{H_0} - \frac{1}{L_0}\right). \tag{245}$$

*Then $a_r^m \geq 0$ for every $m \in [M]$ and $r \geq \tau_0$.*

*Proof.* In order for $a_r^m \geq 0 = a_0^m$, it suffices that $\rho_r^m \leq \rho_0^m$. Since $L_r = \max_{m \in [M]} \rho_r^m$, this can be guaranteed when

$$L_r \leq H_0 := \min_{m \in [M]} \rho_0^m. \tag{246}$$

Therefore, we only need to show that $L_r \leq H_0$ for all $r \geq \tau_0$.

Lemma 15 tells us that $a_r^m \geq -A_0$ for every $m \in [M]$ and $r \geq 0$. Therefore, we can apply Lemma 6 with $q = 0$, $A = A_0$, and any $r \geq 0$ to conclude that

$$L_r \leq \frac{1}{1/L_0 + \nu_0 r/2}. \tag{247}$$

For any $r \geq \tau_0$,

$$L_r \leq \frac{1}{1/L_0 + \nu_0/2\tau_0} = \frac{1}{1/L_0 + (1/H_0 - 1/L_0)} = H_0. \tag{248}$$

This shows that $a_r^m \geq 0$ for every $r \geq \tau_0$. $\qquad \square$

**Theorem 4** (Restatement of Theorem 2). *Define*

$$\tau_1 = \tau_0 + \frac{32(L_0 + 1)^2}{(1 + c)\gamma_{\min}} \left( 2\gamma_{\max} + \frac{1}{H_0} \right). \tag{249}$$

*Then for every $r \geq \tau_1$,*

$$F(\bar{\boldsymbol{w}}_r) \leq \frac{64(L_0 + 1)^2}{(1 + c)\gamma_{\min}^2 \eta K(r - \tau_0)}. \tag{250}$$

*where*

$$L_0 = \max_{m \in [M]} \frac{1}{\gamma_m} \log\left(1 + \eta K \gamma_m^2\right), \qquad H_0 = \min_{m \in [M]} \frac{1}{\gamma_m} \log\left(1 + \eta K \gamma_m^2\right). \tag{251}$$

*Proof.* The result follows by applying a combination of Lemmas 12, 6, 15, and 16.

By Lemma 16, we know that $a_r^m \geq 0$ for all $m \in [M]$ and $r \geq \tau_0$. Therefore we can Lemma 6 with $q = \tau_0$ and $A = 0$, so that for all $r \geq \tau_0$:

$$L_r \leq \frac{1}{1/L_{\tau_0} + \nu_1(r - \tau_0)}. \tag{252}$$

where we denoted

$$\nu_1 = \frac{(1 + c)\gamma_{\min}}{16(L_0 + 1)^2}. \tag{253}$$

By Equation 248 from Lemma 16, we already know that $L_{\tau_0} \leq H_0$, so

$$L_r \leq \frac{1}{1/H_0 + \nu_1(r - \tau_0)}. \tag{254}$$

We would like to use Lemma 12 to bound $F(\bar{\boldsymbol{w}}_r)$ in terms of $L_r$; in order to do this, we need to ensure that the condition of Lemma 12 is satisfied for all $m$, i.e.

$$L_r \leq \min\left\{ \frac{1}{2\gamma_m}, \frac{1}{\gamma_m} \log\left(1 + \frac{\eta K \gamma_m^2}{2 + \eta K \gamma_m^2}\right) \right\}. \tag{255}$$

Notice for each $m$, if $\eta K \gamma_2^2 \leq 1$, then

$$\frac{1}{\gamma_m} \log\left(1 + \frac{\eta K \gamma_m^2}{2 + \eta K \gamma_m^2}\right) \geq \frac{1}{\gamma_m} \log\left(1 + \frac{1}{3}\eta K \gamma_m^2\right) \overset{(i)}{\geq} \frac{1}{3\gamma_m} \log\left(1 + \eta K \gamma_m^2\right) \geq \frac{H_0}{3}, \tag{256}$$

where $(i)$ uses $1 + ax \geq (1 + x)^a \implies \log(1 + ax) \geq a \log(1 + x)$ when $a \in (0, 1)$ by concavity of $(1 + x)^a$. On the other hand, if $\eta K \gamma_2^2 \geq 1$, then

$$\frac{1}{\gamma_m} \log\left(1 + \frac{\eta K \gamma_m^2}{2 + \eta K \gamma_m^2}\right) \geq \frac{1}{\gamma_m} \log\left(1 + \frac{1}{3}\right) \geq \frac{1}{4\gamma_m}. \tag{257}$$

Therefore

$$\frac{1}{\gamma_m} \log\left(1 + \frac{\eta K \gamma_m^2}{2 + \eta K \gamma_m^2}\right) \geq \min\left\{ \frac{1}{4\gamma_m}, \frac{H_0}{3} \right\}. \tag{258}$$

So to prove Equation 255, it suffices to show

$$L_r \leq \min\left\{\frac{1}{4\gamma_m}, \frac{H_0}{3}\right\}. \tag{259}$$

We will show that this is satisfied for every $r \geq \tau_1$. From Equation 254, for all $r \geq \tau_1$:

$$r - \tau_0 \geq \tau_1 - \tau_0 \tag{260}$$

$$r - \tau_0 \geq \frac{2}{\nu_1}\left(2\gamma_{\max} + \frac{1}{H_0}\right) \tag{261}$$

$$\nu_1(r - \tau_0) \geq 4\gamma_{\max} + \frac{2}{H_0} \tag{262}$$

$$1/H_0 + \nu_1(r - \tau_0) \geq 4\gamma_{\max} + \frac{3}{H_0} \tag{263}$$

$$1/H_0 + \nu_1(r - \tau_0) \geq \max\left\{4\gamma_{\max}, \frac{3}{H_0}\right\}. \tag{264}$$

Therefore, from Equation 254,

$$L_r \leq \frac{1}{1/H_0 + \nu_1(r - \tau_0)} \leq \min\left\{\frac{1}{4\gamma_{\max}}, \frac{H_0}{3}\right\}, \tag{265}$$

which is exactly Equation 259. Therefore, the condition of Lemma 12 is satisfied for all $r \geq \tau_1$.

Finally, Equation 254 and Lemma 12 imply that for all $r \geq \tau_1$,

$$F_m(\bar{\boldsymbol{w}}_r) \leq \frac{4L_r}{\eta K \gamma_{\min}} \tag{266}$$

$$\leq \frac{4}{\eta K \gamma_{\min}} \frac{1}{1/H_0 + \nu_1(r - \tau_0)} \tag{267}$$

$$\leq \frac{4}{\nu_1 \gamma_{\min} \eta K (r - \tau_0)} \tag{268}$$

$$= \frac{64(L_0 + 1)^2}{(1 + c)\gamma_{\min}^2 \eta K (r - \tau_0)}. \tag{269}$$

□

## C  EXTENDING WORST-CASE BASELINES

We formally describe the problem in Section C.1, and results are stated in Section C.2.

### C.1  SETUP

We consider the following optimization problem:

$$\min_{\boldsymbol{w} \in \mathbb{R}^d}\left\{F(\boldsymbol{w}) := \frac{1}{M}\sum_{m=1}^{M} F_m(\boldsymbol{w}) := \frac{1}{M}\sum_{m=1}^{M} \mathbb{E}_{\xi \sim \mathcal{D}_m} f(\boldsymbol{w}; \xi)\right\} \tag{270}$$

**Assumption 1.**

- *$f(\cdot, \xi)$ is convex and $H$-smooth for every $\xi$.*

- *There exists $F_* \in \mathbb{R}$ such that $F(\boldsymbol{w}) \geq F_*$ for every $\boldsymbol{w} \in \mathbb{R}^d$.*

- *For all $\boldsymbol{w} \in \mathbb{R}^d$: $\mathbb{E}_{\xi \sim \mathcal{D}_i}\left[\nabla f(\boldsymbol{w}, \xi)\right] = \nabla F_i(\boldsymbol{w})$.*

Note that we do not assume that $f$ achieves its infimum at some point in the domain.

**Assumption 2** (Stochastic Gradient Variance)**.**

---

**Algorithm 4** Local SGD

---

**Input:** Initialization $\bar{w}_0 \in \mathbb{R}^d$, rounds $R \in \mathbb{N}$, local steps $K \in \mathbb{N}$, learning rate $\eta > 0$, averaging weights $\{\alpha_{r,k}\}_{r,k}$
 1: **for** $r = 0, 1, \ldots, R-1$ **do**
 2:     **for** $m \in [M]$ **do**
 3:         $w_{r,0}^m \leftarrow \bar{w}_r$
 4:         **for** $k = 0, \ldots, K-1$ **do**
 5:             Sample $\xi_{r,k}^m \sim \mathcal{D}_m$
 6:             $w_{r,k+1}^m \leftarrow w_{r,k}^m - \eta \nabla f(w_{r,k}^m; \xi_{r,k}^m)$
 7:         **end for**
 8:     **end for**
 9:     $\bar{w}_{r+1} \leftarrow \frac{1}{M} \sum_{m=1}^M w_{r,K}^m$
10: **end for**
11: **return** $\hat{w} = \sum_{r=0}^{R-1} \sum_{k=0}^{K-1} \alpha_{r,k} \left( \frac{1}{M} \sum_{m=1}^M w_{r,k}^m \right)$

---

    *(a) (Global) There exists $\sigma \geq 0$ such that for all $w \in \mathbb{R}^d$ and $m \in [M]$:*

$$\mathbb{E}_{\xi \sim \mathcal{D}_m} \left[ \|\nabla f(w, \xi) - \nabla F_m(w)\|^2 \right] \leq \sigma^2. \tag{271}$$

    *(b) (Local) For every $u \in \mathbb{R}^d$, there exists $\sigma(u) \geq 0$ such that for all $m \in [M]$:*

$$\mathbb{E}_{\xi \sim \mathcal{D}_m} \left[ \|\nabla f(u, \xi) - \nabla F_m(u)\|^2 \right] \leq \sigma^2(u). \tag{272}$$

**Assumption 3** (Objective Heterogeneity).

    *(a) (Global): There exists $\zeta \geq 0$ such that for all $w \in \mathbb{R}^d$ and $m \in [M]$:*

$$\|\nabla F_m(w) - \nabla F(w)\| \leq \zeta. \tag{273}$$

    *(b) (Local): For every $u \in \mathbb{R}^d$, there exists $\zeta(u) \geq 0$ such that for all $m \in [M]$:*

$$\|\nabla F_m(u) - \nabla F(u)\| \leq \zeta(u). \tag{274}$$

Local SGD for the above optimization problem is defined in Algorithm 4.

### C.2 STATEMENT OF GENERAL CONVERGENCE RESULTS

Theorems 5 and 6 below are proven by modifying two existing analyses of Local SGD Woodworth et al. (2020b); Koloskova et al. (2020) which use global and local assumptions (respectively) on stochastic gradient variance and objective heterogeneity, by removing the assumption that the global objective $F$ has a minimizer $w_*$. The resulting rates match the corresponding rates from the original analyses, up to an additional additive term proportional to $F(u) - F_*$. The convex combination weights $\{\alpha_{r,k}\}_{r,k}$ are specified separately in each proof.

**Theorem 5.** *Let $B = \|\bar{w}_0 - u\|$. Under Assumptions 1, 2(a), and 3(a), for any $u \in \mathbb{R}^d$, there exists a choice of $\eta$ such that Local SGD satisfies*

$$\mathbb{E}\left[F(\hat{w}) - F_*\right] \leq \mathcal{O}\left( \frac{HB^2}{KR} + \frac{\sigma B}{\sqrt{MKR}} + \frac{(H\zeta^2 B^4)^{1/3}}{R^{2/3}} + \frac{(H\sigma^2 B^4)^{1/3}}{K^{1/3}R^{2/3}} + (F(u) - F_*) \right). \tag{275}$$

**Theorem 6.** *Let $B = \|\bar{w}_0 - u\|$. Under Assumptions 1, 2(b), and 3(b), for any $u \in \mathbb{R}^d$, there exists a choice of $\eta$ such that Local SGD satisfies*

$$\mathbb{E}[F(\hat{w}) - F_*] \leq$$
$$\mathcal{O}\left( \frac{HB^2}{R} + \frac{\sigma(u)B}{\sqrt{MKR}} + \frac{(H\sigma^2(u)B^4)^{1/3}}{K^{1/3}R^{2/3}} + \frac{(H\zeta^2(u)B^4)^{1/3}}{R^{2/3}} + R(F(u) - F_*) \right). \tag{276}$$

Proofs are given in Appendices C.3 and C.4, respectively.

### C.3 PROOF OF THEOREM 5

For this section, we use Assumptions 1, 2(a), and 3(a). For the analysis, we will consider the absolute timestep $t = Kr + k$, and re-index the algorithm's internal variables as $\boldsymbol{w}_t^m = \boldsymbol{w}_{r,k}^m$, and $\boldsymbol{g}_t^m = \boldsymbol{g}_{r,k}^m$, etc. Also, we will denote

$$\bar{\boldsymbol{w}}_t = \frac{1}{M} \sum_{m=1}^{M} \boldsymbol{w}_t^m. \tag{277}$$

The following lemma is slightly modified from Woodworth et al. (2020b) in order to avoid the assumption that some $\boldsymbol{w}_*$ exists.

**Lemma 17.** *If $\eta \le 1/(4H)$, then for any $\boldsymbol{u} \in \mathbb{R}^d$,*

$$\mathbb{E}\left[F(\bar{\boldsymbol{w}}_t) - F(\boldsymbol{u})\right] \le \frac{1}{\eta}\left(\mathbb{E}\left[\|\bar{\boldsymbol{w}}_t - \boldsymbol{u}\|^2\right] - \mathbb{E}\left[\|\bar{\boldsymbol{w}}_{t+1} - \boldsymbol{u}\|^2\right]\right) + \tag{278}$$

$$\frac{\eta\sigma^2}{M} + \frac{3H}{2M}\sum_{m=1}^{M}\mathbb{E}\left[\|\boldsymbol{w}_t^m - \bar{\boldsymbol{w}}_t\|^2\right] + (F(\boldsymbol{u}) - F_*). \tag{279}$$

*Proof.*

$$\|\bar{\boldsymbol{w}}_{t+1} - \boldsymbol{u}\|^2 = \left\|\bar{\boldsymbol{w}}_t - \boldsymbol{u} - \frac{\eta}{M}\sum_{m=1}^{M}\boldsymbol{g}_t^m\right\|^2 \tag{280}$$

$$= \left\|\bar{\boldsymbol{w}}_t - \boldsymbol{u} - \frac{\eta}{M}\sum_{m=1}^{M}\nabla F_m(\boldsymbol{w}_t^m) - \left(\frac{\eta}{M}\sum_{m=1}^{M}\boldsymbol{g}_t^m - \nabla F_m(\boldsymbol{w}_t^m)\right)\right\|^2. \tag{281}$$

Taking conditional expectation $\mathbb{E}_t[\cdot] := \mathbb{E}[\cdot | \{\xi_s^m : s < t, m \in [M]\}]$:

$$\mathbb{E}_t\left[\|\bar{\boldsymbol{w}}_{t+1} - \boldsymbol{u}\|^2\right] = \left\|\bar{\boldsymbol{w}}_t - \boldsymbol{u} - \frac{\eta}{M}\sum_{m=1}^{M}\nabla F_m(\boldsymbol{w}_t^m)\right\|^2 + \mathbb{E}_t\left[\left\|\frac{\eta}{M}\sum_{m=1}^{M}\boldsymbol{g}_t^m - \nabla F_m(\boldsymbol{w}_t^m)\right\|^2\right]$$

$$\tag{282}$$

$$= \left\|\bar{\boldsymbol{w}}_t - \boldsymbol{u} - \frac{\eta}{M}\sum_{m=1}^{M}\nabla F_m(x_t^m)\right\|^2 + \frac{\eta^2}{M^2}\sum_{m=1}^{M}\mathbb{E}_t\left[\|\boldsymbol{g}_t^m - \nabla F_m(\boldsymbol{w}_t^m)\|^2\right]$$

$$\tag{283}$$

$$\le \underbrace{\left\|\bar{\boldsymbol{w}}_t - \boldsymbol{u} - \frac{\eta}{M}\sum_{m=1}^{M}\nabla F_m(\boldsymbol{w}_t^m)\right\|^2}_{A} + \frac{\eta^2\sigma^2}{M}. \tag{284}$$

To bound $A$, we decompose

$$A = \|\bar{\boldsymbol{w}}_t - \boldsymbol{u}\|^2 + \frac{\eta^2}{M^2}\underbrace{\left\|\sum_{m=1}^{M}\nabla F_m(\boldsymbol{w}_t^m)\right\|^2}_{B_1} + \frac{2\eta}{M}\underbrace{\left\langle\bar{\boldsymbol{w}}_t - \boldsymbol{u}, -\sum_{m=1}^{M}\nabla F_m(\boldsymbol{w}_t^m)\right\rangle}_{B_2}. \tag{285}$$

We can bound $B_1$ and $B_2$ separately:

$$B_1 = \left\| \sum_{m=1}^{M} \nabla F_m(\bar{\boldsymbol{w}}_t) + \sum_{m=1}^{M} (\nabla F_m(\boldsymbol{w}_t^m) - \nabla F_m(\bar{\boldsymbol{w}}_t)) \right\|^2 \tag{286}$$

$$\leq 2 \left\| \sum_{m=1}^{M} \nabla F_m(\bar{\boldsymbol{w}}_t) \right\|^2 + 2 \left\| \sum_{m=1}^{M} \nabla F_m(\boldsymbol{w}_t^m) - \nabla F_m(\bar{\boldsymbol{w}}_t) \right\|^2 \tag{287}$$

$$\leq 2M^2 \|\nabla F(\bar{\boldsymbol{w}}_t)\|^2 + 2M \sum_{m=1}^{M} \|\nabla F_m(\boldsymbol{w}_t^m) - \nabla F_m(\bar{\boldsymbol{w}}_t)\|^2 \tag{288}$$

$$\leq 4HM^2(F(\bar{\boldsymbol{w}}_t) - F_*) + 2H^2 M \sum_{m=1}^{M} \|\boldsymbol{w}_t^m - \bar{\boldsymbol{w}}_t\|^2, \tag{289}$$

and

$$B_2 = - \sum_{m=1}^{M} \langle \bar{\boldsymbol{w}}_t - \boldsymbol{u}, \nabla F_m(\boldsymbol{w}_t^m) \rangle \tag{290}$$

$$= \sum_{m=1}^{M} \langle \boldsymbol{u} - \boldsymbol{w}_t^m, \nabla F_m(\boldsymbol{w}_t^m) \rangle - \sum_{m=1}^{M} \langle \bar{\boldsymbol{w}}_t - \boldsymbol{w}_t^m, \nabla F_m(\boldsymbol{w}_t^m) \rangle \tag{291}$$

$$\overset{(i)}{=} \sum_{m=1}^{M} (F(\boldsymbol{u}) - F(\boldsymbol{w}_t^m)) - \sum_{m=1}^{M} \left( F(\bar{\boldsymbol{w}}_t) - F(\boldsymbol{w}_t^m) - \frac{H}{2} \|\bar{\boldsymbol{w}}_t - \boldsymbol{w}_t^m\|^2 \right) \tag{292}$$

$$= -M(F(\bar{\boldsymbol{w}}_t) - F(\boldsymbol{u})) + \frac{H}{2} \sum_{m=1}^{M} \|\bar{\boldsymbol{w}}_t - \boldsymbol{w}_t^m\|^2, \tag{293}$$

where $(i)$ uses convexity and smoothness of $F$.

Plugging the resulting bound of $A$ back into Equation 284 yields

$$\mathbb{E}_t \left[ \|\bar{\boldsymbol{w}}_{t+1} - \boldsymbol{u}\|^2 \right]$$

$$\leq \|\bar{\boldsymbol{w}}_t - \boldsymbol{u}\|^2 + 4\eta^2 H(F(\bar{\boldsymbol{w}}_t) - F_*) + \frac{2\eta^2 H^2}{M} \sum_{m=1}^{M} \|\boldsymbol{w}_t^m - \bar{\boldsymbol{w}}_t\|^2 \tag{294}$$

$$- 2\eta(F(\bar{\boldsymbol{w}}_t) - F(\boldsymbol{u})) + \frac{\eta H}{M} \sum_{m=1}^{M} \|\bar{\boldsymbol{w}}_t - \boldsymbol{w}_t^m\|^2 + \frac{\eta^2 \sigma^2}{M} \tag{295}$$

$$\overset{(i)}{\leq} \|\bar{\boldsymbol{w}}_t - \boldsymbol{u}\|^2 + \eta(F(\bar{\boldsymbol{w}}_t) - F_*) + \frac{\eta H}{2M} \sum_{m=1}^{M} \|\boldsymbol{w}_t^m - \bar{\boldsymbol{w}}_t\|^2 \tag{296}$$

$$- 2\eta(F(\bar{\boldsymbol{w}}_t) - F(\boldsymbol{u})) + \frac{\eta H}{M} \sum_{m=1}^{M} \|\bar{\boldsymbol{w}}_t - \boldsymbol{w}_t^m\|^2 + \frac{\eta^2 \sigma^2}{M} \tag{297}$$

$$\overset{(ii)}{=} \|\bar{\boldsymbol{w}}_t - \boldsymbol{u}\|^2 - \eta(F(\bar{\boldsymbol{w}}_t) - F(\boldsymbol{u})) + \frac{3\eta H}{2M} \sum_{m=1}^{M} \|\boldsymbol{w}_t^m - \bar{\boldsymbol{w}}_t\|^2 + \eta(F(\boldsymbol{u}) - F_*) + \frac{\eta^2 \sigma^2}{M}, \tag{298}$$

where $(i)$ uses $\eta \leq 1/(4H)$, and $(ii)$ uses $F(\bar{\boldsymbol{w}}_t) - F_* = (F(\bar{\boldsymbol{w}}_t) - F(\boldsymbol{u})) + (F(\boldsymbol{u}) - F_*)$. Taking total expectation and rearranging yields

$$\mathbb{E}\left[F(\bar{\boldsymbol{w}}_t) - F(\boldsymbol{u})\right] \leq \frac{1}{\eta} \left( \mathbb{E}\left[ \|\bar{\boldsymbol{w}}_t - \boldsymbol{u}\|^2 \right] - \mathbb{E}\left[ \|\bar{\boldsymbol{w}}_{t+1} - \boldsymbol{u}\|^2 \right] \right) \tag{299}$$

$$+ \frac{\eta \sigma^2}{M} + \frac{3H}{2M} \sum_{m=1}^{M} \mathbb{E}\left[ \|\boldsymbol{w}_t^m - \bar{\boldsymbol{w}}_t\|^2 \right] + (F(\boldsymbol{u}) - F_*). \tag{300}$$

$\square$

The following lemma is exactly the same as in Woodworth et al. (2020b), and is unaffected by removing the assumption that $x_*$ exists.

**Lemma 18** (Lemma 8 of Woodworth et al. (2020b)). *For any $\eta > 0$,*

$$\frac{1}{M} \sum_{m=1}^{M} \mathbb{E}\left[\|\boldsymbol{w}_t^m - \bar{\boldsymbol{w}}_t\|^2\right] \leq 3K\sigma^2\eta^2 + 6K^2\eta^2\zeta^2.$$

*Proof of Theorem 5.* Let $\hat{\boldsymbol{w}} = \frac{1}{KR} \sum_{t=0}^{KR-1} \bar{\boldsymbol{w}}_t$. Combining Lemma 17 and Lemma 18:

$$\mathbb{E}\left[F(\bar{\boldsymbol{w}}_t) - F(\boldsymbol{u})\right] \leq \frac{1}{\eta}\left(\mathbb{E}\left[\|\bar{\boldsymbol{w}}_t - \boldsymbol{u}\|^2\right] - \mathbb{E}\left[\|\bar{\boldsymbol{w}}_{t+1} - \boldsymbol{u}\|^2\right]\right) \tag{301}$$

$$+ \frac{\eta\sigma^2}{M} + \frac{9}{2}K\sigma^2\eta^2 H + 9K^2\eta^2 H\zeta^2 + (F(\boldsymbol{u}) - F_*). \tag{302}$$

Averaging over $t$ and applying convexity of $F$ yields

$$\mathbb{E}\left[F(\hat{\boldsymbol{w}}) - F(\boldsymbol{u})\right] \leq \frac{1}{\eta KR}\left(\mathbb{E}\left[\|\bar{\boldsymbol{w}}_0 - \boldsymbol{u}\|^2\right] - \mathbb{E}\left[\|\bar{\boldsymbol{w}}_{KR} - \boldsymbol{u}\|^2\right]\right) \tag{303}$$

$$+ \frac{\eta\sigma^2}{M} + \frac{9}{2}K\sigma^2\eta^2 H + 9K^2\eta^2 H\zeta^2 + (F(\boldsymbol{u}) - F_*) \tag{304}$$

$$\leq \frac{\|\bar{\boldsymbol{w}}_0 - \boldsymbol{u}\|^2}{\eta KR} + \frac{9}{2}K\sigma^2\eta^2 H + 9K^2\eta^2 H\zeta^2 + (F(\boldsymbol{u}) - F_*). \tag{305}$$

Denote $B = \|\bar{\boldsymbol{w}}_0 - \boldsymbol{u}\|$. Identically as in Woodworth et al. (2020b), we can choose

$$\eta = \min\left\{\frac{1}{H}, \frac{B\sqrt{M}}{\sigma\sqrt{KR}}, \left(\frac{B^2}{HK^2R\sigma^2}\right)^{1/3}, \left(\frac{B^2}{HK^3R\zeta^2}\right)^{1/3}\right\}, \tag{306}$$

to guarantee

$$\mathbb{E}\left[F(\hat{\boldsymbol{w}}) - F(\boldsymbol{u})\right] \leq \frac{4HB^2}{KR} + \frac{\sigma B}{\sqrt{MKR}} + \frac{(H\zeta^2 B^4)^{1/3}}{R^{2/3}} + \frac{(H\sigma^2 B^4)^{1/3}}{K^{1/3}R^{2/3}} + F(\boldsymbol{u}) - F_*, \tag{307}$$

and rearranging yields the desired result. $\qquad\square$

### C.4 PROOF OF THEOREM 6

For this section, we use Assumptions 1, 2(b), and 3(b). Although our analysis follows a similar technique as that of (Koloskova et al., 2020), our proof is significantly simpler because we only consider a fixed communication structure, where (Koloskova et al., 2020) allows for general communication structures between clients.

**Lemma 19.** *For every $\boldsymbol{u} \in \mathbb{R}^d, t \geq 0$ and $m \in [M]$:*

$$\mathbb{E}_{\xi_t^m}\left[\|\nabla f(\boldsymbol{w}_t^m; \xi_t^m) - \nabla F_m(\boldsymbol{w}_t^m)\|^2\right]$$
$$\leq 3\sigma^2(\boldsymbol{u}) + 3H^2\|\boldsymbol{w}_t^m - \bar{\boldsymbol{w}}_t\|^2 + 6H(F_m(\bar{\boldsymbol{w}}_t) - F_m(\boldsymbol{u})) - 6H\langle\bar{\boldsymbol{w}}_t - \boldsymbol{u}, \nabla F_m(\boldsymbol{u})\rangle. \tag{308}$$

*Proof.* We can decompose:

$$\nabla f(\boldsymbol{w}_t^m; \xi_t^m) - \nabla F_m(\boldsymbol{w}_t^m) = (\nabla f(\boldsymbol{w}_t^m; \xi_t^m) - \nabla f(\bar{\boldsymbol{w}}_t; \xi_t^m) - \nabla F_m(\boldsymbol{w}_t^m) + \nabla F_m(\bar{\boldsymbol{w}}_t)) \tag{309}$$

$$+ (\nabla f(\bar{\boldsymbol{w}}_t; \xi_t^m) - \nabla f(\boldsymbol{u}; \xi_t^m) - \nabla F_m(\bar{\boldsymbol{w}}_t) + \nabla F_m(\boldsymbol{u})) \tag{310}$$

$$+ (\nabla f(\boldsymbol{u}; \xi_t^m) - \nabla F_m(\boldsymbol{u})), \tag{311}$$

so

$$\mathbb{E}_{\xi_t^m}\left[\|\nabla f(\boldsymbol{w}_t^m;\xi_t^m) - \nabla F_m(\boldsymbol{w}_t^m)\|^2\right] \tag{312}$$

$$\leq 3\mathbb{E}_{\xi_t^m}\left[\|\nabla f(\boldsymbol{w}_t^m;\xi_t^m) - \nabla f(\bar{\boldsymbol{w}}_t;\xi_t^m) - \nabla F_m(\boldsymbol{w}_t^m) + \nabla F_m(\bar{\boldsymbol{w}}_t)\|^2\right] \tag{313}$$

$$+ 3\mathbb{E}_{\xi_t^m}\left[\|\nabla f(\bar{\boldsymbol{w}}_t;\xi_t^m) - \nabla f(\boldsymbol{u};\xi_t^m) - \nabla F_m(\bar{\boldsymbol{w}}_t) + \nabla F_m(\boldsymbol{u})\|^2\right] \tag{314}$$

$$+ 3\mathbb{E}_{\xi_t^m}\left[\|\nabla f(\boldsymbol{u};\xi_t^m) - \nabla F_m(\boldsymbol{u})\|^2\right] \tag{315}$$

$$\overset{(i)}{\leq} 3\mathbb{E}_{\xi_t^m}\left[\|\nabla f(\boldsymbol{w}_t^m;\xi_t^m) - \nabla f(\bar{\boldsymbol{w}}_t;\xi_t^m)\|^2\right] + 3\mathbb{E}_{\xi_t^m}\left[\|\nabla f(\bar{\boldsymbol{w}}_t;\xi_t^m) - \nabla f(\boldsymbol{u};\xi_t^m)\|^2\right] \tag{316}$$

$$+ 3\mathbb{E}_{\xi_t^m}\left[\|\nabla f(\boldsymbol{u};\xi_t^m) - \nabla F_m(\boldsymbol{u})\|^2\right] \tag{317}$$

$$\overset{(ii)}{\leq} 3H^2\mathbb{E}_{\xi_t^m}\left[\|\boldsymbol{w}_t^m - \bar{\boldsymbol{w}}_t\|^2\right] \tag{318}$$

$$+ 6H\mathbb{E}_{\xi_t^m}[f(\bar{\boldsymbol{w}}_t;\xi_t^m) - f(\boldsymbol{u};\xi_t^m) - \langle\bar{\boldsymbol{w}}_t - \boldsymbol{u}, \nabla f(\boldsymbol{u};\xi_t^m)\rangle] + 3\sigma^2(\boldsymbol{u}) \tag{319}$$

$$= 3H^2\|\boldsymbol{w}_t^m - \bar{\boldsymbol{w}}_t\|^2 + 6H(F_m(\bar{\boldsymbol{w}}_t) - F_m(\boldsymbol{u})) - 6H\langle\bar{\boldsymbol{w}}_t - \boldsymbol{u}, \nabla F_m(\boldsymbol{u})\rangle + 3\sigma^2(\boldsymbol{u}), \tag{320}$$

where $(i)$ uses the fact that $\mathbb{E}\left[\|X - \mathbb{E}[X]\|^2\right] \leq \mathbb{E}\left[\|X\|^2\right]$, and $(ii)$ uses the fact that $f(\cdot,\xi_t^m)$ is smooth and convex together with Lemma 35. $\qquad\square$

**Lemma 20.** *For any $\boldsymbol{u} \in \mathbb{R}^d$,*

$$\frac{1}{M}\sum_{m=1}^M \mathbb{E}\left[\|\nabla F_m(\boldsymbol{w}_t^m) - \nabla F(\boldsymbol{w}_t)\|^2\right]$$

$$\leq \frac{10H^2}{M}\sum_{m=1}^M \mathbb{E}\left[\|\boldsymbol{w}_t^m - \bar{\boldsymbol{w}}_t\|^2\right] + 5\zeta^2(\boldsymbol{u}) + 10H(F(\bar{\boldsymbol{w}}_t) - F(\boldsymbol{u})) - 20H\mathbb{E}[\langle\bar{\boldsymbol{w}}_t - \boldsymbol{u}, \nabla F(\boldsymbol{u})\rangle]. \tag{321}$$

*Proof.* For any $m \in [M]$, we decompose $\nabla F_m(\boldsymbol{w}_t^m) - \nabla F(\boldsymbol{w}_t^m)$ as:

$$\nabla F_m(\boldsymbol{w}_t^m) - \nabla F(\boldsymbol{w}_t^m) \tag{322}$$

$$= (\nabla F_m(\boldsymbol{w}_t^m) - \nabla F_m(\bar{\boldsymbol{w}}_t)) + (\nabla F_m(\bar{\boldsymbol{w}}_t) - \nabla F_m(\boldsymbol{u})) + (\nabla F_m(\boldsymbol{u}) - \nabla F(\boldsymbol{u})) + \tag{323}$$

$$(\nabla F(\boldsymbol{u}) - \nabla F(\bar{\boldsymbol{w}}_t)) + (\nabla F(\bar{\boldsymbol{w}}_t) - \nabla F(\boldsymbol{w}_t^m)). \tag{324}$$

Then

$$\|\nabla F_m(\boldsymbol{w}_t^m) - \nabla F(\boldsymbol{w}_t^m)\|^2 \tag{325}$$

$$\leq 5\|\nabla F_m(\boldsymbol{w}_t^m) - \nabla F_m(\bar{\boldsymbol{w}}_t)\|^2 + 5\|\nabla F_m(\bar{\boldsymbol{w}}_t) - \nabla F_m(\boldsymbol{u})\|^2 + 5\|\nabla F_m(\boldsymbol{u}) - \nabla F(\boldsymbol{u})\|^2 \tag{326}$$

$$+ 5\|\nabla F(\boldsymbol{u}) - \nabla F(\bar{\boldsymbol{w}}_t)\|^2 + 5\|\nabla F(\bar{\boldsymbol{w}}_t) - \nabla F(\boldsymbol{w}_t^m)\|^2 \tag{327}$$

$$\overset{(i)}{\leq} 10H^2\|\boldsymbol{w}_t^m - \bar{\boldsymbol{w}}_t\|^2 + 5\|\nabla F_m(\bar{\boldsymbol{w}}_t) - \nabla F_m(\boldsymbol{u})\|^2 + 5\|\nabla F_m(\boldsymbol{u}) - \nabla F(\boldsymbol{u})\|^2 \tag{328}$$

$$+ 5\|\nabla F(\boldsymbol{u}) - \nabla F(\bar{\boldsymbol{w}}_t)\|^2 \tag{329}$$

$$\overset{(ii)}{\leq} 10H^2\|\boldsymbol{w}_t^m - \bar{\boldsymbol{w}}_t\|^2 + 5H(F_m(\bar{\boldsymbol{w}}_t) - F_m(\boldsymbol{u}) - 2H\langle\bar{\boldsymbol{w}}_t - \boldsymbol{u}, \nabla F_m(\boldsymbol{u})\rangle) \tag{330}$$

$$+ 5\|\nabla F_m(\boldsymbol{u}) - \nabla F(\boldsymbol{u})\|^2 + 5H(F(\bar{\boldsymbol{w}}_t) - F(\boldsymbol{u}) - 2H\langle\bar{\boldsymbol{w}}_t - \boldsymbol{u}, \nabla F(\boldsymbol{u})\rangle), \tag{331}$$

where $(i)$ uses smoothness of $F_m$ and $F$, and $(ii)$ uses Lemma 35. Taking expectation and averaging over $m \in [M]$:

$$\frac{1}{M} \sum_{m=1}^{M} \mathbb{E}\left[\|\nabla F_m(\boldsymbol{w}_t^m) - \nabla F(\boldsymbol{w}_t^m)\|^2\right] \tag{332}$$

$$\leq \frac{10H^2}{M} \sum_{m=1}^{M} \mathbb{E}\left[\|\boldsymbol{w}_t^m - \bar{\boldsymbol{w}}_t\|^2\right] + 10H\mathbb{E}[F(\bar{\boldsymbol{w}}_t) - F(\boldsymbol{u})] - 20H\mathbb{E}\left[\langle \bar{\boldsymbol{w}}_t - \boldsymbol{u}, \nabla F(\boldsymbol{u})\rangle\right] \tag{333}$$

$$+ \frac{5}{M} \sum_{m=1}^{M} \mathbb{E}\left[\|\nabla F_m(\boldsymbol{u}) - \nabla F(\boldsymbol{u})\|^2\right] \tag{334}$$

$$\leq \frac{10H^2}{M} \sum_{m=1}^{M} \mathbb{E}\left[\|\boldsymbol{w}_t^m - \bar{\boldsymbol{w}}_t\|^2\right] + 10H\mathbb{E}[F(\bar{\boldsymbol{w}}_t) - F(\boldsymbol{u})] \tag{335}$$

$$- 20H\mathbb{E}\left[\langle \bar{\boldsymbol{w}}_t - \boldsymbol{u}, \nabla F(\boldsymbol{u})\rangle\right] + 5\zeta^2(\boldsymbol{u}) \tag{336}$$

$\square$

**Lemma 21.** *If $\eta \leq 1/(4H)$, then for any $\boldsymbol{u} \in \mathbb{R}^d$,*

$$\mathbb{E}\left[\|\bar{\boldsymbol{w}}_{t+1} - \boldsymbol{u}\|^2\right] \leq \mathbb{E}\left[\|\bar{\boldsymbol{w}}_t - \boldsymbol{u}\|^2\right] + \frac{3\eta^2\sigma^2(\boldsymbol{u})}{M} + \frac{2\eta H}{M} \sum_{m=1}^{M} \mathbb{E}\left[\|\boldsymbol{w}_t^m - \bar{\boldsymbol{w}}_t\|^2\right] \tag{337}$$

$$- \eta\mathbb{E}[F(\bar{\boldsymbol{w}}_t) - F_*] + 2\eta(F(\boldsymbol{u}) - F_*) - \frac{6\eta^2 H}{M}\mathbb{E}[\langle \bar{\boldsymbol{w}}_t - \boldsymbol{u}, \nabla F(\boldsymbol{u})\rangle]. \tag{338}$$

*Proof.*

$$\|\bar{\boldsymbol{w}}_{t+1} - \boldsymbol{u}\|^2 = \left\|\bar{\boldsymbol{w}}_t - \boldsymbol{u} - \frac{\eta}{M} \sum_{m=1}^{M} \boldsymbol{g}_t^m\right\|^2 \tag{339}$$

$$= \left\|\bar{\boldsymbol{w}}_t - \boldsymbol{u} - \frac{\eta}{M} \sum_{m=1}^{M} \nabla F_m(\boldsymbol{w}_t^m) - \left(\frac{\eta}{M} \sum_{m=1}^{M} \boldsymbol{g}_t^m - \nabla F_m(\boldsymbol{w}_t^m)\right)\right\|^2. \tag{340}$$

Taking conditional expectation $\mathbb{E}_t[\cdot] := \mathbb{E}[\cdot|\{\xi_s^m : s < t, m \in [M]\}]$:

$$\mathbb{E}_t\left[\|\bar{\boldsymbol{w}}_{t+1} - \boldsymbol{u}\|^2\right] = \left\|\bar{\boldsymbol{w}}_t - \boldsymbol{u} - \frac{\eta}{M} \sum_{m=1}^{M} \nabla F_m(\boldsymbol{w}_t^m)\right\|^2 + \mathbb{E}_t\left[\left\|\frac{\eta}{M} \sum_{m=1}^{M} \boldsymbol{g}_t^m - \nabla F_m(\boldsymbol{w}_t^m)\right\|^2\right] \tag{341}$$

$$= \left\|\bar{\boldsymbol{w}}_t - \boldsymbol{u} - \frac{\eta}{M} \sum_{m=1}^{M} \nabla F_m(\boldsymbol{w}_t^m)\right\|^2 + \frac{\eta^2}{M^2} \sum_{m=1}^{M} \mathbb{E}_t\left[\|\boldsymbol{g}_t^m - \nabla F_m(\boldsymbol{w}_t^m)\|^2\right] \tag{342}$$

$$\overset{(i)}{\leq} \left\|\bar{\boldsymbol{w}}_t - \boldsymbol{u} - \frac{\eta}{M} \sum_{m=1}^{M} \nabla F_m(\boldsymbol{w}_t^m)\right\|^2 + \frac{\eta^2}{M^2} \sum_{m=1}^{M} \left(3\sigma^2(\boldsymbol{u}) \right. \tag{343}$$

$$\left. + 3H^2 \|\boldsymbol{w}_t^m - \bar{\boldsymbol{w}}_t\|^2 + 6H(F_m(\bar{\boldsymbol{w}}_t) - F_m(\boldsymbol{u})) - 6H\langle \bar{\boldsymbol{w}}_t - \boldsymbol{u}, \nabla F_m(\boldsymbol{u})\rangle\right) \tag{344}$$

$$= \underbrace{\left\|\bar{\boldsymbol{w}}_t - \boldsymbol{u} - \frac{\eta}{M} \sum_{m=1}^{M} \nabla F_m(\boldsymbol{w}_t^m)\right\|^2}_{A} + \frac{3\eta^2\sigma^2(\boldsymbol{u})}{M} + \frac{3\eta^2 H^2}{M^2} \sum_{m=1}^{M} \|\boldsymbol{w}_t^m - \bar{\boldsymbol{w}}_t\|^2 \tag{345}$$

$$+ \frac{18\eta^2 H}{M}(F(\bar{\boldsymbol{w}}_t) - F(\boldsymbol{u})) - \frac{6\eta^2 H}{M}\langle \bar{\boldsymbol{w}}_t - \boldsymbol{u}, \nabla F(\boldsymbol{u})\rangle, \tag{346}$$

where $(i)$ uses Lemma 19. To bound $A$, we decompose

$$A = \|\bar{\boldsymbol{w}}_t - \boldsymbol{u}\|^2 + \frac{\eta^2}{M^2}\underbrace{\left\|\sum_{m=1}^{M}\nabla F_m(\boldsymbol{w}_t^m)\right\|^2}_{B_1} + \frac{2\eta}{M}\underbrace{\left\langle \bar{\boldsymbol{w}}_t - \boldsymbol{u}, -\sum_{m=1}^{M}\nabla F_m(\boldsymbol{w}_t^m)\right\rangle}_{B_2}. \tag{347}$$

We can bound $B_1$ and $B_2$ separately:

$$B_1 = \left\|\sum_{m=1}^{M}\nabla F_m(\bar{\boldsymbol{w}}_t) + \sum_{m=1}^{M}\left(\nabla F_m(\boldsymbol{w}_t^m) - \nabla F_m(\bar{\boldsymbol{w}}_t)\right)\right\|^2 \tag{348}$$

$$\leq 2\left\|\sum_{m=1}^{M}\nabla F_m(\bar{\boldsymbol{w}}_t)\right\|^2 + 2\left\|\sum_{m=1}^{M}\nabla F_m(\boldsymbol{w}_t^m) - \nabla F_m(\bar{\boldsymbol{w}}_t)\right\|^2 \tag{349}$$

$$\leq 2M^2\|\nabla F(\bar{\boldsymbol{w}}_t)\|^2 + 2M\sum_{m=1}^{M}\|\nabla F_m(\boldsymbol{w}_t^m) - \nabla F_m(\bar{\boldsymbol{w}}_t)\|^2 \tag{350}$$

$$\leq 4HM^2(F(\bar{\boldsymbol{w}}_t) - F_*) + 2H^2 M\sum_{m=1}^{M}\|\boldsymbol{w}_t^m - \bar{\boldsymbol{w}}_t\|^2, \tag{351}$$

and

$$B_2 = -\sum_{m=1}^{M}\langle \bar{\boldsymbol{w}}_t - \boldsymbol{u}, \nabla F_m(\boldsymbol{w}_t^m)\rangle \tag{352}$$

$$= \sum_{m=1}^{M}\langle \boldsymbol{u} - \boldsymbol{w}_t^m, \nabla F_m(\boldsymbol{w}_t^m)\rangle - \sum_{m=1}^{M}\langle \bar{\boldsymbol{w}}_t - \boldsymbol{w}_t^m, \nabla F_m(\boldsymbol{w}_t^m)\rangle \tag{353}$$

$$\stackrel{(i)}{=} \sum_{m=1}^{M}(F(\boldsymbol{u}) - F(\boldsymbol{w}_t^m)) - \sum_{m=1}^{M}\left(F(\bar{\boldsymbol{w}}_t) - F(\boldsymbol{w}_t^m) - \frac{H}{2}\|\bar{\boldsymbol{w}}_t - \boldsymbol{w}_t^m\|^2\right) \tag{354}$$

$$= -M(F(\bar{\boldsymbol{w}}_t) - F(\boldsymbol{u})) + \frac{H}{2}\sum_{m=1}^{M}\|\bar{\boldsymbol{w}}_t - \boldsymbol{w}_t^m\|^2, \tag{355}$$

where $(i)$ uses convexity and smoothness of $F$.

Plugging the resulting bound of $A$ back into Equation 346 yields

$$\mathbb{E}_t\left[\|\bar{\boldsymbol{w}}_{t+1} - \boldsymbol{u}\|^2\right] \tag{356}$$

$$\leq \|\bar{\boldsymbol{w}}_t - \boldsymbol{u}\|^2 + \frac{\eta^2}{M^2}\left(4HM^2(F(\bar{\boldsymbol{w}}_t) - F_*) + 2H^2 M\sum_{m=1}^{M}\|\boldsymbol{w}_t^m - \bar{\boldsymbol{w}}_t\|^2\right) \tag{357}$$

$$+ \frac{2\eta}{M}\left(-M(F(\bar{\boldsymbol{w}}_t) - F(\boldsymbol{u})) + \frac{H}{2}\sum_{m=1}^{M}\|\bar{\boldsymbol{w}}_t - \boldsymbol{w}_t^m\|^2\right) + \frac{3\eta^2\sigma^2(\boldsymbol{u})}{M} \tag{358}$$

$$+ \frac{3\eta^2 H^2}{M^2}\sum_{m=1}^{M}\|\boldsymbol{w}_t^m - \bar{\boldsymbol{w}}_t\|^2 + \frac{18\eta^2 H}{M}(F(\bar{\boldsymbol{w}}_t) - F(\boldsymbol{u})) - \frac{6\eta^2 H}{M}\langle \bar{\boldsymbol{w}}_t - \boldsymbol{u}, \nabla F(\boldsymbol{u})\rangle \tag{359}$$

$$\leq \|\bar{\boldsymbol{w}}_t - \boldsymbol{u}\|^2 + \frac{3\eta^2\sigma^2(\boldsymbol{u})}{M} + \left(2\eta^2 H^2 + \eta H + \frac{3\eta^2 H^2}{M}\right)\frac{1}{M}\sum_{m=1}^{M}\|\boldsymbol{w}_t^m - \bar{\boldsymbol{w}}_t\|^2 \tag{360}$$

$$- \left(2\eta - 4\eta^2 H - \frac{18\eta^2 H}{M}\right)(F(\bar{\boldsymbol{w}}_t) - F_*) + 2\eta(F(\boldsymbol{u}) - F_*) - \frac{6\eta^2 H}{M}\langle \bar{\boldsymbol{w}}_t - \boldsymbol{u}, \nabla F(\boldsymbol{u})\rangle \tag{361}$$

$$\overset{(i)}{\leq} \|\bar{\boldsymbol{w}}_t - \boldsymbol{u}\|^2 + \frac{3\eta^2\sigma^2(\boldsymbol{u})}{M} + \frac{2\eta H}{M} \sum_{m=1}^{M} \|\boldsymbol{w}_t^m - \bar{\boldsymbol{w}}_t\|^2 - \eta(F(\bar{\boldsymbol{w}}_t) - F_*) + 2\eta(F(\boldsymbol{u}) - F_*)$$

(362)

$$- \frac{6\eta^2 H}{M} \langle \bar{\boldsymbol{w}}_t - \boldsymbol{u}, \nabla F(\boldsymbol{u}) \rangle,$$

(363)

where $(i)$ uses the condition $\eta \leq 1/(22KH)$. Taking total expectation yields

$$\mathbb{E}\left[\|\bar{\boldsymbol{w}}_{t+1} - \boldsymbol{u}\|^2\right] \leq \mathbb{E}\left[\|\bar{\boldsymbol{w}}_t - \boldsymbol{u}\|^2\right] + \frac{3\eta^2\sigma^2(\boldsymbol{u})}{M} + \frac{2\eta H}{M} \sum_{m=1}^{M} \mathbb{E}\left[\|\boldsymbol{w}_t^m - \bar{\boldsymbol{w}}_t\|^2\right]$$

(364)

$$- \eta\mathbb{E}[F(\bar{\boldsymbol{w}}_t) - F_*] + 2\eta(F(\boldsymbol{u}) - F_*) - \frac{6\eta^2 H}{M} \mathbb{E}[\langle \bar{\boldsymbol{w}}_t - \boldsymbol{u}, \nabla F(\boldsymbol{u}) \rangle].$$

(365)

$\square$

**Lemma 22.** *For any $\eta > 0$,*

$$\frac{1}{M} \sum_{m=1}^{M} \mathbb{E}\left[\|\boldsymbol{w}_t^m - \bar{\boldsymbol{w}}_t\|^2\right] \leq 18\eta^2 K\sigma^2(\boldsymbol{u}) + 120\eta^2 K^2\zeta^2(\boldsymbol{u}) + 276\eta^2 KH \sum_{i=t_0}^{t-1} \mathbb{E}[F(\bar{\boldsymbol{w}}_i) - F(\boldsymbol{u})]$$

(366)

$$- 516\eta^2 KH \sum_{i=t_0}^{t-1} \mathbb{E}[\langle \bar{\boldsymbol{w}}_i - \boldsymbol{u}, \nabla F(\boldsymbol{u}) \rangle].$$

(367)

*Proof.* The proof of this Lemma is similar to that of Lemma 8 from Woodworth et al. (2020b), but is modified to use a general comparator $\boldsymbol{u}$ instead of a global minimum $\boldsymbol{w}_*$, and to use a local noise assumption instead of a global one (i.e. $\sigma(\boldsymbol{u})$ instead of $\sigma$).

$$\frac{1}{M} \sum_{m=1}^{M} \mathbb{E}\left[\|\boldsymbol{w}_t^m - \bar{\boldsymbol{w}}_t\|^2\right] = \frac{1}{M} \sum_{m=1}^{M} \mathbb{E}\left[\left\|\frac{1}{M} \sum_{n=1}^{M} (\boldsymbol{w}_t^m - \boldsymbol{w}_t^n)\right\|^2\right]$$

(368)

$$\leq \underbrace{\frac{1}{M^2} \sum_{m=1}^{M} \sum_{n=1}^{M} \mathbb{E}\left[\|\boldsymbol{w}_t^m - \boldsymbol{w}_t^n\|^2\right]}_{R_t}.$$

(369)

We can then establish a recursion over $R_t$ as follows:

$$R_t = \frac{1}{M^2} \sum_{m,n\in[M]} \mathbb{E}\left[\|\boldsymbol{w}_{t-1}^m - \eta\boldsymbol{g}_{t-1}^m - (\boldsymbol{w}_{t-1}^n - \eta\boldsymbol{g}_{t-1}^n)\|^2\right]$$

(370)

$$= \frac{1}{M^2} \sum_{m,n\in[M]} \mathbb{E}\Big[\big\|\boldsymbol{w}_{t-1}^m - \boldsymbol{w}_{t-1}^n - \eta\nabla F_m(\boldsymbol{w}_{t-1}^m) + \eta\nabla F_n(\boldsymbol{w}_{t-1}^n)$$

(371)

$$+ \eta(\boldsymbol{g}_{t-1}^m - \nabla F_m(\boldsymbol{w}_{t-1}^m)) - \eta(\boldsymbol{g}_{t-1}^n - \nabla F_n(\boldsymbol{w}_{t-1}^m))\big\|^2\Big]$$

(372)

$$\overset{(i)}{=} \frac{1}{M^2} \sum_{m,n\in[M]} \mathbb{E}\left[\|\boldsymbol{w}_{t-1}^m - \boldsymbol{w}_{t-1}^n - \eta\nabla F_m(\boldsymbol{w}_{t-1}^m) + \eta\nabla F_n(\boldsymbol{w}_{t-1}^n)\|^2\right]$$

(373)

$$+ \frac{\eta^2}{M^2} \sum_{m,n\in[M]} \left(\mathbb{E}\left[\|\boldsymbol{g}_{t-1}^m - \nabla F_m(\boldsymbol{w}_{t-1}^m)\|^2\right] + \mathbb{E}\left[\|\boldsymbol{g}_{t-1}^n - \nabla F_n(\boldsymbol{w}_{t-1}^m)\|^2\right]\right)$$

(374)

$$\overset{(ii)}{\leq} \frac{1}{M^2} \sum_{m,n\in[M]} \mathbb{E}\left[\|\boldsymbol{w}_{t-1}^m - \boldsymbol{w}_{t-1}^n - \eta\nabla F_m(\boldsymbol{w}_{t-1}^m) + \eta\nabla F_n(\boldsymbol{w}_{t-1}^n)\|^2\right]$$

(375)

$$+ 2\eta^2 \left( 3\sigma^2(\boldsymbol{u}) + \frac{3H^2}{M} \sum_{m=1}^{M} \mathbb{E}\left[ \left\| \boldsymbol{w}_{t-1}^m - \bar{\boldsymbol{w}}_{t-1} \right\|^2 \right] + 6H\mathbb{E}[F(\bar{\boldsymbol{w}}_{t-1}) - F(\boldsymbol{u})] \right. \tag{376}$$

$$\left. - 6H\mathbb{E}[\langle \bar{\boldsymbol{w}}_{t-1} - \boldsymbol{u}, \nabla F(\boldsymbol{u}) \rangle] \right) \tag{377}$$

$$\overset{(iii)}{\leq} \left( 1 + \frac{1}{\gamma} \right) \frac{1}{M^2} \sum_{m,n \in [M]} \mathbb{E}\left[ \left\| \boldsymbol{w}_{t-1}^m - \boldsymbol{w}_{t-1}^n - \eta \nabla F(\boldsymbol{w}_{t-1}^m) + \eta \nabla F(\boldsymbol{w}_{t-1}^n) \right\|^2 \right] \tag{378}$$

$$+ \frac{(1+\gamma)\eta^2}{M^2} \sum_{m,n \in [M]} \mathbb{E}\left[ \left\| -(\nabla F_m(\boldsymbol{w}_{t-1}^m) - \nabla F(\boldsymbol{w}_{t-1}^m)) + (\nabla F_n(\boldsymbol{w}_{t-1}^n) - \nabla F(\boldsymbol{w}_{t-1}^n)) \right\|^2 \right] \tag{379}$$

$$+ 6\eta^2\sigma^2(\boldsymbol{u}) + 6\eta^2 H^2 R_{t-1} + 12\eta^2 H\mathbb{E}[F(\bar{\boldsymbol{w}}_{t-1}) - F(\boldsymbol{u})] - 12\eta^2 H\mathbb{E}[\langle \bar{\boldsymbol{w}}_{t-1} - \boldsymbol{u}, \nabla F(\boldsymbol{u}) \rangle] \tag{380}$$

$$\overset{(iv)}{\leq} \left( 1 + \frac{1}{\gamma} \right) \frac{1}{M^2} \sum_{m,n \in [M]} \mathbb{E}\left[ \left\| \boldsymbol{w}_{t-1}^m - \boldsymbol{w}_{t-1}^n \right\|^2 \right] \tag{381}$$

$$+ \frac{2(1+\gamma)\eta^2}{M^2} \sum_{m,n \in [M]} \left( \mathbb{E}\left[ \left\| \nabla F_m(\boldsymbol{w}_{t-1}^m) - \nabla F(\boldsymbol{w}_{t-1}^m) \right\|^2 \right] \right. \tag{382}$$

$$\left. + \mathbb{E}\left[ \left\| \nabla F_n(\boldsymbol{w}_{t-1}^n) - \nabla F(\boldsymbol{w}_{t-1}^n) \right\|^2 \right] \right) \tag{383}$$

$$+ 6\eta^2\sigma^2(\boldsymbol{u}) + 6\eta^2 H^2 R_{t-1} + 12\eta^2 H\mathbb{E}[F(\bar{\boldsymbol{w}}_{t-1}) - F(\boldsymbol{u})] - 12\eta^2 H\mathbb{E}[\langle \bar{\boldsymbol{w}}_{t-1} - \boldsymbol{u}, \nabla F(\boldsymbol{u}) \rangle] \tag{384}$$

$$\leq \left( 1 + \frac{1}{\gamma} + 6\eta^2 H^2 \right) R_{t-1} + (1+\gamma)\frac{4\eta^2}{M} \sum_{m=1}^{M} \mathbb{E}\left[ \left\| \nabla F_m(\boldsymbol{w}_{t-1}^m) - \nabla F(\boldsymbol{w}_{t-1}^m) \right\|^2 \right] \tag{385}$$

$$+ 6\eta^2\sigma^2(\boldsymbol{u}) + 12\eta^2 H\mathbb{E}[F(\bar{\boldsymbol{w}}_{t-1}) - F(\boldsymbol{u})] - 12\eta^2 H\mathbb{E}[\langle \bar{\boldsymbol{w}}_{t-1} - \boldsymbol{u}, \nabla F(\boldsymbol{u}) \rangle], \tag{386}$$

where $(i)$ uses the fact that $\boldsymbol{g}_{t-1}^m - \nabla F_m(\boldsymbol{w}_t^m)$ has zero mean and is conditionally independent (given $\boldsymbol{w}_{t-1}^m$) of $\boldsymbol{w}_{t-1}^m - \boldsymbol{w}_{t-1}^n - \eta\nabla F_m(\boldsymbol{w}_{t-1}^m) + \eta\nabla F_n(\boldsymbol{w}_{t-1}^n)$, $(ii)$ uses Lemma 19, $(iii)$ uses Young's inequality with arbitrary $\gamma > 0$, and $(iv)$ uses Lemma 36 together with $\|\boldsymbol{a} + \boldsymbol{b}\|^2 \leq 2\|\boldsymbol{a}\|^2 + 2\|\boldsymbol{b}\|^2$. Finally, the heterogeneity term involving $\|\nabla F_m(\boldsymbol{w}_t^m) - \nabla F(\boldsymbol{w}_t^m)\|^2$ can be bounded with Lemma 20, which yields

$$R_t \leq \left( 1 + \frac{1}{\gamma} + 6\eta^2 H^2 \right) R_{t-1} + (1+\gamma)\,4\eta^2 \left( \frac{10H^2}{M} \sum_{m=1}^{M} \mathbb{E}\left[ \left\| \boldsymbol{w}_{t-1}^m - \bar{\boldsymbol{w}}_{t-1} \right\|^2 \right] \right. \tag{387}$$

$$\left. + 5\zeta^2(\boldsymbol{u}) + 10H(F(\bar{\boldsymbol{w}}_{t-1}) - F(\boldsymbol{u})) - 20H\mathbb{E}[\langle \bar{\boldsymbol{w}}_{t-1} - \boldsymbol{u}, \nabla F(\boldsymbol{u}) \rangle] \right) \tag{388}$$

$$+ 6\eta^2\sigma^2(\boldsymbol{u}) + 12\eta^2 H\mathbb{E}[F(\bar{\boldsymbol{w}}_{t-1}) - F(\boldsymbol{u})] - 12\eta^2 H\mathbb{E}[\langle \bar{\boldsymbol{w}}_{t-1} - \boldsymbol{u}, \nabla F(\boldsymbol{u}) \rangle] \tag{389}$$

$$\leq \left( 1 + \frac{1}{\gamma} + 6\eta^2 H^2 + 40(1+\gamma)\eta^2 H^2 \right) R_{t-1} + 6\eta^2\sigma^2(\boldsymbol{u}) + 20(1+\gamma)\eta^2\zeta^2(\boldsymbol{u}) \tag{390}$$

$$+ \left( 12\eta^2 H + 40(1+\gamma)\eta^2 H \right) \mathbb{E}[F(\bar{\boldsymbol{w}}_{t-1}) - F(\boldsymbol{u})] \tag{391}$$

$$- \left( 12\eta^2 H + 80\,(1+\gamma)\,\eta^2 H \right) \mathbb{E}[\langle \bar{\boldsymbol{w}}_{t-1} - \boldsymbol{u}, \nabla F(\boldsymbol{u}) \rangle]. \tag{392}$$

Now we use the choice $\gamma = 2(K-1)$, so

$$R_t \leq \left( 1 + \frac{1}{2(K-1)} + 6\eta^2 H^2 + 80\eta^2 K H^2 \right) R_{t-1} + 6\eta^2\sigma^2(\boldsymbol{u}) + 40\eta^2 K\zeta^2(\boldsymbol{u}) \tag{393}$$

$$+ \left( 12\eta^2 H + 80\eta^2 K H \right) \mathbb{E}[F(\bar{\boldsymbol{w}}_{t-1}) - F(\boldsymbol{u})] - 172\eta^2 H\mathbb{E}[\langle \bar{\boldsymbol{w}}_{t-1} - \boldsymbol{u}, \nabla F(\boldsymbol{u}) \rangle] \tag{394}$$

$$\overset{(i)}{\leq} \left( 1 + \frac{1}{K-1} \right) R_{t-1} + 6\eta^2\sigma^2(\boldsymbol{u}) + 40\eta^2 K\zeta^2(\boldsymbol{u}) + 92\eta^2 K H\mathbb{E}[F(\bar{\boldsymbol{w}}_{t-1}) - F(\boldsymbol{u})] \tag{395}$$

$$- 172\eta^2 K H\mathbb{E}[\langle \bar{\boldsymbol{w}}_{t-1} - \boldsymbol{u}, \nabla F(\boldsymbol{u}) \rangle], \tag{396}$$

where $(i)$ uses the condition $\eta \leq 1/(14KH)$.

Now, we can unroll this recurrence from $t$ to $t_0$, where $t_0 = K\lfloor t/K \rfloor$ is the last synchronization timestep before $t$. Notice that $R_{t_0} = 0$. So

$$R_t \leq \left(1 + \frac{1}{K-1}\right)^{t-t_0} R_{t_0} + \sum_{i=t_0}^{t-1} \left(1 + \frac{1}{K-1}\right)^{t-1-i} \left(6\eta^2\sigma^2(\boldsymbol{u}) + 40\eta^2 K\zeta^2(\boldsymbol{u})\right. \tag{397}$$

$$\left. + 92\eta^2 KH\mathbb{E}[F(\bar{\boldsymbol{w}}_i) - F(\boldsymbol{u})] - 172\eta^2 KH\mathbb{E}[\langle \bar{\boldsymbol{w}}_i - \boldsymbol{u}, \nabla F(\boldsymbol{u})\rangle]\right) \tag{398}$$

$$\leq \left(1 + \frac{1}{K-1}\right)^{t-1-t_0} \sum_{i=t_0}^{t-1} \left(6\eta^2\sigma^2(\boldsymbol{u}) + 40\eta^2 K\zeta^2(\boldsymbol{u}) + 92\eta^2 KH\mathbb{E}[F(\bar{\boldsymbol{w}}_i) - F_*]\right. \tag{399}$$

$$\left. - 172\eta^2 KH\mathbb{E}[\langle \bar{\boldsymbol{w}}_i - \boldsymbol{u}, \nabla F(\boldsymbol{u})\rangle]\right) \tag{400}$$

$$\leq \left(1 + \frac{1}{K-1}\right)^{K-1} \left(6(t-t_0)\eta^2\sigma^2(\boldsymbol{u}) + 40(t-t_0)\eta^2 K\zeta^2(\boldsymbol{u})\right. \tag{401}$$

$$\left. + 92\eta^2 KH\sum_{i=t_0}^{t-1}\mathbb{E}[F(\bar{\boldsymbol{w}}_i) - F(\boldsymbol{u})] - 172\eta^2 KH\sum_{i=t_0}^{t-1}\mathbb{E}[\langle \bar{\boldsymbol{w}}_i - \boldsymbol{u}, \nabla F(\boldsymbol{u})\rangle]\right) \tag{402}$$

$$\leq 18\eta^2 K\sigma^2(\boldsymbol{u}) + 120\eta^2 K^2\zeta^2(\boldsymbol{u}) + 276\eta^2 KH\sum_{i=t_0}^{t-1}\mathbb{E}[F(\bar{\boldsymbol{w}}_i) - F(\boldsymbol{u})] \tag{403}$$

$$- 516\eta^2 KH\sum_{i=t_0}^{t-1}\mathbb{E}[\langle \bar{\boldsymbol{w}}_i - \boldsymbol{u}, \nabla F(\boldsymbol{u})\rangle]. \tag{404}$$

$\square$

*Proof of Theorem 6.* Starting from Lemma 21, applying Lemma 22 to bound the drift term yields

$$\mathbb{E}\left[\|\bar{\boldsymbol{w}}_{t+1} - \boldsymbol{u}\|^2\right] \tag{405}$$

$$\leq \mathbb{E}\left[\|\bar{\boldsymbol{w}}_t - \boldsymbol{u}\|^2\right] + \frac{3\eta^2\sigma^2(\boldsymbol{u})}{M} + \frac{2\eta H}{M}\sum_{m=1}^{M}\mathbb{E}\left[\|\boldsymbol{w}_t^m - \bar{\boldsymbol{w}}_t\|^2\right] - \eta\mathbb{E}[F(\bar{\boldsymbol{w}}_t) - F_*] \tag{406}$$

$$+ 2\eta(F(\boldsymbol{u}) - F_*) - \frac{6\eta^2 H}{M}\mathbb{E}[\langle \bar{\boldsymbol{w}}_t - \boldsymbol{u}, \nabla F(\boldsymbol{u})\rangle] \tag{407}$$

$$\leq \mathbb{E}\left[\|\bar{\boldsymbol{w}}_t - \boldsymbol{u}\|^2\right] + \frac{3\eta^2\sigma^2(\boldsymbol{u})}{M} + 2\eta H\left(18\eta^2 K\sigma^2(\boldsymbol{u}) + 120\eta^2 K^2\zeta^2(\boldsymbol{u})\right. \tag{408}$$

$$\left. + 276\eta^2 KH\sum_{i=t_0}^{t-1}\mathbb{E}[F(\bar{\boldsymbol{w}}_i) - F(\boldsymbol{u})] - 516\eta^2 KH\sum_{i=t_0}^{t-1}\mathbb{E}[\langle \bar{\boldsymbol{w}}_i - \boldsymbol{u}, \nabla F(\boldsymbol{u})\rangle]\right) \tag{409}$$

$$- \eta\mathbb{E}[F(\bar{\boldsymbol{w}}_t) - F_*] + 2\eta(F(\boldsymbol{u}) - F_*) - \frac{6\eta^2 H}{M}\mathbb{E}[\langle \bar{\boldsymbol{w}}_t - \boldsymbol{u}, \nabla F(\boldsymbol{u})\rangle] \tag{410}$$

$$\leq \mathbb{E}\left[\|\bar{\boldsymbol{w}}_t - \boldsymbol{u}\|^2\right] - \frac{6\eta^2 H}{M}\mathbb{E}[\langle \bar{\boldsymbol{w}}_t - \boldsymbol{u}, \nabla F(\boldsymbol{u})\rangle] - 1032\eta^3 KH^2\sum_{i=t_0}^{t-1}\mathbb{E}[\langle \bar{\boldsymbol{w}}_i - \boldsymbol{u}, \nabla F(\boldsymbol{u})\rangle] \tag{411}$$

$$+ \frac{3\eta^2\sigma^2(\boldsymbol{u})}{M} + 36\eta^3 HK\sigma^2(\boldsymbol{u}) + 240\eta^3 K^2 H\zeta^2(\boldsymbol{u}) + 2\eta(F(\boldsymbol{u}) - F_*) \tag{412}$$

$$- \eta\mathbb{E}[F(\bar{\boldsymbol{w}}_t) - F_*] + 552\eta^3 KH^2\sum_{i=t_0}^{t-1}\mathbb{E}[F(\bar{\boldsymbol{w}}_i) - F_*]. \tag{413}$$

Each inner product term $\langle \bar{\boldsymbol{w}}_i - \boldsymbol{u}, \nabla F(\boldsymbol{u})\rangle$ can be bounded as:

$$-\langle \bar{\boldsymbol{w}}_i - \boldsymbol{u}, \nabla F(\boldsymbol{u})\rangle \leq \frac{1}{2\lambda}\|\bar{\boldsymbol{w}}_i - \boldsymbol{u}\|^2 + \frac{\lambda}{2}\|\nabla F(\boldsymbol{u})\|^2 \leq \frac{1}{2\lambda}\|\bar{\boldsymbol{w}}_i - \boldsymbol{u}\|^2 + \lambda H(F(\boldsymbol{u}) - F_*), \tag{414}$$

where we will specify $\lambda > 0$ later. So

$$\mathbb{E}\left[\|\bar{\boldsymbol{w}}_{t+1} - \boldsymbol{u}\|^2\right] \tag{415}$$

$$\leq \left(1 + \frac{3\eta^2 H}{\lambda M}\right) \mathbb{E}\left[\|\bar{\boldsymbol{w}}_t - \boldsymbol{u}\|^2\right] + \frac{516\eta^3 KH^2}{\lambda} \sum_{i=t_0}^{t-1} \mathbb{E}[\|\bar{\boldsymbol{w}}_i - \boldsymbol{u}\|^2] \tag{416}$$

$$+ \frac{3\eta^2 \sigma^2(\boldsymbol{u})}{M} + 36\eta^3 HK\sigma^2(\boldsymbol{u}) + 240\eta^3 K^2 H\zeta^2(\boldsymbol{u}) \tag{417}$$

$$+ \left(2\eta + \frac{6\lambda\eta^2 H^2}{M} + 1032\lambda\eta^3 K^2 H^3\right)(F(\boldsymbol{u}) - F_*) \tag{418}$$

$$- \eta\mathbb{E}[F(\bar{\boldsymbol{w}}_t) - F_*] + 552\eta^3 KH^2 \sum_{i=t_0}^{t-1} \mathbb{E}[F(\bar{\boldsymbol{w}}_i) - F_*] \tag{419}$$

$$\overset{(i)}{\leq} \left(1 + \frac{3\eta^2 H}{\lambda M}\right) \mathbb{E}\left[\|\bar{\boldsymbol{w}}_t - \boldsymbol{u}\|^2\right] + \frac{516\eta^3 KH^2}{\lambda} \sum_{i=t_0}^{t-1} \mathbb{E}[\|\bar{\boldsymbol{w}}_i - \boldsymbol{u}\|^2] \tag{420}$$

$$+ \frac{3\eta^2 \sigma^2(\boldsymbol{u})}{M} + 36\eta^3 HK\sigma^2(\boldsymbol{u}) + 240\eta^3 K^2 H\zeta^2(\boldsymbol{u}) + (2 + 6\lambda)\eta(F(\boldsymbol{u}) - F_*) \tag{421}$$

$$- \eta\mathbb{E}[F(\bar{\boldsymbol{w}}_t) - F_*] + 552\eta^3 KH^2 \sum_{i=t_0}^{t-1} \mathbb{E}[F(\bar{\boldsymbol{w}}_i) - F_*], \tag{422}$$

where $(i)$ uses the condition $\eta \leq 1/(14KH)$. Equation 420 is a recursion of the form

$$a_{t+1} \leq ra_t + p\sum_{i=t_0}^{t-1} a_i + b - qc_t + s\sum_{i=t_0}^{t-1} c_i, \tag{423}$$

with

$$a_t = \mathbb{E}\left[\|\bar{\boldsymbol{w}}_t - \boldsymbol{u}\|^2\right], \quad c_t = \mathbb{E}\left[F(\bar{\boldsymbol{w}}_t) - F_*\right] \tag{424}$$

$$r = 1 + \frac{3\eta^2 H}{\lambda M}, \quad p = \frac{516\eta^3 KH^2}{\lambda}, \quad s = 552\eta^3 KH^2, \quad q = \eta \tag{425}$$

$$b = \frac{3\eta^2 \sigma^2(\boldsymbol{u})}{M} + 36\eta^3 HK\sigma^2(\boldsymbol{u}) + 240\eta^3 K^2 H\zeta^2(\boldsymbol{u}) + (2 + 6\lambda)\eta(F(\boldsymbol{u}) - F_*). \tag{426}$$

Letting $\beta = 1 - \frac{1}{KR}$, we multiply Equation 423 by $\beta^t$ and sum over $t \in \{t_0, \ldots, t_0 + K - 1\}$:

$$\sum_{t=t_0}^{t_0+K-1} \beta^t a_{t+1} \tag{427}$$

$$\leq r\sum_{t=t_0}^{t_0+K-1} \beta^t a_t + p\sum_{t=t_0}^{t_0+K-1}\sum_{i=t_0}^{t-1} \beta^t a_i + b\sum_{t=t_0}^{t_0+K-1} \beta^t - q\sum_{t=t_0}^{t_0+K-1} \beta_t c_t + s\sum_{t=t_0}^{t_0+K-1}\sum_{i=t_0}^{t-1} \beta^t c_i \tag{428}$$

$$\leq \sum_{t=t_0}^{t_0+K-1}\left(r\beta^t + p\sum_{i=t_0}^{t_0+K-1} \beta^i\right) a_t + b\sum_{t=t_0}^{t_0+K-1} \beta^t - \sum_{t=t_0}^{t_0+K-1}\left(q\beta_t - s\sum_{i=t_0}^{t_0+K-1} \beta^i\right) c_t. \tag{429}$$

Combining the sums over $\{a_t\}$ and isolating the sum over $\{c_t\}$:

$$\sum_{t=t_0}^{t_0+K-1} \underbrace{\left( q\beta^t - s \sum_{i=t_0}^{t_0+K-1} \beta^i \right)}_{A_1} c_t \tag{430}$$

$$\leq \left( r\beta^{t_0} + p \sum_{i=t_0}^{t_0+K-1} \beta^i \right) a_{t_0} + \sum_{t=t_0+1}^{t_0+K-1} \underbrace{\left( r\beta^t + p \sum_{i=t_0}^{t_0+K-1} \beta^i - \beta^{t-1} \right)}_{A_2} a_t \tag{431}$$

$$- \beta^{t_0+K-1} a_{t_0+K} + b \sum_{t=t_0}^{t_0+K-1} \beta_t. \tag{432}$$

To bound $A_1$ from below, we claim that $s \sum_{i=t_0}^{t_0+K-1} \beta^i \leq \frac{q}{2}\beta^t$. This is equivalent to

$$552\eta^3 K H^2 \sum_{i=t_0}^{t_0+K-1} \beta^i \leq \frac{\eta}{2}\beta^t \tag{433}$$

$$1104\eta^2 K H^2 \sum_{i=0}^{K-1} \beta^i \leq \beta^{t-t_0} \tag{434}$$

$$1104\eta^2 K H^2 \frac{1-\beta^K}{1-\beta} \leq \beta^{t-t_0} \tag{435}$$

$$1104\eta^2 K H^2 \leq \beta^{t-t_0} \frac{1-\beta}{1-\beta^K}, \tag{436}$$

so it suffices to show

$$1104\eta^2 K H^2 \leq \beta^{K-1}\frac{1-\beta}{1-\beta^K}. \tag{437}$$

Using the definition $\beta = 1 - \frac{1}{KR}$,

$$\beta^{K-1}\frac{1-\beta}{1-\beta^K} = \left(1 - \frac{1}{KR}\right)^{K-1}\frac{1}{KR}\frac{1}{1-\left(1-\frac{1}{KR}\right)^K} = \frac{1}{KR}\left(1 - \frac{1}{KR}\right)\frac{\left(1-\frac{1}{KR}\right)^K}{1-\left(1-\frac{1}{KR}\right)^K}. \tag{438}$$

Using the condition $R \geq 2$,

$$\left(1 - \frac{1}{KR}\right)^K = \left(\left(1 - \frac{1}{KR}\right)^{KR}\right)^{1/R} \geq \left(\left(1 - \frac{1}{2}\right)^2\right)^{1/R} = 4^{-1/R}, \tag{439}$$

so

$$\beta^{K-1}\frac{1-\beta}{1-\beta^K} \geq \frac{1}{KR}\left(1 - \frac{1}{KR}\right)\frac{4^{-1/R}}{1-4^{-1/R}} \geq \frac{1}{4K}\frac{1/R}{1-4^{-1/R}} \geq \frac{1}{4\log 4K}, \tag{440}$$

where the last inequality follows from the fact that $f(x) = (x(1-4^{-1/x}))^{-1}$ is decreasing for $x > 0$, and $\lim_{x\to\infty} f(x) = 1/(\log 4)$. Equation 437 follows by applying the condition $\eta \leq 1/(80KH)$. This proves the claim, so $A_1 \leq -\frac{q}{2}\beta^t$.

Returning to Equation 432, we claim that $A_2 \leq 0$. This is equivalent to

$$r\beta^t + p \sum_{i=t_0}^{t_0+K-1} \beta^i - \beta^{t-1} \leq 0 \tag{441}$$

$$r\beta^{t-t_0} + p \sum_{i=0}^{K-1} \beta^i \leq \beta^{t-t_0-1} \tag{442}$$

$$r\beta^{t-t_0} + p\frac{1-\beta^K}{1-\beta} \leq \beta^{t-t_0-1} \tag{443}$$

$$r\beta + p\frac{1-\beta^K}{\beta^{t-t_0-1}(1-\beta)} \leq 1, \tag{444}$$

so it suffices to prove that

$$r\beta + p\frac{1-\beta^K}{\beta^{K-1}(1-\beta)} \le 1. \tag{445}$$

From the definition $\beta = 1 - \frac{1}{KR}$,

$$\frac{1-\beta^K}{\beta^{t-t_0-1}(1-\beta)} = \frac{1-\left(1-\frac{1}{KR}\right)^K}{\left(1-\frac{1}{KR}\right)^K}KR = \left(\left(1-\frac{1}{KR}\right)^{-K}-1\right)KR \tag{446}$$

$$= \left(\left(\left(1-\frac{1}{KR}\right)^{-KR}\right)^{1/R}-1\right)KR \overset{(i)}{\le} \left(4^{1/R}-1\right)KR \overset{(ii)}{\le} 4K, \tag{447}$$

where $(i)$ uses the fact that $(1-1/x)^{-x}$ is decreasing together with $R \ge 2$, and $(ii)$ uses that $(4^{1/x}-1)x$ is decreasing together with $R \ge 2$. So we need to show $r\beta + 4Kp \le 1$. Using the definitions of $r, p$,

$$r\beta + 4Kp = \left(1+\frac{3\eta^2 H}{\lambda M}\right)\left(1-\frac{1}{KR}\right) + \frac{2064\eta^3 K^2 H^2}{\lambda} \tag{448}$$

$$\le 1 + \frac{3\eta^2 H}{\lambda M} - \frac{1}{KR} + \frac{2064\eta^3 K^2 H^2}{\lambda} \tag{449}$$

$$\le 1 + \left(\frac{1}{\lambda}\left(\frac{3\eta^2 H}{M}+2064\eta^3 K^2 H^2\right) - \frac{1}{KR}\right) \tag{450}$$

$$\overset{(i)}{\le} 1, \tag{451}$$

where $(i)$ uses the choice $\lambda = \left(\frac{3}{M}+2064\eta K^2 H\right)\eta^2 KRH$. This proves the claim that $A_2 \le 0$. Returning to Equation 432 and applying our bounds for $A_1$ and $A_2$,

$$\frac{q}{2}\sum_{t=t_0}^{t_0+K-1}\beta^t c_t \le \left(r\beta^{t_0}+p\sum_{i=t_0}^{t_0+K-1}\beta^i\right)a_{t_0} - \beta^{t_0+K-1}a_{t_0+K} + b\sum_{t=t_0}^{t_0+K-1}\beta_t. \tag{452}$$

We can now sum over $t_0 \in \{0, K, 2K, \ldots, (R-1)K\}$:

$$\frac{q}{2}\sum_{t=0}^{KR-1}\beta^t c_t \le \left(r+p\sum_{i=0}^{K-1}\beta^i\right)a_0 + \sum_{t_0\in\{K,\ldots,(R-1)K\}}\left(r\beta^{t_0}+p\sum_{i=t_0}^{t_0+K-1}\beta^i-\beta^{t_0-1}\right)a_{t_0} \tag{453}$$

$$- \beta^{KR-1}a_{KR} + b\sum_{t=0}^{KR-1}\beta_t \tag{454}$$

$$\overset{(i)}{\le} \left(r+p\sum_{i=0}^{K-1}\beta^i\right)a_0 - \beta^{KR-1}a_{KR} + b\sum_{t=0}^{KR-1}\beta_t \tag{455}$$

$$\le \left(r+p\sum_{i=0}^{K-1}\beta^i\right)a_0 + b\sum_{t=0}^{KR-1}\beta_t, \tag{456}$$

where $(i)$ uses $r\beta^{t_0} + p\sum_{i=t_0}^{t_0+K-1}\beta^i - \beta^{t_0-1} \leq 0$, which can be proved similarly as the bound of $A_2$. Let $\alpha_t = \beta^t/\sum_{i=0}^{KR-1}\beta^t$, so

$$\sum_{t=0}^{KR-1}\alpha_t c_t \leq \left(\frac{r}{\sum_{i=0}^{KR-1}\beta^i} + p\sum_{i=0}^{K-1}\alpha_i\right)\frac{2a_0}{q} + \frac{2b}{b} \tag{457}$$

$$\overset{(i)}{\leq} \left(r\frac{1-\beta}{1-\beta^{KR}} + p\right)\frac{2a_0}{q} + \frac{2b}{q} \tag{458}$$

$$\overset{(ii)}{=} \left(\frac{2r}{KR} + p\right)\frac{2a_0}{q} + \frac{2b}{q} \tag{459}$$

$$= \left(\frac{2}{KR}\left(1 + \frac{3\eta^2 H}{\lambda M}\right) + \frac{516\eta^3 KH^2}{\lambda}\right)\frac{2a_0}{q} + \frac{2b}{q} \tag{460}$$

$$= \frac{1}{KR}\left(2 + \frac{1}{\lambda}\left(\frac{6\eta^2 H}{M} + 516\eta^3 K^2 RH^2\right)\right)\frac{2a_0}{q} + \frac{2b}{q} \tag{461}$$

$$\overset{(iii)}{=} \frac{6a_0}{qKR} + \frac{2b}{q}, \tag{462}$$

where $(i)$ uses $\sum_{i=0}^{K-1}\alpha_i \leq \sum_{i=0}^{KR-1}\alpha_i = 1$, $(ii)$ uses $1 - \beta^{KR} = 1 - \left(1 - \frac{1}{KR}\right)^{KR} \geq 1 - 1/e \geq 1/2$, and $(iii)$ uses the choice $\lambda = \left(\frac{3}{M} + 2064\eta K^2 H\right)\eta^2 KRH$.

Finally, we can plug the definitions of $q, a_0, c_t$, and $b$ to obtain

$$\sum_{t=0}^{KR-1}\alpha_t \mathbb{E}[F(\bar{\boldsymbol{w}}_t) - F_*] \tag{463}$$

$$\leq \frac{6\|\bar{\boldsymbol{w}}_0 - \boldsymbol{u}\|^2}{\eta KR} + \frac{6\eta\sigma^2(\boldsymbol{u})}{M} + 72\eta^2 HK\sigma^2(\boldsymbol{u}) + 480\eta^2 K^2 H\zeta^2(\boldsymbol{u}) + (4 + 12\lambda)(F(\boldsymbol{u}) - F_*) \tag{464}$$

$$\overset{(i)}{\leq} \frac{6\|\bar{\boldsymbol{w}}_0 - \boldsymbol{u}\|^2}{\eta KR} + \frac{6\eta\sigma^2(\boldsymbol{u})}{M} + 72\eta^2 HK\sigma^2(\boldsymbol{u}) + 480\eta^2 K^2 H\zeta^2(\boldsymbol{u}) + 3R(F(\boldsymbol{u}) - F_*), \tag{465}$$

where $(i)$ uses the condition $\eta \leq 1/(80KH)$ to bound $4 + 12\lambda$ as

$$4 + 12\lambda \leq 4 + 12\left(\frac{3}{M} + 2064\eta K^2 H\right)\eta^2 KRH \leq 4 + (3 + 26K)12\eta^2 KRH \tag{466}$$

$$\leq 4 + 348\eta^2 K^2 RH \leq R + 4 \leq 3R. \tag{467}$$

Denoting $\hat{\boldsymbol{w}} = \sum_{i=0}^{KR-1}\alpha_i\bar{\boldsymbol{w}}_i$ and applying convexity of $F$ yields

$$\mathbb{E}[F(\hat{\boldsymbol{w}}) - F_*] \leq \frac{6\|\bar{\boldsymbol{w}}_0 - \boldsymbol{u}\|^2}{\eta KR} + \frac{6\eta\sigma^2(\boldsymbol{u})}{M} + 72\eta^2 HK\sigma^2(\boldsymbol{u}) + 480\eta^2 K^2 H\zeta^2(\boldsymbol{u}) + 3R(F(\boldsymbol{u}) - F_*). \tag{468}$$

Lastly, denoting $B = \|\bar{\boldsymbol{w}}_0 - \boldsymbol{u}\|$, we choose $\eta$ as

$$\eta = \min\left\{\frac{1}{80KH}, \frac{B\sqrt{M}}{\sigma(\boldsymbol{u})\sqrt{KR}}, \left(\frac{B^2}{HK^2 R\sigma^2(\boldsymbol{u})}\right)^{1/3}, \left(\frac{B^2}{HK^3 R\zeta^2(\boldsymbol{u})}\right)^{1/3}\right\}, \tag{469}$$

which yields

$$\mathbb{E}[F(\hat{\boldsymbol{w}}) - F_*] \tag{470}$$

$$\leq \frac{480HB^2}{R} + \frac{6\sigma(\boldsymbol{u})B}{\sqrt{MKR}} + \frac{72(H\sigma^2(\boldsymbol{u})B^4)^{1/3}}{K^{1/3}R^{2/3}} + \frac{480(H\zeta^2(\boldsymbol{u})B^4)^{1/3}}{R^{2/3}} + 3R(F(\boldsymbol{u}) - F_*). \tag{471}$$

$\square$

## C.5 Proofs of Corollaries 1 and 2

### C.5.1 Bounding problem parameters

For now, we will only consider the deterministic case, so we can set $\sigma = 0$.

Next, we bound the smoothness constant $H$. For the loss of a single sample $\ell(\boldsymbol{w}, \boldsymbol{x}, y) = \log(1 + \exp(-y\langle \boldsymbol{w}, \boldsymbol{x}\rangle))$, the Hessian $\nabla^2 \ell$ (with respect to $\boldsymbol{w}$) is

$$\frac{\partial^2 \ell}{\partial \boldsymbol{w}^2}(\boldsymbol{w}, \boldsymbol{x}, y) = \frac{\exp(y\langle \boldsymbol{w}, \boldsymbol{x}\rangle)}{(1 + \exp(y\langle \boldsymbol{w}, \boldsymbol{x}\rangle))^2} \boldsymbol{x}\boldsymbol{x}^T, \tag{472}$$

so the smoothness constant of $\ell$ is

$$\left\| \frac{\partial^2 \ell}{\partial \boldsymbol{w}^2}(\boldsymbol{w}, \boldsymbol{x}, y) \right\| = \sup_{\boldsymbol{w} \in \mathbb{R}^d} \left\{ \frac{\exp(y\langle \boldsymbol{w}, \boldsymbol{x}\rangle)}{(1 + \exp(y\langle \boldsymbol{w}, \boldsymbol{x}\rangle))^2} \|\boldsymbol{x}\|^2 \right\} = \frac{1}{4}\|\boldsymbol{x}\|^2. \tag{473}$$

Therefore, the smoothness constant of $F$ (and similarly each $F_m$) can be bounded as

$$\|\nabla^2 F(\boldsymbol{w})\| = \left\| \frac{1}{nM} \sum_{m=1}^{M} \sum_{i=1}^{n} \frac{\partial^2 \ell}{\partial w^2}(w, x_{mi}, y_{mi}) \right\| \tag{474}$$

$$\leq \frac{1}{nM} \sum_{m=1}^{M} \sum_{i=1}^{n} \left\| \frac{\partial^2 \ell}{\partial w^2}(w, x_{mi}, y_{mi}) \right\| \tag{475}$$

$$\leq \frac{1}{4nM} \sum_{m=1}^{M} \sum_{i=1}^{n} \|x_{mi}\|^2 \tag{476}$$

$$\overset{(i)}{\leq} \frac{1}{4}, \tag{477}$$

where $(i)$ uses the assumption from Section 3 that $\|\boldsymbol{x}_{mi}\| \leq 1$ for every $m \in [M], i \in [n]$. This allows us to use $H \leq 1/4$ when applying Theorem 5 to the case of logistic regression.

To upper bound the data heterogeneity $\zeta$, we need a bound for

$$\max_{m \in [M]} \sup_{\boldsymbol{w} \in \mathbb{R}^d} \|\nabla F_m(\boldsymbol{w}) - \nabla F(\boldsymbol{w})\|. \tag{478}$$

Notice that $\ell$ is Lipschitz in terms of $\boldsymbol{w}$, since

$$\left\| \frac{\partial \ell}{\partial \boldsymbol{w}}(\boldsymbol{w}, \boldsymbol{x}, y) \right\| = \left\| \frac{-y}{1 + \exp(y\langle \boldsymbol{w}, \boldsymbol{x}\rangle)} \boldsymbol{x} \right\| = \frac{1}{1 + \exp(y\langle \boldsymbol{w}, \boldsymbol{x}\rangle)} \|\boldsymbol{x}\| \leq \|\boldsymbol{x}\|. \tag{479}$$

This leads to a simple upper bound of the gradient dissimilarity as:

$$\|\nabla F_m(\boldsymbol{w}) - \nabla F(\boldsymbol{w})\| \leq \|\nabla F_m(\boldsymbol{w})\| + \|\nabla F(\boldsymbol{w})\| \tag{480}$$

$$\leq \frac{1}{n} \sum_{i=1}^{n} \left\| \frac{\partial \ell}{\partial \boldsymbol{w}}(\boldsymbol{w}, \boldsymbol{x}_{mi}, y_{mi}) \right\| + \frac{1}{nM} \sum_{m'=1}^{M} \sum_{i=1}^{n} \left\| \frac{\partial \ell}{\partial \boldsymbol{w}}(\boldsymbol{w}, \boldsymbol{x}_{m'i}, y_{m'i}) \right\| \tag{481}$$

$$\leq \frac{1}{n} \sum_{i=1}^{n} \|\boldsymbol{x}_{mi}\| + \frac{1}{nM} \sum_{m'=1}^{M} \sum_{i=1}^{n} \|\boldsymbol{x}_{m'i}\| \tag{482}$$

$$\leq 2. \tag{483}$$

Although this bound may appear pessimistic, the following lemma shows that the bound achieved (up to constant factors) in a simple case.

**Lemma 23.** *For $d = 2, M = 2, n = 2$ there exist client datasets for logistic regression such that the corresponding optimization problem satifies*

$$\max_{m \in [M]} \sup_{\boldsymbol{w} \in \mathbb{R}^d} \|\nabla F_i(\boldsymbol{w}) - \nabla F(\boldsymbol{w})\| \geq \frac{1}{2}. \tag{484}$$

*Proof.* The previous bound on $\zeta$ can be achieved in a situation where, for a particular weight $\boldsymbol{w}$, both samples from client $m = 1$ are classified correctly, while both samples from client $m = 2$ are classified incorrectly. Letting $\|\boldsymbol{w}\| \to \infty$ while preserving the direction of $\boldsymbol{w}$ achieves the desired bound.

Let $\boldsymbol{w}_* = e_2$, and consider the following client datasets:

$$D_1 = \{((0, 1), 1), ((0, -1), -1)\} \tag{485}$$

$$D_2 = \{((-2/\sqrt{5}, 1/\sqrt{5}), 1), ((2/\sqrt{5}, -1/\sqrt{5}), -1)\}. \tag{486}$$

It is straightforward to verify that this dataset is consistent with the ground truth parameter $\boldsymbol{w}_*$, and that $1 = \sup_{m \in [M]} \left\{ \frac{1}{n} \sum_{i=1}^{n} \|\boldsymbol{x}_{mi}\| \right\}$. For any $\boldsymbol{w} = (w_1, w_2) \in \mathbb{R}^2$, the gradient of each local objective has the following closed form:

$$\nabla F_1(\boldsymbol{w}) = \frac{1}{2} \left( \frac{-1}{\exp(w_2) + 1} \boldsymbol{e}_2 + \frac{1}{\exp(w_2) + 1} (-\boldsymbol{e}_2) \right) = \frac{-1}{\exp(w_2) + 1} \boldsymbol{e}_2 \tag{487}$$

$$\nabla F_2(\boldsymbol{w}) = \frac{1}{2} \left( \frac{-1}{\exp(-2w_1 + w_2) + 1} \frac{1}{\sqrt{5}} (-2\boldsymbol{e}_1 + \boldsymbol{e}_2) + \frac{1}{\exp(-2w_1 + w_2) + 1} \frac{1}{\sqrt{5}} (2\boldsymbol{e}_1 - \boldsymbol{e}_2) \right) \tag{488}$$

$$= \frac{1}{\sqrt{5}(\exp(-2w_1 + w_2) + 1)} (2\boldsymbol{e}_1 - \boldsymbol{e}_2). \tag{489}$$

Now consider $\boldsymbol{w} = \lambda(1, 1)$ for $\lambda > 0$. This yields

$$\nabla F_1(\boldsymbol{w}) = \frac{-1}{\exp(\lambda) + 1} \boldsymbol{e}_2 \tag{490}$$

$$\nabla F_2(\boldsymbol{w}) = \frac{1}{\sqrt{5}(\exp(-2\lambda) + 1)} (2\boldsymbol{e}_1 - \boldsymbol{e}_2). \tag{491}$$

Therefore, as $\lambda \to \infty$, the local (and global) gradients approach

$$\nabla F_1(\boldsymbol{w}) \to 0 \tag{492}$$

$$\nabla F_2(\boldsymbol{w}) \to \left( \frac{2}{\sqrt{5}} \boldsymbol{e}_1 - \frac{1}{\sqrt{5}} \boldsymbol{e}_2 \right) \tag{493}$$

$$\nabla F(\boldsymbol{w}) \to \frac{1}{2} \left( \frac{2}{\sqrt{5}} \boldsymbol{e}_1 - \frac{1}{\sqrt{5}} \boldsymbol{e}_2 \right). \tag{494}$$

Finally, consider the gradient dissimilarity $\|\nabla F_2(\boldsymbol{w}) - \nabla F(\boldsymbol{w})\|$ as $\lambda \to \infty$:

$$\max_{m \in [M]} \sup_{\boldsymbol{w} \in \mathbb{R}^d} \|\nabla F_m(\boldsymbol{w}) - \nabla F(\boldsymbol{w})\| \geq \lim_{\lambda \to \infty} \|\nabla F_2(\boldsymbol{w}) - \nabla F(\boldsymbol{w})\| \tag{495}$$

$$= \frac{1}{2} \left\| \frac{2}{\sqrt{5}} \boldsymbol{e}_1 - \frac{1}{\sqrt{5}} \boldsymbol{e}_2 \right\| \tag{496}$$

$$= \frac{1}{2}. \tag{497}$$

$\square$

Lemma 23 demonstrates that the bound $\zeta \leq 2$ is tight up to constant factors (in the worst-case over possible datasets).

We can also bound the local heterogeneity $\zeta(\boldsymbol{u})$ at an arbitrary point $\boldsymbol{u}$ as:

$$\zeta^2(\boldsymbol{u}) = \frac{1}{M} \sum_{m=1}^{M} \|\nabla F_m(\boldsymbol{u}) - \nabla F(\boldsymbol{u})\|^2 \tag{498}$$

$$\leq \frac{2}{M} \sum_{m=1}^{M} \left( \|\nabla F_m(\boldsymbol{u})\|^2 + \|\nabla F(\boldsymbol{u})\|^2 \right) \tag{499}$$

$$\leq \frac{4H}{M} \sum_{m=1}^{M} \left( (F_m(\boldsymbol{u}) - F_m*) + (F(\boldsymbol{u}) - F_*) \right) \tag{500}$$

$$\overset{(i)}{\leq} 8H(F(\boldsymbol{u}) - F_*) \tag{501}$$

$$\overset{(ii)}{\leq} 2(F(\boldsymbol{u}) - F_*), \tag{502}$$

where $(i)$ uses the fact that $\frac{1}{M} \sum_{m=1}^{M} F_m^* = F_*$, and $(ii)$ uses the previously derived bound $H \leq 1/4$.

### C.5.2 Choosing a comparator

Let $\hat{\boldsymbol{w}}$ denote the output of Local SGD for the logistic regression problem. For simplicity, we will assume that $\bar{\boldsymbol{w}}_0 = 0$.

**Global Heterogeneity/Noise** For the deterministic case ($\sigma = 0$), we can restate the convergence rate from Theorem 5 as:

$$\mathbb{E}\left[F(\hat{\boldsymbol{w}}) - F_*\right] \leq \frac{4H\|\boldsymbol{u}\|^2}{KR} + \frac{(H\zeta^2\|\boldsymbol{u}\|^4)^{1/3}}{R^{2/3}} + 2(F(\boldsymbol{u}) - F_*). \tag{503}$$

Also, we can plug in the bounds $H \leq 1/4$ and $\zeta \leq 2$ (from Section C.5.1) to obtain

$$\mathbb{E}\left[F(\hat{\boldsymbol{w}}) - F_*\right] \leq \frac{\|\boldsymbol{u}\|^2}{KR} + \frac{\|\boldsymbol{u}\|^{4/3}}{R^{2/3}} + 2(F(\boldsymbol{u}) - F_*). \tag{504}$$

Recall that $\boldsymbol{w}_*$ is the maximum margin predictor for the global dataset with $\|\boldsymbol{w}_*\| = 1$. We will choose our comparator as $\boldsymbol{u} = \lambda \boldsymbol{w}_*$ for some $\lambda > 0$ that will be chosen later. The error $F(\boldsymbol{u}) - F_*$ can then be bounded as

$$F(\boldsymbol{u}) - F_* = \frac{1}{nM} \sum_{m=1}^{M} \sum_{i=1}^{n} \log(1 + \exp(-\lambda y_{mi} \langle \boldsymbol{w}_*, \boldsymbol{x}_{mi} \rangle)) \tag{505}$$

$$\overset{(i)}{\leq} \frac{1}{nM} \sum_{m=1}^{M} \sum_{i=1}^{n} \log(1 + \exp(-\lambda \gamma)) \tag{506}$$

$$= \log(1 + \exp(-\lambda \gamma)) \tag{507}$$

$$\overset{(ii)}{\leq} \exp(-\lambda \gamma), \tag{508}$$

where $(i)$ uses the definition of the margin $\gamma$ from Equation 2, and $(ii)$ uses $\log(1 + x) \leq x$. The convergence rate then simplifies to

$$\mathbb{E}\left[F(\hat{\boldsymbol{w}}) - F_*\right] \leq \frac{\lambda^2}{KR} + \frac{\lambda^{4/3}}{R^{2/3}} + 2\exp(-\lambda \gamma). \tag{509}$$

Denoting $[x]_+ = \max\{0, x\}$, we will use the choice

$$\lambda = \frac{1}{\gamma} \left[ \min \left\{ \log\left(KR\gamma^2\right), \log\left(R^{2/3}\gamma^{4/3}\right) \right\} \right]_+. \tag{510}$$

So the last term of Equation 509 can be bounded as

$$\exp(-\lambda \gamma) \leq \max \left\{ \frac{1}{KR\gamma^2}, \frac{1}{R^{2/3}\gamma^{4/3}} \right\} \leq \frac{1}{KR\gamma^2} + \frac{1}{R^{2/3}\gamma^{4/3}}. \tag{511}$$

So plugging the choice of $\lambda$ into Equation 509 yields

$$\mathbb{E}\left[F(\hat{\boldsymbol{w}}) - F_*\right] \leq \frac{1}{KR\gamma^2}\left[\log\left(KR\gamma^2\right)\right]_+^2 + \frac{1}{R^{2/3}\gamma^{4/3}}\left[\log\left(R^{2/3}\gamma^{4/3}\right)\right]_+^{4/3} + 2\exp(-\lambda\gamma) \tag{512}$$

$$\leq \frac{1}{KR\gamma^2}\left(2 + \left[\log\left(KR\gamma^2\right)\right]_+^2\right) + \frac{1}{R^{2/3}\gamma^{4/3}}\left(2 + \left[\log\left(R^{2/3}\gamma^{4/3}\right)\right]_+^{4/3}\right). \tag{513}$$

This proves Corollary 1, since Equation 513 is exactly the upper bound from Corollary 1.

**Local Heterogeneity/Noise**   Restating the convergence rate from Theorem 6 (with $\sigma(\boldsymbol{u}) = 0$):

$$\mathbb{E}[F(\hat{\boldsymbol{w}}) - F_*] \leq \frac{H\|\boldsymbol{u}\|^2}{R} + \frac{(H\zeta^2(\boldsymbol{u})\|\boldsymbol{u}\|^4)^{1/3}}{R^{2/3}} + R(F(\boldsymbol{u}) - F_*), \tag{514}$$

we can use the our bounds on $H$ and $\zeta(\boldsymbol{u})$ from Section C.5.1 to obtain

$$\mathbb{E}[F(\hat{\boldsymbol{w}}) - F_*] \leq \frac{\|\boldsymbol{u}\|^2}{R} + \frac{((F(\boldsymbol{u}) - F_*)\|\boldsymbol{u}\|^4)^{1/3}}{R^{2/3}} + R(F(\boldsymbol{u}) - F_*). \tag{515}$$

We again choose $\boldsymbol{u} = \lambda\boldsymbol{w}_*$, so that

$$\mathbb{E}[F(\hat{\boldsymbol{w}}) - F_*] \leq \frac{\lambda^2}{R} + \frac{\lambda^{4/3}}{R^{2/3}}\exp^{1/3}(-\gamma\lambda) + R\exp(-\gamma\lambda). \tag{516}$$

Here, we use the choice

$$\lambda = \frac{2}{\gamma}\left[\log(R)\right]_+, \tag{517}$$

which yields

$$\mathbb{E}[F(\hat{\boldsymbol{w}}) - F_*] \leq \frac{1}{\gamma^2 R}\left(1 + [\log(R)]_+^2\right) + \frac{1}{R^{2/3}}\frac{1}{\gamma^{4/3}}[\log(R)]_+^{4/3}\left(\frac{1}{R^2}\right)^{1/3} \tag{518}$$

$$= \frac{1}{\gamma^2 R}\left(1 + [\log(R)]_+^2\right) + \frac{1}{\gamma^{4/3}R^{4/3}}[\log(R)]_+^{4/3}. \tag{519}$$

This proves Corollary 2, since Equation 519 is exactly the upper bound from Corollary 2.

## C.6   AN ADDITIONAL BASELINE USING A REGULARIZED OBJECTIVE

Since logistic regression is not in the problem class analyzed by Woodworth et al. (2020b); Koloskova et al. (2020), we previously extended their analysis for the case that a minimizer of the global objective does not exist (Corollaries 1 and 2). An alternative baseline, which we derive in this section, is to apply their original results directly to a regularized objective that does admit a minimizer. Specifically, we define

$$\tilde{F}_m(\boldsymbol{w}) = F_m(\boldsymbol{w}) + \frac{\lambda}{2}\|\boldsymbol{w}\|^2, \tag{520}$$

and $\tilde{F} = \frac{1}{M}\sum_{m=1}^{M}\tilde{F}_m$. These objectives are strongly convex, so we can apply the results of Woodworth et al. (2020b); Koloskova et al. (2020) for Local SGD with strongly convex objectives. By choosing the regularization strength $\lambda > 0$ small enough, we can guarantee that the output $\hat{\boldsymbol{w}}$ of Local SGD is an $\epsilon$-approximate solution for the unregularized problem $F$. This leads to the following results.

**Corollary 4** (Using Woodworth et al. (2020b))**.** *Let $\epsilon > 0$, and let $\hat{\boldsymbol{w}}$ be the output of Local GD on objective $\tilde{F}$. Then $F(\hat{\boldsymbol{w}}) \leq \epsilon$ when*

$$R \geq \tilde{\Omega}\left(\frac{1}{\gamma^2 K\epsilon} + \frac{1}{\gamma^2\epsilon^{3/2}}\right). \tag{521}$$

**Corollary 5** (Using Koloskova et al. (2020)). *Let $\epsilon > 0$, and let $\hat{w}$ be the output of Local GD on objective $\tilde{F}$. Then $F(\hat{w}) \leq \epsilon$ when*

$$R \geq \tilde{\Omega}\left(\frac{1}{\gamma^2 \epsilon}\right). \tag{522}$$

Comparing with Table 1, we see that the complexities of these regularized baselines match those of Corollaries 1 and 2 in the dominating terms.

In the remainder of this section, we prove Corollaries 4 and 5. First, we must bound the problem parameters (smoothness constant, gradient dissimilarity, etc) of the regularized problem.

For the unregularized objective $F$, we already bounded $H \leq 1/4$ and $\zeta \leq 2$. We denote the corresponding parameters of $\tilde{F}$ as $\tilde{H}$ and $\tilde{\zeta}$, so that

$$\tilde{H} := \sup_{\boldsymbol{w} \in \mathbb{R}^d} \|\nabla^2 \tilde{F}(\boldsymbol{w})\| = \sup_{\boldsymbol{w} \in \mathbb{R}^d} \|\nabla^2 F(\boldsymbol{w}) + \lambda I_d\| \leq H + \lambda, \tag{523}$$

and

$$\tilde{\zeta} := \max_{m \in [M]} \sup_{\boldsymbol{w} \in \mathbb{R}^d} \|\nabla \tilde{F}_m(\boldsymbol{w}) - \nabla \tilde{F}(\boldsymbol{w})\| = \max_{m \in [M]} \sup_{\boldsymbol{w} \in \mathbb{R}^d} \|\nabla F_m(\boldsymbol{w}) - \nabla F(\boldsymbol{w})\| = \zeta. \tag{524}$$

So $\tilde{H} \leq 1/4 + \lambda$ and $\tilde{\zeta} \leq 2$. We also denote $\tilde{\boldsymbol{w}}_* = \arg\min_{\boldsymbol{w} \in \mathbb{R}^d} \tilde{F}(\boldsymbol{w})$ and $\tilde{B} = \|\tilde{\boldsymbol{w}}_*\|$. We can then bound $\tilde{B}$ as follows.

$$\frac{\lambda}{2}\|\tilde{\boldsymbol{w}}_*\|^2 = \tilde{F}(\tilde{\boldsymbol{w}}_*) - F(\tilde{\boldsymbol{w}}_*) \leq \tilde{F}(\tilde{\boldsymbol{w}}_*) \leq \tilde{F}(\boldsymbol{u}), \tag{525}$$

where $\boldsymbol{u} \in \mathbb{R}^d$ is arbitrary. Set $\boldsymbol{u} = t\boldsymbol{w}_*$ with $t = \frac{1}{\gamma}\log(e + \gamma/\lambda)$, so

$$\tilde{F}(\boldsymbol{u}) = F(\boldsymbol{u}) + \frac{\lambda}{2}\|\boldsymbol{u}\|^2 \tag{526}$$

$$= \frac{1}{Mn}\sum_{m=1}^{M}\sum_{i=1}^{n} \log(1 + \exp(-t\langle \boldsymbol{w}_*, \boldsymbol{x}_{mi}\rangle)) + \frac{\lambda}{2}t^2 \tag{527}$$

$$\leq \frac{1}{Mn}\sum_{m=1}^{M}\sum_{i=1}^{n} \exp(-t\langle \boldsymbol{w}_*, \boldsymbol{x}_{mi}\rangle) + \frac{\lambda}{2}t^2 \tag{528}$$

$$\leq \exp(-\gamma t) + \frac{\lambda}{2}t^2 \tag{529}$$

$$= \frac{1}{e + \gamma/\lambda} + \frac{\lambda}{2\gamma^2}\log^2(e + \gamma/\lambda) \tag{530}$$

$$\leq \frac{\lambda}{\gamma} + \frac{\lambda}{2\gamma^2}\log^2(e + \gamma/\lambda) \tag{531}$$

$$\leq \frac{3\lambda}{2\gamma^2}\log^2(e + \gamma/\lambda) \tag{532}$$

Combining with Equation 525 yields

$$\frac{\lambda}{2}\|\tilde{\boldsymbol{w}}_*\|^2 \leq \frac{3\lambda}{2\gamma^2}\log^2(e + \gamma/\lambda) \tag{533}$$

$$\|\tilde{\boldsymbol{w}}_*\|^2 \leq \frac{3}{\gamma^2}\log^2(e + \gamma/\lambda) \tag{534}$$

$$\|\tilde{\boldsymbol{w}}_*\| \leq \frac{\sqrt{3}}{\gamma}\log(e + \gamma/\lambda), \tag{535}$$

so that $\tilde{B} \leq \widetilde{\mathcal{O}}(1/\gamma)$. Also, we denote $\tilde{\zeta}_*^2 = \frac{1}{M} \sum_{m=1}^{M} \|\nabla \tilde{F}_m(\tilde{\boldsymbol{w}}_*)\|^2$, which is required for the guarantee of Koloskova et al. (2020), and we can bound this quantity as

$$\tilde{\zeta}_*^2 = \frac{1}{M} \sum_{m=1}^{M} \|\nabla \tilde{F}_m(\tilde{\boldsymbol{w}}_*) - \nabla \tilde{F}(\tilde{\boldsymbol{w}}_*)\|^2 \tag{536}$$

$$= \frac{1}{M} \sum_{m=1}^{M} \|\nabla F_m(\tilde{\boldsymbol{w}}_*) - \nabla F(\tilde{\boldsymbol{w}}_*)\|^2 \tag{537}$$

$$\overset{(i)}{=} \zeta_*^2(\tilde{\boldsymbol{w}}_*) \tag{538}$$

$$\overset{(ii)}{\leq} 2F(\tilde{\boldsymbol{w}}_*) \tag{539}$$

$$\leq 2\tilde{F}(\tilde{\boldsymbol{w}}_*) \tag{540}$$

$$\leq 2\tilde{F}(\boldsymbol{u}) \tag{541}$$

$$\overset{(iii)}{\leq} \frac{3\lambda}{\gamma^2} \log^2(e + \gamma/\lambda), \tag{542}$$

where $(i)$ uses the definition of $\zeta_*^2$ from Equation 498, $(ii)$ uses Equation 502, and $(iii)$ uses Equation 532. Lastly, we will denote $\tilde{F}_* = \inf_{\boldsymbol{w} \in \mathbb{R}^d} \tilde{F}(\boldsymbol{w})$, which we have already bounded in Equation 540 through Equation 542 as

$$\tilde{F}_* = \tilde{F}(\tilde{\boldsymbol{w}}_*) \leq \frac{3\lambda}{2\gamma^2} \log^2(e + \gamma/\lambda). \tag{543}$$

With these bounds of $\tilde{H}, \tilde{\zeta}, \tilde{B}, \tilde{\zeta}_*$, and $\tilde{F}_*$, we can now prove Corollaries 4 and 5.

*Proof of Corollary 4.* From Theorem 3 of Woodworth et al. (2020b), we know that

$$\tilde{F}(\hat{\boldsymbol{w}}) - \tilde{F}_* \leq \widetilde{\mathcal{O}} \left( \frac{\tilde{H}^2 \tilde{B}^2}{\tilde{H} K R + \lambda K^2 R^2} + \frac{\tilde{H} \tilde{\zeta}^2}{\lambda^2 R^2} \right) \tag{544}$$

$$\tilde{F}(\hat{\boldsymbol{w}}) \leq \tilde{F}_* + \widetilde{\mathcal{O}} \left( \frac{\tilde{H}^2 \tilde{B}^2}{\tilde{H} K R + \lambda K^2 R^2} + \frac{\tilde{H} \tilde{\zeta}^2}{\lambda^2 R^2} \right) \tag{545}$$

$$\tilde{F}(\hat{\boldsymbol{w}}) \overset{(i)}{\leq} \widetilde{\mathcal{O}} \left( \frac{\lambda}{\gamma^2} + \frac{\tilde{H}^2 \tilde{B}^2}{\tilde{H} K R + \lambda K^2 R^2} + \frac{\tilde{H} \tilde{\zeta}^2}{\lambda^2 R^2} \right) \tag{546}$$

$$F(\hat{\boldsymbol{w}}) \overset{(ii)}{\leq} \widetilde{\mathcal{O}} \left( \frac{\lambda}{\gamma^2} + \frac{\tilde{H}^2 \tilde{B}^2}{\tilde{H} K R + \lambda K^2 R^2} + \frac{\tilde{H} \tilde{\zeta}^2}{\lambda^2 R^2} \right) \tag{547}$$

$$F(\hat{\boldsymbol{w}}) \leq \widetilde{\mathcal{O}} \left( \frac{\lambda}{\gamma^2} + \frac{\tilde{H} \tilde{B}^2}{K R} + \frac{\tilde{H} \tilde{\zeta}^2}{\lambda^2 R^2} \right), \tag{548}$$

where $(i)$ uses $\tilde{F}_* \leq \widetilde{\mathcal{O}}(\lambda/\gamma^2)$ and $(ii)$ uses $F \leq \tilde{F}$. Now we choose

$$\lambda = \frac{\tilde{H}^{1/3} \tilde{\zeta}^{1/3} \gamma^{2/3}}{R^{2/3}}, \tag{549}$$

which yields

$$F(\hat{\boldsymbol{w}}) \leq \widetilde{\mathcal{O}} \left( \frac{\tilde{H} \tilde{B}^2}{K R} + \frac{\tilde{H}^{1/3} \tilde{\zeta}^{1/3}}{\gamma^{4/3} R^{2/3}} \right) \tag{550}$$

$$\overset{(i)}{\leq} \widetilde{\mathcal{O}} \left( \frac{1}{\gamma^2 K R} + \frac{1}{\gamma^{4/3} R^{2/3}} \right), \tag{551}$$

where $(i)$ uses $\tilde{H}, \tilde{\zeta} \leq \mathcal{O}(1)$ and $\tilde{B} \leq \widetilde{\mathcal{O}}(1/\gamma)$. Therefore, the condition $F(\hat{\boldsymbol{w}}) \leq \epsilon$ is ensured whenever

$$R \geq \widetilde{\Omega} \left( \frac{1}{\gamma^2 K \epsilon} + \frac{1}{\gamma^2 \epsilon^{3/2}} \right). \tag{552}$$

$\square$

*Proof of Corollary 5.* We use the result from Koloskova et al. (2020) as stated in Table 2 of Wood-worth et al. (2020b): if $R \geq \widetilde{\Omega}(\tilde{H}/\lambda)$, then

$$\tilde{F}(\hat{\boldsymbol{w}}) - \tilde{F}_* \leq \widetilde{\mathcal{O}}\left(\frac{\tilde{H}\tilde{\zeta}_*^2}{\lambda^2 R^2}\right) \tag{553}$$

$$\tilde{F}(\hat{\boldsymbol{w}}) \leq \tilde{F}_* + \widetilde{\mathcal{O}}\left(\frac{\tilde{H}\tilde{\zeta}_*^2}{\lambda^2 R^2}\right) \tag{554}$$

$$\tilde{F}(\hat{\boldsymbol{w}}) \overset{(i)}{\leq} \widetilde{\mathcal{O}}\left(\frac{\lambda}{\gamma^2} + \frac{\tilde{H}\tilde{\zeta}_*^2}{\lambda^2 R^2}\right) \tag{555}$$

$$F(\hat{\boldsymbol{w}}) \overset{(ii)}{\leq} \widetilde{\mathcal{O}}\left(\frac{\lambda}{\gamma^2} + \frac{\tilde{H}\tilde{\zeta}_*^2}{\lambda^2 R^2}\right) \tag{556}$$

$$F(\hat{\boldsymbol{w}}) \overset{(iii)}{\leq} \widetilde{\mathcal{O}}\left(\frac{\lambda}{\gamma^2} + \frac{1}{\lambda\gamma^2 R^2}\right), \tag{557}$$

where $(i)$ uses $\tilde{F}_* \leq \widetilde{\mathcal{O}}(\lambda/\gamma^2)$, $(ii)$ uses $F \leq \tilde{F}$, and $(iii)$ uses $\tilde{H} \leq \mathcal{O}(1)$ and $\tilde{\zeta}_*^2 \leq \widetilde{\mathcal{O}}(\lambda/\gamma^2)$. Now we choose $\lambda \leq \widetilde{\mathcal{O}}(1/R)$, so that

$$F(\hat{\boldsymbol{w}}) \leq \widetilde{\mathcal{O}}\left(\frac{1}{\gamma^2 R}\right). \tag{558}$$

Note also that this choice of $\lambda$ is compatible with the requirement $R \geq \widetilde{\Omega}(\tilde{H}/\lambda) = \widetilde{\Omega}(1/\lambda)$. There-fore, we can ensure that $F(\hat{\boldsymbol{w}}) \leq \epsilon$ whenever $R \geq \widetilde{\Omega}(1/(\gamma^2 \epsilon))$. □

# D   TECHNICAL LEMMAS

## D.1   LEMMAS FOR SECTION 4/THEOREM 1

**Lemma 24.** *For all $z \in \mathbb{R}$,*
$$0 < \ell''(z) \leq |\ell'(z)| \leq \ell(z). \tag{559}$$

*Also, for all $z \geq 0$,*
$$\ell(z) \leq 2|\ell'(z)|. \tag{560}$$

*Proof.* From the definition of $\ell$,

$$\ell'(z) = \frac{-\exp(-z)}{1 + \exp(-z)} = \frac{-1}{\exp(z) + 1} \tag{561}$$

$$\ell''(z) = \frac{\exp(z)}{(\exp(z) + 1)^2}. \tag{562}$$

Therefore $\ell''(z) > 0$, and

$$\ell''(z) = \frac{\exp(z)}{\exp(z) + 1}\frac{1}{\exp(z) + 1} \leq \frac{1}{\exp(z) + 1} = |\ell'(z)|. \tag{563}$$

Also

$$\ell(z) = \log(1 + \exp(-z)) \overset{(i)}{\geq} \frac{\exp(-z)}{1 + \exp(-z)} = \frac{1}{\exp(z) + 1} = |\ell'(z)|, \tag{564}$$

where $(i)$ uses the inequality $\log(1 + x) \geq \frac{x}{1+x}$, which can be derived as

$$\log(1 + x) = \log(1) + \int_0^x \frac{d\log(1 + t)}{dt}\bigg|_{t=s} ds = \int_0^x \frac{1}{1 + s}ds \geq \int_0^x \frac{1}{1 + x}ds = \frac{x}{1 + x}. \tag{565}$$

This proves Equation 559.

For Equation 560,

$$\ell(z) = \log(1 + \exp(-z)) \overset{(i)}{\leq} \exp(-z) \overset{(ii)}{\leq} \frac{2}{\exp(z) + 1} = 2|\ell'(z)|, \tag{566}$$

where $(i)$ uses $\log(1 + x) \leq x$ for all $x$, and $(ii)$ uses the condition $z \geq 0$. $\qquad\square$

**Lemma 25.** *For all $\boldsymbol{w} \in \mathbb{R}^d$ and $m \in [M]$,*

$$\|\nabla F_m(\boldsymbol{w})\| \leq F_m(\boldsymbol{w}) \tag{567}$$

$$\|\nabla^2 F_m(\boldsymbol{w})\| \leq F_m(\boldsymbol{w}). \tag{568}$$

*Consequently,*

$$\|\nabla F(\boldsymbol{w})\| \leq F(\boldsymbol{w}) \tag{569}$$

$$\|\nabla^2 F(\boldsymbol{w})\| \leq F(\boldsymbol{w}). \tag{570}$$

*Proof.* From the definition of $F_m$,

$$\nabla F_m(\boldsymbol{w}) = \frac{1}{n} \sum_{i=1}^{n} \ell'(y_{mi}\langle \boldsymbol{w}, \boldsymbol{x}_{mi}\rangle)(-y_{mi}\boldsymbol{x}_{mi}) = \frac{1}{n} \sum_{i=1}^{n} |\ell'(y_{mi}\langle \boldsymbol{w}, \boldsymbol{x}_{mi}\rangle)|y_{mi}\boldsymbol{x}_{mi}, \tag{571}$$

therefore

$$\|\nabla F_m(\boldsymbol{w})\| \leq \frac{1}{n} \sum_{i=1}^{n} |\ell'(y_{mi}\langle \boldsymbol{w}, \boldsymbol{x}_{mi}\rangle)|\|\boldsymbol{x}_{mi}\| \tag{572}$$

$$\overset{(i)}{\leq} \frac{1}{n} \sum_{i=1}^{n} |\ell'(y_{mi}\langle \boldsymbol{w}, \boldsymbol{x}_{mi}\rangle)| \tag{573}$$

$$\overset{(ii)}{\leq} \frac{1}{n} \sum_{i=1}^{n} \ell(y_{mi}\langle \boldsymbol{w}, \boldsymbol{x}_{mi}\rangle) \tag{574}$$

$$= F_m(\boldsymbol{w}), \tag{575}$$

where $(i)$ uses $\|\boldsymbol{x}_{mi}\| \leq 1$ and $(ii)$ uses Equation 559. This proves Equation 567.

Similarly,

$$\nabla^2 F_m(\boldsymbol{w}) = \frac{1}{n} \sum_{i=1}^{n} \ell''(y_{mi}\langle \boldsymbol{w}, \boldsymbol{x}_{mi}\rangle)\boldsymbol{x}_{mi}\boldsymbol{x}_{mi}^{\mathsf{T}}, \tag{576}$$

so

$$\|\nabla^2 F_m(\boldsymbol{w})\| \leq \frac{1}{n} \sum_{i=1}^{n} \ell''(y_{mi}\langle \boldsymbol{w}, \boldsymbol{x}_{mi}\rangle) \|\boldsymbol{x}_{mi}\boldsymbol{x}_{mi}^{\mathsf{T}}\| \tag{577}$$

$$= \frac{1}{n} \sum_{i=1}^{n} \ell''(y_{mi}\langle \boldsymbol{w}, \boldsymbol{x}_{mi}\rangle) \|\boldsymbol{x}_{mi}\|^2 \tag{578}$$

$$\overset{(i)}{\leq} \frac{1}{n} \sum_{i=1}^{n} \ell''(y_{mi}\langle \boldsymbol{w}, \boldsymbol{x}_{mi}\rangle) \tag{579}$$

$$\overset{(ii)}{\leq} \frac{1}{n} \sum_{i=1}^{n} \ell(y_{mi}\langle \boldsymbol{w}, \boldsymbol{x}_{mi}\rangle) \tag{580}$$

$$= F_m(\boldsymbol{w}), \tag{581}$$

where $(i)$ uses $\|\boldsymbol{x}_{mi}\| \leq 1$ and $(ii)$ uses 559. This proves Equation 568

Equation 569 follows from Equation 567 by

$$\|\nabla F(\boldsymbol{w})\| = \left\|\frac{1}{M} \sum_{m=1}^{M} \nabla F_m(\boldsymbol{w})\right\| \leq \frac{1}{M} \sum_{m=1}^{M} \|\nabla F_m(\boldsymbol{w})\| \leq \frac{1}{M} \sum_{m=1}^{M} F_m(\boldsymbol{w}) = F(\boldsymbol{w}), \tag{582}$$

and Equation 570 follows from Equation 568 by

$$\|\nabla^2 F(\boldsymbol{w})\| = \left\| \frac{1}{M} \sum_{m=1}^{M} \nabla^2 F_m(\boldsymbol{w}) \right\| \le \frac{1}{M} \sum_{m=1}^{M} \|\nabla^2 F_m(\boldsymbol{w})\| \le \frac{1}{M} \sum_{m=1}^{M} F_m(\boldsymbol{w}) = F(\boldsymbol{w}). \tag{583}$$

$\square$

**Lemma 26.** *Suppose* $\boldsymbol{w} \in \mathbb{R}^d$ *such that* $y_{mi}\langle \boldsymbol{x}_{mi}, \boldsymbol{w} \rangle \ge 0$ *for all* $m \in [M], i \in [n]$.

$$\|\nabla F(\boldsymbol{w})\| \ge \frac{\gamma}{2} F(\boldsymbol{w}), \tag{584}$$

*where* $\gamma$ *denotes the maximum margin of the combined dataset.*

*Proof.* Recall that $\boldsymbol{w}_*$ is the maximum margin classifier of the combined dataset, so $y_{mi}\langle \boldsymbol{w}_*, \boldsymbol{x}_{mi} \rangle \ge \gamma$ for all $m \in [M]$ and $i \in [n]$. From the definitions of $L_2$ norm and inner product, we have for any $\boldsymbol{z} \in \mathbb{R}^d$:

$$\|\nabla F(\boldsymbol{w})\| = \left\langle \nabla F(\boldsymbol{w}), \frac{\nabla F(\boldsymbol{w})}{\|\nabla F(\boldsymbol{w})\|} \right\rangle \ge \left\langle \nabla F(\boldsymbol{w}), \frac{\boldsymbol{z}}{\|\boldsymbol{z}\|} \right\rangle \tag{585}$$

In particular, choosing $\boldsymbol{z} = \boldsymbol{w}_*$ yields

$$\|\nabla F(\boldsymbol{w})\| \ge \langle \nabla F(\boldsymbol{w}), \boldsymbol{w}_* \rangle \tag{586}$$

$$= \frac{1}{Mn} \sum_{m=1}^{M} \sum_{i=1}^{n} |\ell'(y_{mi}\langle \boldsymbol{w}, \boldsymbol{x}_{mi} \rangle)| y_{mi} \langle \boldsymbol{x}_{mi}, \boldsymbol{w}_* \rangle \tag{587}$$

$$\overset{(i)}{\ge} \frac{\gamma}{Mn} \sum_{m=1}^{M} \sum_{i=1}^{n} |\ell'(y_{mi}\langle \boldsymbol{w}, \boldsymbol{x}_{mi} \rangle)| \tag{588}$$

$$\overset{(ii)}{\ge} \frac{\gamma}{2Mn} \sum_{m=1}^{M} \sum_{i=1}^{n} \ell(y_{mi}\langle \boldsymbol{w}, \boldsymbol{x}_{mi} \rangle) \tag{589}$$

$$= \frac{\gamma}{2} F(\boldsymbol{w}), \tag{590}$$

where $(i)$ uses the definition of $\boldsymbol{w}_*$ and $(ii)$ uses Equation 560 together with the condition $y_i\langle \boldsymbol{w}, \boldsymbol{x}_{mi} \rangle \ge 0$. $\square$

**Lemma 27.** *Suppose* $f : [0, \infty) \to \mathbb{R}$ *is continuously differentiable and*

$$f'(t) < \phi_1(t) + \phi_2(t)f(t), \tag{591}$$

*where* $\phi_1, \phi_2 : [0, \infty) \to [0, \infty)$ *are continuous and* $\phi_2(t) > 0$ *when* $t > 0$. *Then*

$$f(t) \le \exp\left( \int_0^t \phi_2(s)ds \right) \left( f(0) + \int_0^t \exp\left( -\int_0^s \phi_2(r)dr \right) \phi_1(s)ds \right), \tag{592}$$

*and consequently*

$$f'(t) \le \phi_1(t) + \phi_2(t) \exp\left( \int_0^t \phi_2(s)ds \right) \left( f(0) + \int_0^t \exp\left( -\int_0^s \phi_2(r)dr \right) \phi_1(s)ds \right). \tag{593}$$

*Proof.* Let $g : [0, \infty) \to \mathbb{R}$ be the unique solution to the following initial value problem:

$$g'(t) = \phi_1(t) + \phi_2(t)g(t) \tag{594}$$

$$g(0) = f(0), \tag{595}$$

and let $h(t) = g(t) - f(t)$. Then

$$h'(t) = g'(t) - f'(t) \tag{596}$$

$$> (\phi_1(t) + \phi_2(t)g(t)) - (\phi_1(t) + \phi_2(t)f(t)) \tag{597}$$

$$= \phi_2(t)(g(t) - f(t)) \tag{598}$$

$$= \phi_2(t)h(t). \tag{599}$$

So $h'(0) > 0$. Note that $h$ is continuously differentiable, since both $f, g$ are. Therefore there exists some $t_0 > 0$ such that $h'(t) > 0$ for all $t \in [0, t_0]$, and consequently $h(t) > 0$ for all $t \in [0, t_0]$.

Now assume for the sake of contradiction that $h(t_1) \leq 0$ for some $t_1 > 0$. Then let $T = \{t \geq 0 : h(t) \leq 0\}$. $T$ is not empty, since $t_1 \in T$. So $t_2 := \inf T$ exists. Since $h(t) > 0$ for all $t \in [0, t_0]$, we know that $t_2 > t_0 > 0$. Therefore

$$h(t_2) = \int_0^{t_2} h'(t)dt \overset{(i)}{=} \int_0^{t_2} \phi_2(t)h(t)dt \overset{(ii)}{>} 0, \tag{600}$$

where $(i)$ uses Equation 599 and $(ii)$ uses $t < t_2 \implies t \notin T$ together with $t_2 > 0$ and $t > 0 \implies \phi_2(t) > 0$. Therefore

$$h'(t_2) = \phi_2(t_2)h(t_2) > 0. \tag{601}$$

But since $h$ is continuously differentiable, there exists some $t_3 > t_2$ such that $h'(t) > 0$ for all $t \in [t_2, t_3]$. Then $h'(t) > 0$ for all $t \in [0, t_3]$, so $t_3 \leq \inf T = t_2$, but this contradicts the construction of $t_3 > t_2$. Therefore, $h(t) > 0$ for all $t > 0$. This means $f(t) < g(t)$ for all $t > 0$, and in particular that $f(t) \leq g(t)$ for all $t$.

It only remains to solve for $g$ in terms of $\phi_1, \phi_2$, which is a standard exercise in ordinary differential equations. We include the solution here for completeness. We know that

$$g'(s) - \phi_2(s)g(s) = \phi_1(s), \tag{602}$$

so multiplying by an "integrating factor",

$$\exp\left(-\int_0^s \phi_2(r)dr\right) g'(s) - \exp\left(-\int_0^s \phi_2(r)dr\right) \phi_2(s)g(s) = \exp\left(-\int_0^s \phi_2(r)dr\right) \phi_1(s) \tag{603}$$

$$\left(\exp\left(-\int_0^s \phi_2(r)dr\right) g(s)\right)' = \exp\left(-\int_0^s \phi_2(r)dr\right) \phi_1(s). \tag{604}$$

Integrating from $s = 0$ to $s = t$,

$$\exp\left(-\int_0^t \phi_2(r)dr\right) g(t) - \exp(0)g(0) = \int_0^t \exp\left(-\int_0^s \phi_2(r)dr\right) \phi_1(s)ds, \tag{605}$$

so

$$g(t) = \exp\left(\int_0^t \phi_2(r)dr\right) \left(f(0) + \int_0^t \exp\left(-\int_0^s \phi_2(r)dr\right) \phi_1(s)ds\right). \tag{606}$$

Since $f(t) \leq g(t)$, this proves Equation 592. Equation 593 follows by combining Equation 591 with Equation 592. $\qquad\square$

**Lemma 28.** *For any $a > 0$ and $n > 0$, if*

$$x \geq \max\left\{2, \frac{(2n)^n a}{n} \log^n(n^n a)\right\}, \tag{607}$$

*then*

$$\frac{x}{\log^n x} \geq a. \tag{608}$$

*Proof.* The desired inequality is equivalent to

$$\frac{x^{1/n}}{\log x} \geq a^{1/n} \tag{609}$$

$$\frac{x^{1/n}}{n \log x^{1/n}} \geq a^{1/n} \tag{610}$$

$$\frac{x^{1/n}}{\log x^{1/n}} \geq n a^{1/n}. \tag{611}$$

So denoting $y = x^{1/n}$ and $b = na^{1/n}$, we want to show that

$$\frac{y}{\log y} \geq b. \tag{612}$$

From the definition of $y$ and $b$,

$$y = x^{1/n} = \max\left\{2^{1/n}, 2na^{1/n}\log(na^{1/n})\right\} = \max\left\{2^{1/n}, 2b\log(b)\right\}. \tag{613}$$

We consider two cases depending on the magnitude of $b$. If $b \leq e$, then we are done, since $z/\log(z) \geq e$ for every $z > 1$, so that $y/\log(y) \geq b$. Otherwise, $2b\log(b) \geq 2e \geq 2^{1/n}$, so that the second term of the $\max$ in the definition is larger, i.e. $y = 2b\log(b)$. Therefore

$$\frac{y}{\log y} = \frac{2b\log(b)}{\log(2b\log(b))} = \frac{2b\log(b)}{\log(b) + \log(2\log(b))} \overset{(i)}{\geq} \frac{2b\log(b)}{2\log(b)} = b, \tag{614}$$

where $(i)$ used $\log(b) > 0$ together with $\forall z : \log(z) \leq z/2$ to show $\log(2\log(b)) \leq \log(b)$. This proves Equation 612 in the second case, so that it always holds. $\square$

### D.2 Lemmas for Section 5/Theorem 2

Recall from Section 5 the definition

$$\Phi(b, x) := \frac{W(\exp(b + \exp(x) + x))}{\exp(x)}, \tag{615}$$

where $W(x)$ denotes the principal branch of the Lambert W function, i.e. the unique solution in $z$ to

$$z\exp(z) = x, \tag{616}$$

for $x \geq 0$. Notice that $W(x) > 0$ whenever $x > 0$.

Throughout this section, for a fixed $b > 0$, we will denote

$$\psi(x) := \Phi(b, \log(1/x)) = xW(\exp(b + 1/x + \log(1/x))). \tag{617}$$

**Lemma 29.** *For every $x > 0$:*

*(a)* $W(x) > 0$.

*(b)* $W'(x) = \frac{W(x)}{x(1+W(x))}$.

*(c)* $\Phi(b, x) > 1$.

*(d)* $\psi(x) > 1$.

*Proof.* **(a)** $W(x)\exp(W(x)) = x > 0$, so $W(x) > 0$.

**(b)** This is a well-known property of the Lambert W function, which can be shown by implicitly differentiating the definition of $W$:

$$W(x)\exp(W(x)) = x \tag{618}$$

$$W(x)\exp(W(x))W'(x) + W'(x)\exp(W(x)) = 1 \tag{619}$$

$$xW'(x) + W'(x)\frac{x}{W(x)} = 1 \tag{620}$$

$$W'(x)x\left(1 + 1/W(x)\right) = 1 \tag{621}$$

$$W'(x) = \frac{W(x)}{x(1 + W(x))}. \tag{622}$$

**(c)** Denote $y = \exp(x)$ and $w = W(\exp(b + x + \exp(x)))$. Then

$$w\exp(w) = \exp(b + x + \exp(x)) \tag{623}$$

$$w + \log w = b + x + \exp(x) \tag{624}$$

$$w + \log w = b + y + \log y \tag{625}$$

$$w + \log w > y + \log y. \tag{626}$$

Since $f(x) = x + \log x$ is monotonic, the above means that $w > y$. So

$$\Phi(b, x) = \frac{W(\exp(b + x + \exp(x)))}{\exp(x)} = \frac{w}{y} > 1. \tag{627}$$

**(d)** $\psi(x) = \Phi(b, \log(1/x)) > 1$. $\qquad\qquad\qquad\qquad\qquad\qquad\qquad\qquad\square$

The following lemma is a well-known property of the Lambert W function. We include it here for completeness.

**Lemma 30.** *For every $b > 0$, $\Phi(b, x)$ is strictly decreasing in $x$.*

*Proof.* For any $b > 0$, to show that $\Phi(b, \cdot)$ is decreasing, it suffices to show that $\psi$ is increasing, since $\psi(x) = \Phi(b, \log(1/x))$. Also, denote $z(x) = b + 1/x - \log(x)$. Then

$$\psi(x) = xW(\exp(b + 1/x + \log(1/x))) = xW(\exp(z(x))), \tag{628}$$

so

$$\psi'(x) = W(\exp(z(x))) + xW'(\exp(z(x)))\exp(z(x))z'(x) \tag{629}$$

$$\overset{(i)}{=} W(\exp(z(x))) + x\frac{W(\exp(z(x)))}{\exp(z(x))(1 + W(\exp(z(x))))}\exp(z(x))\left(\frac{-1}{x^2} - \frac{1}{x}\right) \tag{630}$$

$$= W(\exp(z(x)))\left(1 - \frac{1}{1 + W(\exp(z(x)))}\left(\frac{1}{x} + 1\right)\right) \tag{631}$$

$$= \frac{W(\exp(z(x)))}{1 + W(\exp(z(x)))}\left(W(\exp(z(x))) - 1/x\right) \tag{632}$$

$$= \frac{W(\exp(z(x)))}{x(1 + W(\exp(z(x))))}\left(xW(\exp(z(x))) - 1\right) \tag{633}$$

$$= \frac{W(\exp(z(x)))}{x(1 + W(\exp(z(x))))}\left(\psi(x) - 1\right) \tag{634}$$

$$\overset{(ii)}{>} 0. \tag{635}$$

where $(i)$ uses Lemma 29(b), and $(ii)$ uses Lemma 29(a) and 29(d). $\qquad\qquad\square$

**Lemma 31.** *If $\Phi(b, x) \leq 1 + \frac{b}{b+2}$, then $x \geq \log(1 + b)$.*

*Proof.* By Lemma 30, $\Phi(b, x)$ is decreasing in $x$. So to prove the lemma, it suffices to show that $\Phi(b, \log(1 + b)) \geq 1 + b/(b + 2)$, since then

$$\Phi(b, x) \leq 1 + \frac{b}{b + 2} \implies \Phi(b, x) \leq \Phi(b, \log(1 + b)) \implies x \geq \log(1 + b). \tag{636}$$

From the definition of $\Phi$,

$$\Phi(b, \log(1 + b)) = \frac{W(\exp(b + (1 + b) + \log(1 + b)))}{1 + b} = \frac{W(\exp(1 + 2b + \log(1 + b)))}{1 + b}. \tag{637}$$

Let $z = W(\exp(1 + 2b + \log(1 + b)))$. Then by the definition of $W$,

$$z\exp(z) = \exp(1 + 2b + \log(1 + b)) \tag{638}$$

$$z + \log(z) = 1 + 2b + \log(1 + b). \tag{639}$$

Denoting $f(x) = x + \log(x)$, this means

$$f(z) = 1 + 2b + \log(1 + b) = b + f(1 + b). \tag{640}$$

By the concavity of $f$,

$$f(z) \leq f(1 + b) + (z - (1 + b))f'(1 + b) \tag{641}$$

$$z \geq (1 + b) + \frac{f(z) - f(1 + b)}{f'(1 + b)} \tag{642}$$

$$z \geq (1 + b) + \frac{b}{1 + 1/(1 + b)} = (1 + b) + (1 + b)\frac{b}{b + 2} = (1 + b)\left(1 + \frac{b}{b + 2}\right). \tag{643}$$

Plugging $z > 1 + b$ into Equation 637 yields

$$\Phi(b, \log(1 + b)) = \frac{z}{1 + b} > 1 + \frac{b}{b + 2}. \tag{644}$$

$\square$

**Lemma 32.** *If $x \geq \log(1 + b)$, then $\Phi(b, x) \geq \sqrt{1 + \frac{b}{\exp(x)}}$.*

*Proof.* Let $x \geq \log(1 + b)$, and denote $z = W(\exp(b + x + \exp(x)))$ and $y = \exp(x)$. Then $\Phi(b, x) = z/y$, so the statement we want to prove is

$$\frac{z}{y} \geq \sqrt{1 + \frac{b}{y}} \tag{645}$$

$$\log z - \log y \geq \frac{1}{2} \log\left(1 + \frac{b}{y}\right) \tag{646}$$

$$\log z \geq \log y + \frac{1}{2} \log\left(1 + \frac{b}{y}\right) \tag{647}$$

$$z + \log z \geq z + \log y + \frac{1}{2} \log\left(1 + \frac{b}{y}\right). \tag{648}$$

From the definition of $z$,

$$z \exp(z) = \exp(b + x + \exp(x)) \tag{649}$$
$$z + \log(z) = b + x + \exp(x) \tag{650}$$
$$z + \log(z) = b + y + \log(y), \tag{651}$$

so Equation 648 can be rewritten as

$$b + y + \log y \geq z + \log y + \frac{1}{2} \log\left(1 + \frac{b}{y}\right) \tag{652}$$

$$z \leq b + y - \frac{1}{2} \log\left(1 + \frac{b}{y}\right). \tag{653}$$

All steps above are reversible, so Equation 653 is equivalent to the desired result.

Define $f(x) = x + \log(x)$. Then $f$ is concave, so

$$f(y) \leq f(z) + (y - z)f'(z) \tag{654}$$
$$y + \log(y) \leq z + \log(z) + (y - z)(1 + 1/z) \tag{655}$$
$$(z - y)\frac{z + 1}{z} \leq z + \log(z) - y - \log(y) \tag{656}$$
$$(z - y)\frac{z + 1}{z} \leq b \tag{657}$$
$$z \leq y + \frac{bz}{z + 1} \tag{658}$$
$$z \leq y + b - \frac{b}{z + 1}. \tag{659}$$

Also, the condition $x \geq \log(1 + b)$ implies $b \leq y - 1$. Therefore, Equation 659 implies

$$z \leq y + b \leq 2y - 1, \tag{660}$$

so $z + 1 \leq 2y$. Again from Equation 659,

$$z \leq y + b - \frac{b}{z + 1} \tag{661}$$
$$\leq y + b - \frac{b}{2y} \tag{662}$$
$$\overset{(i)}{\leq} y + b - \frac{1}{2} \log\left(1 + \frac{b}{y}\right). \tag{663}$$

where $(i)$ uses $\log(1 + x) \leq x$. This is exactly Equation 653. $\square$

**Lemma 33.** $\psi$ *(defined in Equation 617) is concave.*

*Proof.* We will show that $\psi''(x) < 0$ for every $x$. Denoting $z(x) = b + 1/x + \log(1/x)$, we have from Equation 631:

$$\psi'(x) = W(\exp(z(x))) \left( 1 - \frac{1 + 1/x}{1 + W(\exp(z(x)))} \right) \tag{664}$$

$$= W(\exp(z(x))) \left( 1 - \frac{x + 1}{x + xW(\exp(z(x)))} \right) \tag{665}$$

$$= W(\exp(z(x))) \left( 1 - \frac{x + 1}{x + \psi(x)} \right). \tag{666}$$

Therefore, differentiating again yields

$$\psi''(x) = W(\exp(z(x))) \left( -\frac{(x + \psi(x)) - (x + 1)(1 + \psi'(x))}{(x + \psi(x))^2} \right) \tag{667}$$

$$+ \underbrace{W'(\exp(z(x))) \exp(z(x)) z'(x) \left( 1 - \frac{x + 1}{x + \psi(x)} \right)}_{A_1}. \tag{668}$$

The term $A_1$ above can be simplified as:

$$A_1 = \frac{W(\exp(z(x)))}{\exp(z(x))(1 + W(\exp(z(x))))} \exp(z(x)) \left( \frac{-1}{x^2} - \frac{1}{x} \right) \left( 1 - \frac{x + 1}{x + \psi(x)} \right) \tag{669}$$

$$= -\frac{W(\exp(z(x)))}{1 + W(\exp(z(x)))} \frac{x + 1}{x^2} \frac{\psi(x) - 1}{x + \psi(x)} \tag{670}$$

$$= -\frac{W(\exp(z(x)))}{x + xW(\exp(z(x)))} (1 + 1/x) \frac{\psi(x) - 1}{x + \psi(x)} \tag{671}$$

$$= -\frac{W(\exp(z(x)))}{(x + \psi(x))^2} (1 + 1/x) (\psi(x) - 1), \tag{672}$$

so

$$\psi''(x) = -\frac{W(\exp(z(x)))}{(x + \psi(x))^2} \left( \underbrace{(x + \psi(x)) - (x + 1)(1 + \psi'(x)) + (1 + 1/x)(\psi(x) - 1)}_{A_2} \right). \tag{673}$$

Recall that $W(y) > 0$ whenever $y > 0$ (Lemma 29(a)), so $\text{sign}(\psi''(x)) = -\text{sign}(A_2)$. To simplify $A_2$, we can rewrite $\psi'(x)$ (starting from Equation 666) as:

$$\psi'(x) = W(\exp(z(x))) \left( 1 - \frac{x + 1}{x + \psi(x)} \right) \tag{674}$$

$$\psi'(x) = \frac{\psi(x)}{x} \left( 1 - \frac{x + 1}{x + \psi(x)} \right) \tag{675}$$

$$\psi'(x) = \frac{\psi(x)}{x} \frac{\psi(x) - 1}{x + \psi(x)} \tag{676}$$

so

$$(x + 1)(1 + \psi'(x)) = (x + 1) \left( 1 + \frac{\psi(x)}{x} \frac{\psi(x) - 1}{x + \psi(x)} \right) \tag{677}$$

$$= (x + 1) + (1 + 1/x) \frac{\psi(x)(\psi(x) - 1)}{\psi(x) + x}. \tag{678}$$

Therefore, $A_2$ can be rewritten as

$$A_2 = (x + \psi(x)) - \left( (x+1) + (1+1/x) \frac{\psi(x)(\psi(x)-1)}{\psi(x)+x} \right) + (1+1/x)(\psi(x)-1) \quad (679)$$

$$= (\psi(x) - 1) - (1+1/x) \frac{\psi(x)(\psi(x)-1)}{\psi(x)+x} + (1+1/x)(\psi(x)-1) \quad (680)$$

$$= (\psi(x) - 1) + (\psi(x)-1)(1+1/x) \left( -\frac{\psi(x)}{\psi(x)+x} + 1 \right) \quad (681)$$

$$= (\psi(x) - 1) + (\psi(x)-1)(1+1/x) \frac{x}{\psi(x)+x} \quad (682)$$

$$= (\psi(x) - 1) + (\psi(x)-1) \frac{1+x}{\psi(x)+x} \quad (683)$$

$$= (\psi(x) - 1) \left( 1 + \frac{1+x}{\psi(x)+x} \right) \quad (684)$$

$$\overset{(i)}{>} 0, \quad (685)$$

where $(i)$ uses Lemma 29(d). Plugging back to Equation 673, this shows that $\psi''(x) < 0$, so $\psi$ is concave. $\qquad\square$

**Lemma 34.** *For every $x \in \mathbb{R}$, $b > 0$, and $a \in \mathbb{R}$:*

$$\Phi(b, x+a) \le \Phi(b,x) \left( 1 + (\exp(-a) - 1) \frac{\Phi(b,x)-1}{\Phi(b,x)+\exp(-x)} \right). \quad (686)$$

*In particular, if $a < 0$, then*

$$\Phi(b, x+a) \le \Phi(b,x) \exp(-a). \quad (687)$$

*Proof.* The idea of this proof is to leverage the concavity of $\psi$ to upper bound $\psi$ by its tangent line at $x$, then convert this to an upper bound of $\Phi$.

Since $\psi$ is concave (Lemma 33),

$$\psi(v) \le \psi(u) + (v-u)\psi'(u) \quad (688)$$

$$\overset{(i)}{=} \psi(u) + (v-u) \frac{\psi(u)}{u} \frac{\psi(u)-1}{u+\psi(u)} \quad (689)$$

$$= \psi(u) \left( 1 + \left( \frac{v}{u} - 1 \right) \frac{\psi(u)-1}{\psi(u)+u} \right) \quad (690)$$

$$\quad (691)$$

where $(i)$ uses Equation 676. The definition $\psi(x) = \Phi(x, -\log(x))$ implies

$$\Phi(b, -\log(v)) \le \Phi(b, -\log(u)) \left( 1 + \left( \frac{v}{u} - 1 \right) \frac{\Phi(b, -\log(u))-1}{\Phi(b, -\log(u))+u} \right). \quad (692)$$

Choosing $u = \exp(-x)$ and $v = \exp(-x-a)$:

$$\Phi(b, x+a) \le \Phi(b,x) \left( 1 + (\exp(-a)-1) \frac{\Phi(b,x)-1}{\Phi(b,x)+\exp(-x)} \right). \quad (693)$$

This proves Equation 686.

In the case that $a < 0$, $\exp(-a) - 1 > 0$. Also, $\Phi(b,x) - 1 > 0$ from Lemma 29(c). So by Equation 686,

$$\Phi(b, x+a) \le \Phi(b,x)(1 + (\exp(-a)-1)) = \Phi(b,x)\exp(-a). \quad (694)$$

This proves Equation 687 $\qquad\square$

### D.3 LEMMAS FOR WORST-CASE BASELINES

**Lemma 35.** *For a convex and $H$-smooth function $F$ and any $\boldsymbol{x}, \boldsymbol{y} \in dom(F)$,*

$$\|\nabla F(\boldsymbol{x}) - \nabla F(\boldsymbol{y})\|^2 \leq 2H(F(\boldsymbol{x}) - F(\boldsymbol{y}) - \langle \boldsymbol{x} - \boldsymbol{y}, \nabla F(\boldsymbol{y}) \rangle). \tag{695}$$

**Lemma 36.** *For a convex and $H$-smooth function $F$, any $\boldsymbol{x}, \boldsymbol{y} \in dom(F)$ and any $0 < \eta \leq \frac{2}{H}$,*

$$\|(\boldsymbol{x} - \eta \nabla F(\boldsymbol{x})) - (\boldsymbol{y} - \eta \nabla F(\boldsymbol{y}))\| \leq \|\boldsymbol{x} - \boldsymbol{y}\|. \tag{696}$$

## E ADDITIONAL EXPERIMENTAL DETAILS

### E.1 SYNTHETIC DATASET

The dataset consists of only two data points $\boldsymbol{x}_1, \boldsymbol{x}_2 \in \mathbb{R}^2$ (one for each client). For parameters $\delta > 0$ and $g \in [1, \infty)$, let

$$\boldsymbol{w}_1^* = \left( \frac{1}{\sqrt{1 + \delta^2}}, \frac{\delta}{\sqrt{1 + \delta^2}} \right) \tag{697}$$

$$\boldsymbol{w}_2^* = \left( -\frac{1}{\sqrt{1 + \delta^2}}, \frac{\delta}{\sqrt{1 + \delta^2}} \right). \tag{698}$$

and $\gamma_1 = 1, \gamma_2 = 1/g$. We then define $\boldsymbol{x}_m = \gamma_m \boldsymbol{w}_m^*$ and $y_m = 1$. Then in the notation of Section 5, this dataset has

$$c = \langle \boldsymbol{w}_1^*, \boldsymbol{w}_2^* \rangle = \frac{\delta^2 - 1}{\delta^2 + 1} \tag{699}$$

and $\gamma_{\max}/\gamma_{\min} = g$. So as $\delta \to 0$ and $g \to \infty$, we should expect that the negative effect of heterogeneity on optimization efficiency becomes worse and worse. For our experiments, we set $\delta = 0.1$ and $g = 5$.

### E.2 MNIST DATASET

Similarly to previous work on GD for logistic regression (Wu et al., 2024b;a), we use a subset of 1000 images from MNIST. For our distributed setting, we partition the data into $M = 5$ client datasets with $n = 200$ data points each. This partitioning is done according to the protocol used by Karimireddy et al. (2020), where $s\%$ of each local dataset is allocated uniformly at random from the 1000 images, and the remaining $(1 - s)\%$ is allocated to each client in order from a subset of data that is sorted by label. This has the effect that, when $s$ is small, the majority of each client's dataset has a small number of labels. For our dataset with 10 digits and $M = 5$ clients, we set $s = 5\%$, so that $95\%$ of each local dataset contains data for only two digits.

Note that we binarize this classification problem, so that the model is trained to predict whether a given image depicts an even digit or an odd digit. However, the heterogeneity partitioning protocol above is performed before replacing class labels. This means that each client has roughly the same label distribution (about half of examples have label 0, half have label 1), but very different feature distributions.

According to this protocol, client 1 will have $42.5\%$ of its data be images of the digit zero, $42.5\%$ of its data be images of the digit one, and $5\%$ of its data have uniform probability of being any digit from 0 to 9. Again, the labels for each client are either 0 or 1, according to whether the depicted digit is even or odd.

### E.3 TWO-STAGE STEPSIZE

To choose the number of rounds $r_0$, in the first stage of the two-stage stepsize schedule, we follow the requirement from Theorem 1 that $r_0$ scale linearly with $K$. We therefore set $r_0 = \lfloor \lambda K \rfloor$ and tune $\lambda \in \{2^{-4}, 2^{-3}, 2^{-2}, 2^{-1}, 2^0, 2^1, 2^2, 2^3\}$. The final value of $\lambda$ is selected by choosing the smallest value which ensures that the transition to a larger learning rate does not cause the training loss to increase above its value at initialization for any value of $K$.

The final tuned values of $\lambda$ are $\lambda = 4$ for the synthetic experiment and $\lambda = 1/16$ for the MNIST experiment. The larger value of $\lambda$ for synthetic data aligns with the fact that the synthetic data is designed to be highly heterogeneous, and generally requires smaller local model updates in order to avoid increases in the objective due to model averaging.

Because $\lambda = 4$ for the synthetic experiment, the training run with $K = 1024$ does not enter the second stage during the $R = 2048$ rounds used for training.

## F  DEEP LEARNING EXPERIMENTS

In this section, we provide additional experiments to compare Local SGD and Minibatch SGD for training deep neural networks, which lies outside of the theoretical scope considered in this paper. The purpose of these experiments is to verify the motivating claims from Sections 1 and 7 that in practice (1) Local SGD outperforms Minibatch SGD, and (2) Local SGD can converge faster by increasing the number of local steps $K$.

**Setup**  We train a ResNet-50 (He et al., 2016) for image classification on a distributed version of the CIFAR-10 dataset, using cross-entropy loss. For both algorithms, we train for $R = 1500$ communication rounds while varying the number of local steps $K \in \{1, 2, 4, 8, 16\}$. We split CIFAR-10 into $M = 8$ client datasets according to the same data heterogeneity protocol as we used for MNIST (see Section E.2) with data similarity $s = 50\%$. Unlike the previous MNIST setting, for this experiment we keep the original 10-way labels of the CIFAR-10 dataset.

For both algorithms, we tune the initial learning rate $\eta$ with grid search over $\{0.003, 0.01, 0.03, 0.1, 0.3, 1.0\}$ by choosing the value that achieved the smallest training loss after $R = 150$ training rounds with $K = 4$. We reuse this tuned value for all settings of $K$. For both algorithms, the best choice was $\eta = 0.03$. We also applied learning rate decay by a factor of $0.5$ after 750 rounds, and again after 1125 rounds. Lastly, we use a batch size of 128 for each local gradient update.

Note that we do not use momentum, gradient clipping, or other bells and whistles not mentioned here. Our goal is to methodically study the behavior of these two algorithms, not necessarily to achieve the smallest loss possible.

**Results**  Training loss and testing accuracy for both algorithms with all choices of $K$ are shown in Figure 2.

First, Local SGD is significantly faster when using a larger number of local steps $K$, up to a threshold. The final training loss of Local SGD improves steadily as $K$ increases from $K = 1$ to $K = 8$. When the number of local steps is large ($K = 16$), training becomes less stable, although the training loss is still smaller than that reached by every $K \leq 4$. These results suggest that our theoretical results about the optimization benefit of local steps also apply for scenarios beyond logistic regression.

Also, Local SGD significantly outperforms Minibatch SGD in this setting, which corroborates the often quoted folklore around these two algorithms Lin et al. (2019); Woodworth et al. (2020b); Wang et al. (2022); Patel et al. (2024). As $K$ increases, the training loss of Minibatch SGD is nearly unchanged; recall that changes to $K$ only affects the effective batch size of Minibatch SGD, but not the number of model updates. This underscores the gap between ML in practice and existing theory of distributed optimization algorithms. Local SGD is dominated by Minibatch SGD in many natural regimes Woodworth et al. (2020b); Patel et al. (2024), but this worst-case analysis does not seem representative of performance when training deep networks with real world data.

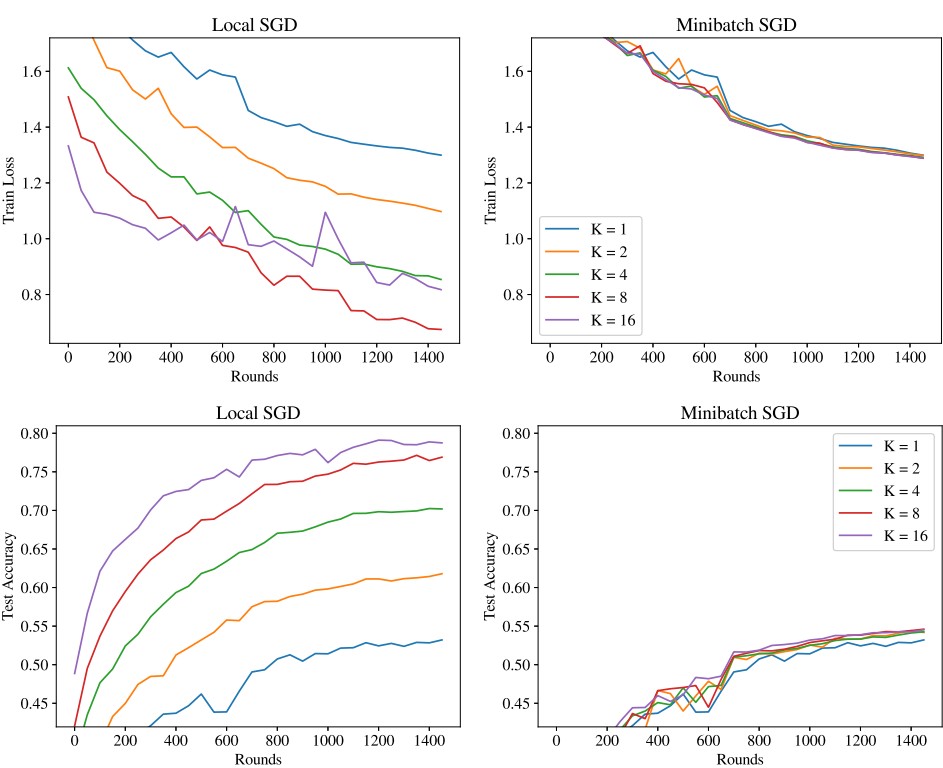

Figure 2: Train loss and testing accuracy for heterogeneous, distributed CIFAR-10 with ResNet-50.

