# OpenReview forum: "Local Steps Speed Up Local GD for Heterogeneous Distributed Logistic Regression"
_ICLR.cc/2025/Conference — ICLR 2025 Poster_

### Official Review · Reviewer_8QYH · 2024-10-21

**Soundness:** 2
**Presentation:** 2
**Contribution:** 2
**Rating:** 3
**Confidence:** 3

**Summary:**

For logistic regression, the paper proves convergence of local GD improves with number of local updates, based on a two-stage estimator (two stages with two different choice of step size)

**Strengths:**

The paper shows for a special case that the convergence rate is inversely proportional to number of local udpates

**Weaknesses:**

The analysis only applied to logistic regression, and only applies to linearly separately case (which is of course a simple case that usually is not satisfied, or unknown whether it is satisfied)

**Questions:**

1. The paper's presentation keeps switching between local GD and local SGD, this is very confusing why you use SGD
2. Is there any other loss whose derivative is bounded by loss?
3. is the rate optimal in some sense?

---

> ### Author Response · Authors · 2024-11-20
> **Response**
>
> Thank you for your efforts in the review process. Below we have addressed your points.
>
> **W: Individual objective** Our focus on a single objective is motivated by the fact that all previous analyses of Local GD that consider general objectives yield results that are not aligned with empirical observations. There are no known general conditions which imply that Local GD can be accelerated by taking more local steps, and finding such a condition may very well require a more narrow scope. We agree that discovering such a condition is desirable, but we believe that it is an important first step to show that acceleration by local steps can happen for any single problem, which has not previously been shown in the literature. This same approach has been taken by seminal works in theoretical ML, e.g. for implicit bias [(Soudry et al, 2018)](https://arxiv.org/abs/1710.10345), [(Ji \& Telgarsky, 2018)](https://arxiv.org/abs/1803.07300) and edge of stability [(Wu et al 2024a)](https://arxiv.org/abs/2402.15926), [(Wu et al, 2024b)](https://arxiv.org/abs/2305.11788). These works first focused on logistic regression with deterministic gradients, and we have done the same. We believe that our analysis in this paper may be generalized in future work.
>
> **Q1: SGD vs GD** Let us clarify when we consider both the stochastic and the deterministic setting. In Sections 1 and 7 (Introduction and Discussion), we consider previous results for Local SGD and Minibatch SGD in the stochastic setting, and compare these theoretical results to the experimental behavior of these algorithms observed in practice. We use this discrepancy between theory and practice to motivate the study of local steps in distributed optimization for problems arising in ML, as opposed to general optimization problems. We then focus on a single optimization problem, and for the rest of the paper (Sections 2-6), the only setting considered theoretically or empirically is the deterministic setting, where we consider Local GD. Overall, the stochastic setting is described in high-level motivation (Section 1 and 7), and the deterministic setting is used for all theory and experiments (Sections 2-6). We follow a previous line of work on logistic regression that first considers the deterministic setting [(Soudry et al, 2018)](https://arxiv.org/abs/1710.10345) [(Ji \& Telgarsky, 2018)](https://arxiv.org/abs/1803.07300), [(Gunasekar et al, 2018)](https://arxiv.org/abs/1802.08246), [(Wu et al 2024a)](https://arxiv.org/abs/2402.15926), [(Wu et al, 2024b)](https://arxiv.org/abs/2305.11788) as a first step towards understanding phenomenon that may also occur in the stochastic setting.
>
> **Q2: Derviative bounded by loss** To answer your question, the derivative can be bounded in terms of the loss for any smooth function: $$
>     \lVert \nabla F(x) \rVert^2 \leq 2L (F(x) - F_*),
> $$ which is a well-known property of smooth functions. However, this is not the key property used in our analysis. For our analysis, we relate the second derivative of the loss to the loss value: $\lVert \nabla^2 F(x) \rVert \leq F(x)$, which was also used in previous work on GD for logistic regression [(Wu et al 2024a)](https://arxiv.org/abs/2402.15926), [(Wu et al, 2024b)](https://arxiv.org/abs/2305.11788).
>
> One relevant concept in the literature is self-concordant objectives (mentioned also by other reviewer zs85), in which the third derivative can be bounded as a function of the second derivative (see [(Nesterov, 2013)](https://link.springer.com/book/10.1007/978-3-319-91578-4), [(Bach, 2014)](https://jmlr.org/papers/v15/bach14a.html)). The situation in our paper is not exactly the same, since we related the second derivative and the zeroth derivative of the loss. There may be some connection between the two (see our response to Reviewer zs85), but self-concordance is not directly used in our analysis. Investigating this potential connection is a larger question that will require further investigation.
>
> **Q3: Optimal rate** In one sense, our rate is better than optimal in that it *breaks the lower bounds* for Local GD [(Woodworth et al, 2020b)](https://arxiv.org/abs/2006.04735), [(Glasgow et al, 2022)](https://arxiv.org/abs/2111.03741), [(Patel et al, 2024)](https://arxiv.org/abs/2405.11667).  More specifically, our rate is better than the optimal rate of Local GD in the worst-case over the class of objectives considered in the previous lower bounds. On the other hand, it is open whether an even better analysis of Local GD could be achieved for this individual problem. Answering this question would require a lower bound for this problem, and to the best of our knowledge there do not exist any lower bounds for logistic regression with any algorithm in the distributed setting. In this work we focus first on providing upper bounds, and our result is the first time that a variant of Local GD is proven to exhibit acceleration by local steps, which is a fundamental question in distributed optimization.

---

> > ### Author Response · Authors · 2024-11-25
> >
> > Thank you again for reviewing our paper. We are just following up with a gentle reminder that the discussion period will end tomorrow. If you have any more questions or concerns please let us know. If we have addressed your concerns, we kindly ask that you reconsider your score.

---

### Official Review · Reviewer_FGN8 · 2024-11-01

**Soundness:** 3
**Presentation:** 3
**Contribution:** 3
**Rating:** 6
**Confidence:** 3

**Summary:**

The paper "Local Steps Speed up Local GD for Heterogeneous Distributed Logistic Regression" proposes a two phase variant of Local SGD for which the authors can establish that despite the presence of heterogeneity will asymptotically benefit from increasing the size of the local mini-batch. This is an improvement when compared to existing lower bounds for general convex functions as it exploits additional structure of the logistic regression problem.

**Strengths:**

1) The paper is very well written
2) The contribution is interesting as it provides a stepping stone to understand when local SGD can be used with clear benefits, despite the fact that for the classical set of families of functions studied we know it does not.
3) The authors do a good job at providing a sketch of the proof of the main result that manages to capture the idea behind the proof.

**Weaknesses:**

1) Limitation in scope: from this reviewer's perspective, the conditions under under which the bias introduced by heterogeneity could potentially apply to other problems (that is total loss controls the gradient and Hessian size). The condition of separability may potentially be replaced by an interpolation condition. What is the limitation that allows the work to focus exclusively on logistic regression?
2) Dependence on gamma: as gamma decreases the amount of time required within the first stage increases. This implies that a lower bound on gamma is required. A lower bound to it would have to be computed and shared across agents.

**Questions:**

See weaknesses.

---

> ### Author Response · Authors · 2024-11-20
> **Response**
>
> Thank you for your positive review and for the constructive comments. Below we have addressed the points in your review.
>
> **W1: Interpolation as a general assumption** This is a great question. As of now, it is not known what general conditions (if any) will imply the optimization benefit of local steps. One possibility you mentioned is an interpolation condition. In fact, the lower bound from [(Patel et al, 2024)](https://arxiv.org/abs/2405.11667) considers interpolation as a special case. They consider the setting of bounded heterogeneity at optima, that is, they consider problems satisfying $$
>     \frac{1}{M} \sum_{m=1}^M \lVert \nabla F_m(x_*) \rVert^2 \leq \zeta^2
> $$ (see Assumption 1 of [(Patel et al, 2024)](https://arxiv.org/abs/2405.11667)). In the convex setting, choosing $\zeta = 0$ essentially gives an interpolation condition, since it implies that all local clients have zero gradient (i.e. achieve their global minimum) at $x_*$. However, even under this condition, their lower bound *still* shows that Local SGD does not improve over Minibatch SGD, and the rate of Local SGD has dominating terms which do not improve with $K$ (see their Table 1, Local SGD Lower Bound). Note that these terms do not disappear even in the deterministic setting. This means that interpolation is not sufficient to achieve acceleration by local steps, unfortunately.
>
> Our choice to focus on an individual problem is motivated by the lack of known general conditions that could imply acceleration by local steps for Local GD or Local SGD. It is an important problem to identify general conditions under which this acceleration might hold, and we believe that identifying any problems where this happens is an important first step.
>
> **W2: Knowledge of $\gamma$** Thank you for pointing this out. We agree that choosing $r_0$ according to the guarantee of Theorem 1 requires knowledge of $\gamma$, which may not be easily computed.  However, we consider the classical setting in which problem parameters are known, which is often used in optimization to assume knowledge of smoothness constants, stochastic gradient variance, distance to solution, etc. This is a standard theoretical assumption [(Nesterov, 2013)](https://link.springer.com/book/10.1007/978-3-319-91578-4) [(Ghadimi \& Lan, 2013)](https://arxiv.org/abs/1309.5549). It may be possible to provide an adaptive guarantee that does not require knowledge of the parameter $\gamma$, as in parameter-free optimization [(Orabona \& Pal, 2016)](https://arxiv.org/abs/1602.04128), [(Ivgi et al, 2023)](https://arxiv.org/abs/2302.12022), but in this work we focus first on achieving optimization results assuming known parameters. Achieving adaptability to unknown problem parameters is an important direction for future work.

---

### Official Review · Reviewer_bszu · 2024-11-04

**Soundness:** 2
**Presentation:** 2
**Contribution:** 2
**Rating:** 8
**Confidence:** 4

**Summary:**

This paper considers distributed logistic regression and analyzes two variants of Local Gradient Descent, one with two-phases and a restart in the middle, and another (unimplementable) algorithm where the gradient flow is used as the "local optimizer" on each node before averaging. The main motivation for the paper is the longstanding open problem in distributed optimization: why do local methods (like Local GD) work well when the theoretical guarantees are outperformed by (accelerated) minibatch SGD in the worst case? By restricting specifically to distributed logistic regression, we can study this question in a specific setting.

**Strengths:**

1. I really like the idea of studying the performance of Local GD in the setting of distributed logistic regression. The paper's suggestion that the benefit of local methods might be dependent on the loss landscape rather than decreasing client dissimilarity is an interesting one, that I think might inspire much further work.

2. The paper is written clearly, and is well-organized.

My main issue with this paper is that I believe there might be an issue with the proof (see weaknesses section). For this reason, I can't really recommend acceptance.

**Weaknesses:**

My main issue with the paper is the proof issue mentioned below, which affects the paper's main theorem.

1. (Proof issue) When going from equation (296) to equation (297), you use the condition $\eta \leq 1/(4H)$ to bound $-(2\eta-4\eta^2 H) (F(\bar{w}_t) - F(u))$ by $-\eta (F(\bar{w}_t) - F(u))$. But you can only do this in case $F(\bar{w}_t) - F(u) \geq 0$. Here, $u$ is not necessarily the minimizer. Lemma 17 should hold for any $u$ but clearly this is not true. This issue is repeated when going from (352) to (353). Since the results of Theorem 1 relies on these corollaries, it is affected as well. I'm not really sure how to fix this issue. I guess you could restrict $u$ to only comes from the points that have a lower loss, but then the proofs of Corollaries 1 and 2 would have to be modified to accommodate this.

2. (Toy setting) The Local Gradient Flow guarantee is in a very toy setting, $M=2$ and $n=1$ plus an unimplementable gradient flow subroutine is in general very questionable as a model for anything realistic.

**Questions:**

1. Please address the proof issue mentioned in the weaknesses section.
2. Can we recover the same results you have by including some small regularization on the logistic regression problem, then use the fact that this problem has a minimizer?
3. (GD baseline) Shouldn't we compare not against the Local GD bounds, but also include the "trivial" GD/Minibatch SGD bound as a baseline? After all, the paper's hypothesis (that we can use larger stepsizes, hence the improved rate) would motivate us to compare against GD where we also use a warmup period.

---

> ### Author Response · Authors · 2024-11-20
> **Response**
>
> Thank you for the positive comments and for your scrutiny in examining our proofs. We have responded to our main points below.
>
> **W1: Proof bug** Thank you for finding this issue, which we acknowledge is a bug in our proof.  However, there is an extremely quick fix for this bug that only requires modifying a few lines (Equations 293-297), and this completely eliminates the issue. In the following, we use $w_t$ to denote the average model at iteration $t$, due to issues with inline Latex rendering our original notation. This fix is quite simple: notice that the problem term $F(w_t) - F(u)$ in Equation 295 originates by decomposing $$
>     4 \eta^2 H (F(w_t) - F_*) = 4 \eta^2 H (F(w_t) - F(u)) + 4 \eta^2 H (F(u) - F_*)
> $$ in Equation 293. After this decomposition is when our original proof (mistakenly) applies the bound $4 \eta H \leq 1$ to obtain $$
>     4 \eta^2 H (F(w_t) - F_*) \leq \eta (F(w_t) - F(u)) + \eta (F(w_t) - F(u).
> $$ To fix this issue, all that we have to do is switch the order of decomposition and application of $4 \eta H \leq 1$. So starting from Equation 293, we first use the $\eta$ bound: $$
>     4 \eta^2 H (F(w_t) - F_*) \leq \eta (F(w_t) - F_*),
> $$ and after this we perform the decomposition $$
>     \eta (F(w_t) - F_*) = \eta (F(w_t) - F(u)) + \eta (F(u) - F_*).
> $$ This allows us to exactly recover Equation 297 without any modifications.
>
> For the second instance (Equations 352-353), we do not see the same bug. There, we only apply the $\eta$ bound to coefficients of positive terms (e.g. $F(w_t) - F_*)$), so the same bug should not appear. Please let us know if you agree.
>
> In summary, we can completely fix the bug with only a small change to a couple of lines in the proof, and we can completely recover the results from the original submission. We have included this fix in the updated paper. We hope that with this fix, you will reconsider the score.
>
> **W2: Scope of Theorem 2** We agree that the scope of Theorem 2 is limited. This result is preliminary, and we decided to include it in order to lay the foundation for a possible future work that may tackle the general problem. Our main result is Theorem 1, which shows for the first time that a natural variant of Local GD achieves acceleration with local steps for an ML optimization problem.
>
> **Q2: Regularization** Since we have fixed the bug you pointed out, our original results stand, and we can achieve the acceleration by local steps even with the unregularized objective. We have chosen to focus on the unregularized problem following a line of work on logistic regression [(Soudry et al, 2018)](https://arxiv.org/abs/1710.10345), [(Ji \& Telgarsky, 2018)](https://arxiv.org/abs/1803.07300), [(Gunasekar et al, 2018)](https://arxiv.org/abs/1802.08246), [(Nacson et al, 2019)](https://proceedings.mlr.press/v89/nacson19a.html) [(Wu et al 2024a)](https://arxiv.org/abs/2402.15926), [(Wu et al, 2024b)](https://arxiv.org/abs/2305.11788).
>
> You are correct that adding regularization would imply the existence of a minimizer, placing us in the setting considered by previous works on distributed stochastic convex optimization [(Woodworth et al, 2020b)](https://arxiv.org/abs/2006.04735), [(Koloskova et al, 2020)](https://arxiv.org/abs/2003.10422), [(Glasgow et al, 2022)](https://arxiv.org/abs/2111.03741), [(Patel et al, 2024)](https://arxiv.org/abs/2405.11667). It's possible that we could achieve similar results for this regularized problem, but many details remain to be worked out to get a complete result. Notice that the regularization strength is a lower bound on the Hessian norm for the regularized objective, so we need to choose a small regularization parameter to enable the use of a large learning rate in the second stage. Of course, if the regularization strength is too small, then the convergence rate will slow down accordingly. Such a construction might be made to work, but we reiterate that we can already achieve our stated results for the unregularized problem.
>
> **Q3: Comparison with Minibatch SGD** This is an important point that requires clarification, and we have provided a full answer in the general response. Essentially, Minibatch SGD in the deterministic setting is equivalent to Local GD with $K=1$, which we have compared against. See the general response for a thorough discussion of this point.

---

> > ### Comment · Reviewer_bszu · 2024-11-23
> >
> > W1. Ok, I think this does fix the proof. I will raise my score accordingly-- I now lean towards acceptance.
> >
> > W2. "the first time that a natural variant of Local GD achieves acceleration with local steps for an ML optimization problem." I'm not really sure how this is true-- there are plenty of problems on which you can find Local GD performing well, the discussion in p. 7 does indeed show that for convex problems with limited heterogeneity, Local GD is optimal (just put $\sigma = \sigma_{\ast} = 0$ in their results to get Local GD).
> >
> > Q2. I strongly believe you should include a comparison with the version you obtain using regularization. For one, you consider a two-stage variant of Local GD that requires tuning an extra learning rate and also choosing a switching time ($r_0$)-- the regularized version of the algorithm would be a variant that requires only tuning one extra parameter (the regularization) and won't be two-stage. It is an important baseline.
> >
> > W3. You compare against Local GD with $K=1$ in your own analysis, but do we have a reason to believe that this is optimal? Since the smoothness constant is effectively decreasing over time, using something that adapts to it (e.g. the Polyak stepsize) should enable GD to converge faster on this problem over time. Do we have any lower bounds for first-order methods applied to this problem?
> >
> > [1] Woodworth, B. E., Patel, K. K., & Srebro, N. (2020). Minibatch vs local sgd for heterogeneous distributed learning. Advances in Neural Information Processing Systems, 33, 6281-6292.

---

> > > ### Author Response · Authors · 2024-11-24
> > >
> > > Thank you for the positive comments and interesting questions.
> > >
> > > **W2** We do not believe that [(Woodworth et al, 2020b)](https://arxiv.org/abs/2006.04735) demonstrates acceleration of Local GD by local steps for any non-degenerate cases. In [(Woodworth et al, 2020b)](https://arxiv.org/abs/2006.04735) with $\sigma = 0$, Local GD can exhibit faster convergence (in dominating terms) with $K$, but this acceleration requires vanishingly small heterogeneity (i.e. $\zeta \lesssim 1/R$ in their notation). There are two reasons why this setting is degenerate. First, for any fixed problem with $\zeta > 0$, the condition $\zeta \lesssim 1/R$ cannot hold for sufficiently small $\epsilon$ (target optimization accuracy). To achieve small $\epsilon$, the algorithm must use large $R$, so the heterogeneity condition $\zeta \lesssim 1/R$ eventually fails. Essentially, the acceleration by local steps does not hold for all $\epsilon > 0$. Second, when $\sigma = 0$ and $\zeta = 0$, the problem can be solved with **zero** communication cost. In this case, every client can access the gradient of the global objective, so there is no need for communication, and the distributed optimization problem is degenerate. For these two reasons, we don't believe that [(Woodworth et al, 2020b)](https://arxiv.org/abs/2006.04735) demonstrates acceleration of Local GD by local steps for any non-degenerate cases. We are not aware of any works that demonstrate this acceleration. If you know of any others, please let us know.
> > >
> > > **Q2**: We agree that a comparison with a regularized baseline is important, but we need to ask for clarification on this point. Are you referring to a baseline that would solve the unregularized problem by applying the analysis of [(Woodworth et al, 2020b)](https://arxiv.org/abs/2006.04735) to Local GD for the regularized problem, using a black-box reduction from convex problems to strongly convex problems? If so, we can definitely include this baseline, which achieves a similar complexity as Corollary 1. In particular, this baseline cannot show acceleration by local steps. Please let us know if this is what you refer to, and we will write it and include it in an appendix.
> > >
> > > **W3**: No, we do not believe that $K=1$ is optimal, but there are other important reasons why we compare against $K=1$. First, there are probably other ways to improve performance for this problem: adapting to local smoothness, using a multi-stage learning rate (as mentioned by Reviewer zs85), possibly using a very large learning rate (as in [(Wu et al, 2024a)](https://arxiv.org/abs/2402.15926)), or other algorithms that are specifically tailored for this problem. Indeed, we are not aware of any lower bounds for first-order methods for this problem. However, instead of achieving the optimal rate for this particular problem, we are primarily interested in developing theory for practically relevant algorithms that seem to work for many problems (e.g. fixed stepsize Local SGD). Faster rates for this particular problem can likely be achieved through various means, but for this work we focus on Local GD.
> > >
> > > Second, the reason that we compare against $K=1$ is not because $K=1$ is optimal, but because previous works [(Woodworth et al, 2020b)](https://arxiv.org/abs/2006.04735), [(Koloskova et al, 2020)](https://arxiv.org/abs/2003.10422) fail to show that $K > 1$ results in a faster rate for Local GD than $K = 1$, and we want to fill this gap. This gap can be seen in the baseline guarantees (Equations (3) and (4)), where $K$ does not appear in the dominating terms of the convergence rate. Yet, the improvement of Local GD with $K > 1$ is empirically observed, and theoretically explaining this improvement (for any setting at all) is an important problem, which we address by comparing $K > 1$ against $K = 1$.

---

> > > > ### Comment · Reviewer_bszu · 2024-11-24
> > > >
> > > > W2. I find this discussion a little puzzling. Yes, the setting in Woodworth et al. requires the heterogeneity to be quite small, but how is that different from a setting where the smoothness gets so small that we can essentially use an unbounded learning rate (i.e. the current setting)? Polyak's stepsize solves the problem in almost no time anyway, because of this property.
> > > >
> > > > Q2. Yes, I want the baseline you described, plus the baseline of just plain gradient descent (which should give us $1/R$ rather than $1/R^{2/3}$, I think). You don't have to do it right now, only for the accepted paper should it be accepted.
> > > >
> > > > W3. What I meant is that, the *analysis* of the $K=1$ itself is suboptimal, because gradient descent on this problem should do at least $1/R$ rather than $1/R^{2/3}$. I agree that the optimal $K$ may not be $1$, at least when considering constant or non-adaptive stepsizes.

---

> ### Author Response · Authors · 2024-11-24
>
> W2: There are a few reasons why we see our condition of small smoothness as different from the condition of small heterogeneity from (Woodworth et al, 2020b). First, in order to get acceleration from local steps for every $\epsilon > 0$, Woodworth requires that $\zeta$ and $\sigma$ are both **strictly equal** to $0$, which trivializes the distributed optimization problem (see our previous comment). Second, the condition $\zeta \lesssim 1/R$ enforces a condition on the objective function over the **entire domain** ($\zeta$ is the maximum gradient dissimilarity across the domain), whereas we only use the fact that the smoothness goes to zero along the algorithm's trajectory. Lastly, our condition of small smoothness arises naturally from a standard optimization problem, but we don't see any examples of natural objectives which satisfy the condition of near-zero heterogeneity from Woodworth. Even if we had an example satisfying $\zeta \lesssim 1/R$ for some fixed $R$, it does not exhibit acceleration by local steps for every $\epsilon > 0$.
>
> Also, we agree that Polyak's stepsize can solve the problem very quickly without taking local steps. However, we reiterate that our goal here is to investigate the behavior of Local GD with $K > 1$ for a practical problem, not necessarily to achieve the fastest possible rate for this individual problem. There are many ways of accelerating convergence for this problem, but we don't want to overfit to logistic regression: we aim to study the standard distributed optimization algorithm. Also, from the perspective of understanding the effect of local steps, it is not immediately clear how one might define and analyze a variant of Polyak's stepsize for Local GD with $K > 1$.
>
> Q2: Okay, thanks for the clarification. We can include this in a future version, should the paper be accepted. Also, we agree that GD should be included in the comparison.
>
> W3: Your observation is correct that the $1/R^{2/3}$ term appears to create a gap in the case $K=1$. However, the $1/R^{2/3}$ term from Woodworth's analysis actually disappears when $K=1$, although this is not clear from the usual presentation of that result. This means that the analysis is tight for the case $K=1$, since the resulting algorithm is just GD, whose upper and lower bounds match.

---

> > ### Comment · Reviewer_bszu · 2024-11-24
> >
> > W2. Well I disagree; First, the results of Woodworth can trivially be extended to require the heterogeneity to be bounded only in a ball around the minimum, and we can use projections to ensure we stay in that ball. Tiny smoothness also "trivializes the distributed optimization problem" because using GD with Polyak stepsize would solve the problem in a couple of steps. While the problem you're considering is natural, the regime in which you see benefits is very unnatural. We can just obtain the same result by, rather than doing a two-phase algorithm, do a couple of communication rounds with GD to get to the tiny smoothness regime and then converge in like 4 steps. This setting is no more natural than optimizing quadratics with small heterogeneity between them. As far as I understand, we're talking about a setting in logistic regression where we have essentially converged on the right direction and just need to keep making the weights larger and larger in that direction.
> >
> > W3. Your statement of the results of Local GD with $K=1$ and the subsequent discussion makes no indication that the second term vanishes at $K=1$. Neither does your proof, e.g. equation (499). This should be fixed.

---

> > > ### Author Response · Authors · 2024-11-25
> > >
> > > Thank you for your comments and continued discussions with us. We really appreciate it.
> > >
> > > **W2** For your comments about Woodworth's paper, we do not see how the requirement on $\zeta$ over the entire domain can be easily relaxed. For the projections you suggested, we do not see how to implement a projection into a ball around the minimum without assuming knowledge of the minimum. Please let us know if we have misunderstood your suggestion about relaxing the requirement on $\zeta$. Also, previous work already attempts to relax the requirement on $\zeta$ over the entire domain by only assuming gradient dissimilarity at a minimum, such as [(Koloskova et al, 2020)](https://arxiv.org/abs/2003.10422). The resulting upper bound loses a $K$ in the denominator of the convergence rate: compare Theorem 3 of [(Woodworth et al, 2020b)](https://arxiv.org/abs/2006.04735) with row 1, Table 1 of [(Patel et al, 2024)](https://arxiv.org/abs/2405.11667). Note that this upper bound using the relaxed condition on gradient dissimilarity is tight, since it matches the lower bound of [(Patel et al, 2024)](https://arxiv.org/abs/2405.11667). Therefore, we do not see any easy way to relax the condition of (Woodworth et al, 2020b) on the gradient dissimilarity over the entire domain, while retaining the $K$ dependence. Also, the setting you mentioned about quadratics with limited heterogeneity still does not exhibit acceleration by local steps for all $\epsilon > 0$ according to the analysis of (Woodworth et al, 2020b), unless all local objectives are identical up to an additive constant: the condition $\zeta \lesssim 1/R$ fails for sufficiently small $\epsilon$ if $\zeta > 0$. Please let us know if we have overlooked anything here.
> > >
> > > When we say that $\sigma = 0$ and $\zeta = 0$ trivializes the distributed optimization problem, we mean that any non-distributed algorithm can solve the problem without communicating at all. That is, we can just run any non-distributed algorithm locally on each client, which solves the problem with $R = 0$ communication cost. We agree that GD with Polyak stepsizes can also solve the logistic regression problem very quickly even with $\zeta > 0$, but not literally with $R = 0$ communication. Anyway, the superior rate of Polyak stepsizes is orthogonal to our goal of investigating the effect of local steps for Local GD. We do not claim that our results provide the best possible rate for this problem, but we do claim that our results are the first to show that Local GD can be accelerated by local steps for a problem with heterogeneous objectives (i.e. $\zeta > 0$). We can add to the paper a discussion that GD with Polyak stepsizes can leverage the decreasing smoothness to converge quickly.
> > >
> > > **W3**: We agree that the current presentation is confusing, and the $1/R$ rate for the case $K=1$ should be explicit. Thank you for pointing this out, we will add this.
> > >
> > > **Summary of changes** Thank you for the detailed discussion. From our conversation, we will make the following changes to the paper:
> > > - Additional comparisons: (1) Plain GD and (2) Worst-case Local GD analysis on regularized objective.
> > > - Fix the presentation of worst-case rate for Local GD, so $1/R^{2/3}$ term should disappear when $K=1$.
> > > - Add a discussion about the possible acceleration of Polyak stepsizes.

---

### Official Review · Reviewer_zs85 · 2024-11-04

**Soundness:** 4
**Presentation:** 3
**Contribution:** 3
**Rating:** 8
**Confidence:** 3

**Summary:**

This paper studies distributed logistic regression in the context of local gradient descent (GD) with heterogeneous, separable data.

It introduces two variants of Local GD: **Two-Stage Local GD** and **Local Gradient Flow (GF)**.

The authors demonstrate a faster convergence rate than prior work for Local GD with $O(1/(KR))$ when using large local steps $K$ and large communication rounds $R$, contrasting previous results that show convergence at $O(1/R)$.

The key insight is leveraging the logistic regression objective, which allows using a large step size $\eta \gg 1/K$ after a warmup period, instead of the conventional $\eta \leq 1/K$.

They support the theoretical findings with experimental results on synthetic and MNIST datasets.

**Strengths:**

1. **Novel Insight into Local Steps for Distributed Logistic Regression:**

   The paper identifies a specific problem where local updates accelerate convergence, contrasting with past worst-case complexity results for Local GD. Though It focuses on the logistic loss, the result is quite new and interesting.

2. **Clear Theoretical Contributions:**

   The convergence rates for Two-Stage Local GD and Local GF are supported with rigorous theoretical analysis and are novel improvements over existing results in the distributed logistic regression setting. The proof sketch also helps readers understand the key idea better.

3. **Experimental Validation:**
   The paper includes experiments on both synthetic and real datasets, demonstrating the behavior of the proposed algorithms under different stepsizes and local steps, which supports their claims empirically

**Weaknesses:**

1. **Lack of Generality:**

   While the theoretical results are interesting, the scope is limited to distributed logistic regression.

2. **Weak Experimental Setup:**
   The experiments focus primarily on very small synthetic and MNIST datasets, which do not fully reflect the complexity of real-world distributed learning scenarios.

3. **Unclear Justification of Baseline Comparisons:**

   The comparison to previous methods could be more thorough, particularly regarding the baselines. The paper briefly mentions the worst-case results for Local GD but does not provide detailed analysis or experiments to compare directly to existing algorithms like Minibatch SGD.

**Questions:**

1. **Extending to Self-Concordant Functions:**

   Could the analysis be extended to **self-concordant functions**? As introduced in [Francis Bach’s blog](https://francisbach.com/self-concordant-analysis-for-logistic-regression/), the self-concordant framework might offer additional insights. Some references are missing from the related work section, particularly the paper by **Francis Bach (2014)** on **adaptivity of stochastic gradient descent**. Including such references would strengthen the connection between your work and existing literature on logistic regression.

2. **Multiple Stages and Adaptive Stepsizes:**

   The paper discusses a **two-stage approach**, but have you considered the potential of using **multiple stages** with **adaptive stepsizes**? Would this lead to better communication round complexity, or do you believe two stages are already sufficient?

3. **Clarification of Equation (13):**

   Can the authors clarify why the bound in **line 297** (Equation 13) holds? Specifically, the reasoning behind using Lemma 3 to bound the right-hand side of (13) is unclear and would benefit from further explanation.

4. **Interpretation of $r_0$:**

   Could the authors explain the meaning of **$r_0$** in **Theorem 1**? This parameter plays a crucial role in the analysis, but its practical interpretation or impact on the algorithm’s behavior is not clearly discussed.

5. **Strange subfigures in Figure 1:**

The left column of Figure 1 is quite strange as only one curve is there. No explanation is provided.

---

> ### Author Response · Authors · 2024-11-20
> **Response 1/2**
>
> Thank you very much for the positive review and helpful criticisms. Below we have responded to the key points of your review:
>
> **W2: Small-scale experiments** Based on your feedback, we have included larger-scale experiments comparing Local SGD and Minibatch SGD for training deep neural networks (image classification with resnets). Our motivation is to verify in this more practical setting that (1) more local steps (up to some threshold) will accelerate Local SGD and (2) Local SGD outperforms Minibatch SGD. Our experiments compare Local SGD to Minibatch SGD when controlling for computation/communication budgets, so that both algorithms use the same number of total gradient computations and communication operations. The results are included in Appendix F of the updated version, and indeed both of the above claims are supported by the results. This suggests that the optimization benefit of local steps could apply for settings beyond the linear model considered in this paper, and we consider this a central direction for future work.
>
> **W3: Comparison against Minibatch SGD** This is an important point. Since the same question was asked by another reviewer, we address this in the general response. Essentially, Minibatch SGD in the deterministic setting is equivalent to Local GD with $K=1$, which we already included in the evaluation. See the general response for a detailed discussion.
>
> **Q1: Relation to self-concordance** This is a great question that requires some investigation. First, we agree that we should mention self-concordance in the paper given the similarity between self-concordance and the properties of the logistic loss which we have leveraged. There are some important differences between our work and the refernce you mentioned [(Bach, 2014)](https://jmlr.org/papers/v15/bach14a.html). First, this paper considers a strongly convex objective, which means that the setting is not compatible with our setting: separable logistic regression is convex, not strongly convex. Also, [(Bach, 2014)](https://jmlr.org/papers/v15/bach14a.html) achieves adaptivity to an unknown strong convexity parameter, which is a different goal from our goal of achieving improved rates by leveraging properties of the loss function. We have updated the paper to reference [(Bach, 2014)](https://jmlr.org/papers/v15/bach14a.html).
>
> The bigger question is: can self-concordance improve convergence rates of Local (S)GD? From the available evidence, we believe that the answer is likely no. To see this, notice that the hard objective from the lower bound of Local SGD from [(Patel et al, 2024)](https://arxiv.org/abs/2405.11667) is quadratic, so it is self-concordant. The fact that Local SGD still converges slowly on this function (the rate is independent of $K$) suggests that self-concordance alone may not be sufficient to imply faster rates of Local SGD than those established in [(Patel et al, 2024)](https://arxiv.org/abs/2405.11667). On the other hand, a previous lower bound [(Woodworth et al, 2020b)](https://arxiv.org/abs/2006.04735) uses a hard instance with unbounded third derivative (the second derivative has a jump discontinuity), so it is certainly not self-concordant, and enforcing self-concordance would remove this hard instance from consideration. Based on these examples, it's possible that self-concordance (or a related property) could play a role in accelerating Local SGD, but it is not a sufficient condition to enable acceleration by local steps. Fully resolving this question is likely to require further investigation.
>
> **Q2: Multi-stage learning rate** It is likely that our analysis can be extended to use a multi-stage learning rate, and it's possible that this could lead to a faster rate. We believe that the analysis of large stepsize GD for single-machine logistic regression (Wu et al, 2024) can also be extended to a multi-stage version. However, we are primarily interested in developing theory for practically relevant algorithms that work for many problems (e.g.  fixed stepsize Local SGD). We see our two-stage Local GD as a close relative of fixed stepsize Local GD, whereas the multi-stage approach is further away. The multi-stage approach may yield a faster rate for the logistic regression problem, but this could also be seen as "overfitting" to this problem in some sense, and is orthogonal to our main goal of theoretically understanding Local GD.

---

> ### Author Response · Authors · 2024-11-20
> **Response 2/2**
>
> **Q3: Proof of Equation (13)** We describe the proof of Equation (13) in our proof sketch (Lines 279-283), and we can elaborate here. Note that in the following, we use $w_r$ to denote the global model at round $r$ due to formatting issues with inline Latex. We should emphasize that Lemma 3 is not used at all in the proof of Equation (13), in fact Equation (13) just describes the objective for any two points $w_1, w_2$. However, Lemma 3 is proven by applying Equation (13), and this is why we reference Equation (13) directly after the statement of Lemma 3. To prove Lemma 3, we apply Equation (13) with $w_1 = w_r$ and $w_2 = w_{r,k}^m$, then we use Lemma 2 to bound those terms in Equation (13) of the form $\lVert w_{r,k}^m - w_r \rVert$. After applying Lemma 2 to get $\lVert w_{r,k}^m - w_r \rVert \leq \eta K F_m(w_r)$, the condition $F(w_r) \leq 1/(\eta K M)$ assumed in Lemma 3 then implies that $F_m(w_r) \leq 1/(\eta K)$, so the bound from Lemma 2 is strengthened to $\lVert w_{r,k}^m - w_r \rVert \leq 1$. This is what we meant by the sentence on Line 296-297. Please let us know if we have cleared this up, and if so we will include this explanation in the updated version.
>
> **Q4: Interpretation of $r_0$** $r_0$ is the number of rounds used in the first stage of Two-Stage Local GD, so that a small learning rate $\eta_1$ will be used for the first $r_0$ rounds, and a larger learning rate $\eta_2$ will be used for the remaining $R-r_0$ rounds.  Theoretically, we need to make sure that $r_0$ is large enough that the first round will decrease the loss (and therefore the smoothness) sufficiently that a larger learning rate can be employed. In practice, choosing $r_0$ is a trade-off between stability and speed: a large $r_0$ will yield stability at the cost of slower optimization due to a small learning rate in the first stage, while a small $r_0$ will increase speed by quickly switching to a larger learning rate, potentially at the cost of stability.
>
> **Q5: Overlapping loss curves** There is a very simple reason why there appears to only be a single curve in the left column of Figure 1. The algorithm performance is nearly identical under different $K$ if the learning rate is chosen as $\eta = 1/(KH)$. That is, the curves for different $K$ are overlapping! This underscores our point that the stepsize choice $\eta = 1/(KH)$ from previous work is overly conservative in practice, and indeed using this stepsize eliminates any potential benefit of local steps in our experiments. We explained these overlapping curves in the updated paper.

---

> ### Comment · Reviewer_zs85 · 2024-11-22
> **Thanks for the responses.**
>
> Thanks for the responses which address almost all of my questions. I appreciate the effort the authors paid in addressing those confusions.
>
> I noticed that the author also revised the paper accordingly. I have a suggestion below. I like the discussion about why self-concordance might not improve convergence rates of Local (S)GD. I suggest the author add this discussion to the paper to help future theoretical studies.
>
> After the rebuttal, I tend to increase my point, because I feel the paper might inspire future study on proposing new conditions in which Local SGD methods can be accelerated.
>
> Given that my initial point is 6 and there is not a 7 score, I increase my score to 8.

---

> > ### Author Response · Authors · 2024-11-22
> > **Thank you for your feedback!**
> >
> > Dear Reviewer zs85,
> >
> > We are glad that our response addresses your concerns. We will incorporate the discussion of the condition of self-concordance into the revised version of the paper.
> >
> > Thank you for reviewing our paper.
> >
> > Best,
> > Authors

---

### Author Response · Authors · 2024-11-20
**General Response**

Thank you to all of the reviewers for your efforts in the review process. We have responded individually to each of your reviews, and we have updated our paper accordingly. To easily point out changes in our submission, all new text is shown in red. Here we summarize the major changes to our paper.

1. In response to Reviewer zs85, we have added deep learning experiments in Appendix F. Our experiments compare Local SGD and Minibatch SGD with different local steps $K$ for training ResNet-50 on a distributed version of CIFAR-10. The results demonstrate that (1) Local SGD converges faster with more local steps (up to a threshold) and (2) Local SGD outperforms Minibatch SGD. These results suggest that the optimization benefit of local steps may extend beyond the logistic regression setting of our paper, and we consider this a central direction for future work.

2. Reviewer bszu pointed out an issue in our proof of Corollary 1, and we have fixed this issue. This fix only required modifying a few lines (Equations 293-297) in order to completely recover our stated results. See our response to the review of bszu for more information.

Also, two reviewers asked why we did not compare against Minibatch SGD in the experiments, so we address that point here. The short answer is that Minibatch SGD in the deterministic setting is equivalent to Local GD with a single local step, so we only need to compare against Local GD with $K=1$. The long answer: In our introduction and discussion sections, we talked about Local SGD vs. Minibatch SGD in the stochastic setting as motivation, whereas our analysis and experiments (of the original submission) only consider Local GD (deterministic setting). We do not include an analogue of Minibatch SGD for the deterministic setting because *the Minibatch SGD algorithm with deterministic gradients is equivalent to Local GD with a single local step, while wasting a huge amount of computation*. Notice that Minibatch SGD is similar to a naive parallelization of SGD, but with the batch size increased by a factor of $K$. When deterministic gradients are used, there is no need for this extra batch size, and using a large batch size just computes the same gradient $K$ times. So to compare against Minibatch SGD in the deterministic setting, we just need to compare against Local GD with $K=1$. We have mentioned this point in the updated paper. Lastly, since our new experiments in Appendix F consider neural networks trained with stochastic gradients, we do compare against Minibatch SGD in this setting.

---

### Meta-Review · Area_Chair_ZkXU · 2024-12-10

**Metareview:**

The paper investigates two variants of Local Gradient Descent (Local GD) applied to distributed binary classification using logistic regression on heterogeneous and linearly separable data. It establishes a faster convergence rate compared to existing guarantees for Local GD on general convex functions, demonstrating that local steps can accelerate convergence. This improvement is achieved by moving beyond worst-case analyses over broad classes of problems defined by convexity, smoothness, and other general assumptions, instead leveraging the specific structure of distributed logistic regression.

Overall, the paper is well-written and presents several interesting ideas. While the setting considered is relatively narrow in scope, the results are compelling and make a significant contribution to the field of distributed optimization.

**Additional Comments On Reviewer Discussion:**

Concerns were raised regarding the limited scope of the results, which focus on logistic regression with linearly separable data. The authors have acknowledged this in the paper as well as in their rebuttal. While this does represent a narrow class of problems, the paper effectively uses this focus to exploit the problem's structure and achieve improved convergence rates. This focused approach could serve as a pathway for refining many worst-case convergence analyses, which often fail to reflect the behavior of algorithms on practical applications.

---

### Decision · Program_Chairs · 2025-01-22

Accept (Poster)